# Scientists' First Exam: Probing Cognitive Abilities of MLLM via Perception, Understanding, and Reasoning

**PrismaX Team**\*

Shanghai Artificial Intelligence Laboratory
Dataset: https://huggingface.co/datasets/PrismaX/SFE
Website: https://prismax.opencompass.org.cn/
Code: https://github.com/PrismaX-Team/sfe

## Abstract

Scientific discoveries increasingly rely on complex multimodal reasoning that integrates information-intensive scientific data and domain-specific expertise. Empowered by expert-level scientific benchmarks, scientific Multimodal Large Language Models (MLLMs) hold the potential to significantly enhance this discovery process in realistic workflows. However, current scientific benchmarks mostly focus on evaluating the knowledge understanding capabilities of MLLMs, leading to an inadequate assessment of their perception and reasoning abilities. To address this gap, we present the Scientists First Exam (SFE) benchmark, designed to evaluate the scientific cognitive capacities of MLLMs through three cognitive levels: *scientific signal perception*, *scientific attribute understanding*, *scientific comparative reasoning*. Specifically, SFE comprises 830 expert-verified VQA pairs across three question types, spanning 66 multimodal tasks across five high-value disciplines. Extensive experiments reveal that current *state-of-the-art* GPT-o3 and InternVL-3 achieve only 34.08% and 26.52% on SFE, highlighting significant room for MLLMs to improve in scientific realms. We hope the insights obtained in SFE will facilitate further developments in AI-enhanced scientific discoveries.

## 1 Introduction

Scientific discoveries rely on investing significant time in analyzing large-scale, complex, and diverse data. Researchers require domain-specific knowledge to interpret scientific data across various modalities, and apply problem-solving skills to address specific scientific challenges [64]. Recent advances in multimodal large language models (MLLMs) have achieved remarkable performance on a wide range of benchmarks, comparable to or even surpassing human-level understanding in both general-level (e.g., MMLU [28], SuperGLUE [67], TriviaQA [31]) and graduate-level (e.g., GPQA [60], HumanEval [12], GSM8K [15]) knowledge domains. As MLLMs continue to progress from general-purpose understanding to domain-specific knowledge, scientific discovery has emerged as a critical frontier for evaluating and extending their abilities [5, 10, 11, 35, 42, 44, 84].

The process of scientific discovery often involves specialized scientific analysis of data modalities (e.g., molecular structures, spectra, protein sequences) from various scientific fields. SuperG-PQA [17] extends conventional domains by incorporating long-tail disciplines, ensuring accessibility to real-world professional expertise. CURIE [16] establishes a ten-task benchmark for evaluating scientific reasoning in long-context scenarios. HLE [59] is introduced to evaluate model capabilities

---

\*Please refer to Appendix A for all team members

39th Conference on Neural Information Processing Systems (NeurIPS 2025) Track on Datasets and Benchmarks.

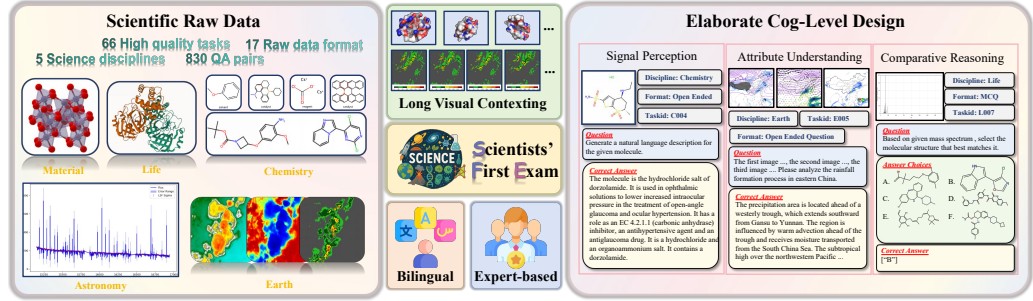

(a) Examples from our proposed SFE. SFE is designed to comprehensively evaluate the scientific capabilities of MLLMs in depth and breadth.

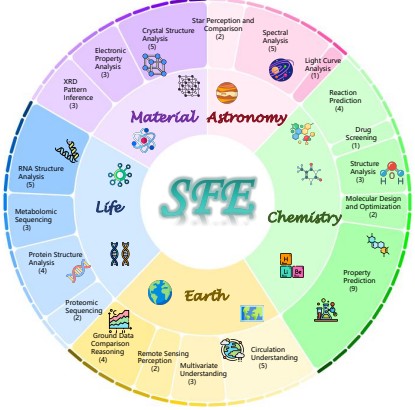

(b) The structure of SFE includes 5 disciplines, 19 scientific directions, and 66 tasks.

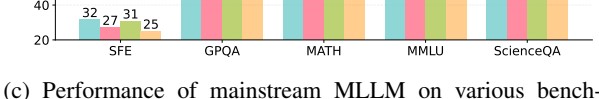

(c) Performance of mainstream MLLM on various benchmarks.

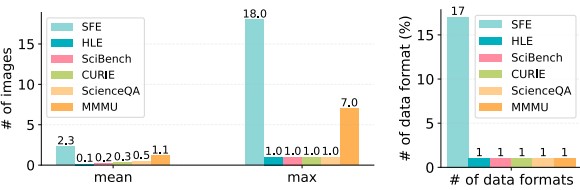

(d) Number of images per QA on various benchmarks.

(e) Number of released data formats.

Figure 1: Overview of the Scientists' First Exam (SFE) benchmark.

through challenging and expert-authored questions. However, despite the growing interest in the scientific domain, most existing scientific benchmarks extract tasks from secondary sources such as academic materials [16, 27, 78] and textbooks [70, 78]. As a result, they inadequately probe the cognitive abilities (e.g., perception, understanding, and reasoning) required for analyzing scientific data encountered in real-world research. Moreover, these benchmarks tend to focus exclusively on a single ability to interpret domain knowledge from the data, while neglecting the full spectrum from perception to reasoning. This capability gap exposes a fundamental challenge: *How to granularly measure MLLMs' scientific cognitive capabilities across multiple disciplines for scientific discovery?*

To bridge this gap, we introduce the Scientists' First Exam (SFE) benchmark, designed to comprehensively evaluate the scientific cognitive capabilities of MLLMs through three cognitive levels (cog-levels): *Scientific Signal Perception* (**L1**) characterizes the capacity to discern critical components within visualizations of scientific raw data; *Scientific Attribute Understanding* (**L2**) demonstrates the ability to interpret domain-expert knowledge; *Scientific Comparative Reasoning* (**L3**) manifests the ability to derive phenomenological insights through structured comparison of multiple scientific visual sources. SFE encompasses **66** expert-curated, high-value multimodal tasks across five disciplines: Astronomy, Chemistry, Earth, Life, and Materials Sciences (Fig. 1b). Each task is constructed from native scientific raw data formats (Fig. 1a) and formulated as visual question answering (VQA) pairs (Fig. 1e), designed to probe specific levels of scientific cognition. All tasks are bilingual (English & Chinese) to support broad accessibility. These tasks are designed not only to require a deep understanding of domain-specific knowledge and data analysis skills but also to significantly enhance research efficiency and facilitate advancements that benefit society.

We benchmark 16 *state-of-the-art* open and closed weight MLLMs using SFE, as illustrated in Fig. 1c. As observed, while these MLLMs perform well on benchmarks such as MMLU [78] and ScienceQA [48], they all exhibit suboptimal results on the SFE benchmark. This indicates that SFE serves as a challenging frontier for scientific MLLM development. Our contributions:

Table 1: Comparison of large language model (LLM) benchmarks related to science. `Astro`, `Chem`, `Phy`, `Bio`, `CS`, `QC`, `Geo`, and `Mat` are abbreviations for Astronomy, Chemistry, Physics, Biology, Computer Science, Quantum Computing, Geospatial Analysis, and Material, respectively. `AD`, `Bus.`, `Sci.`, `Med.`, `HSS`, and `TE` refer to Art & Design, Business, Science, Health & Medicine, Humanities & Social Science, and Tech & Engineering, respectively. The Question Types include MCQ (Multiple-Choice Questions), EM (Exact Match), and OQ (Open Questions). EN/ZH denotes the language of each benchmark (English/Chinese).

| Benchmark | Discipline | Multi-Modal | Question Type | Raw Data | Task #Count | Question Source | Language |
|---|---|---|---|---|---|---|---|
| MMMU[78] | `AD, Bus., Sci. Med., HSS, TE` | ✓ | MCQ OQ | `.png` | ✗ | Textbooks e-Resources | EN |
| SCIENCEQA[48] | Natural, social and language science | ✓ | MCQ | `.png` | ✗ | Curricula | EN |
| SCIBENCH[70] | `Math, Chem Physics` | ✓ | OQ | `.png` | ✗ | Textbooks | EN |
| CMMU[27] | 7 School subjects | ✓ | OQ | `.png` | ✗ | Exams | ZH |
| ChemBench[26] | `Chem` | ✗ | EM | - | 8 | Public datasets | EN |
| SuperGPQA[17] | 13 Disciplines | ✗ | MCQ | - | ✗ | Experts | EN |
| SciEval[63] | `Chem, Phy, Bio` | ✗ | MCQ, EM judgment | - | ✗ | Knowledge base | EN |
| HLE[59] | `Math, Phy, Bio HSS, CS, Chem,TE` | ✓ | MCQ, EM | `.png` | ✗ | Experts | EN |
| CURIE[16] | `Mat, Phy, QC, Geo Bio, Proteins` | ✓ | OQ | `.png` | 10 | Experts | EN |
| **SFE** | `Astro,Chem, Earth Life,Mat` | ✓ | MCQ EM, OQ | `.png,.mgf .sto,.txt`$,^{...}$ | 66 | Experts | EN, ZH |

1. We propose the first benchmark to categorize scientific tasks by cognitive capacity, introducing a three-level taxonomy: Scientific Signal Perception (**L1**), Scientific Attribute Understanding (**L2**), and Scientific Comparative Reasoning (**L3**). This formulation enables fine-grained evaluation of how MLLMs engage with different layers of scientific research.

2. We release the bilingual SFE benchmark, encompassing 66 expert-curated multimodal tasks across five scientific disciplines and covering three question types. All tasks are constructed from native scientific data formats and aligned with three cognitive capacity levels.

3. We comprehensively evaluate 16 state-of-the-art MLLMs, revealing that GPT-o3 achieves the best overall performance, and newer model versions show clear improvements in L3 tasks.

## 2 Related Works

**Science LLMs / MLLMs**. Recent advancements in domain-specific large language models (LLMs) have significantly impacted various scientific fields. In the biomedical domain, LLMs have been employed for tasks such as clinical documentation, information retrieval, and hypothesis generation [35, 36, 49–51, 56]. In chemistry, recent studies have focused on tasks such as drug property prediction, molecular discovery, chemical reaction extraction, and protein structure understanding [11, 39, 40, 43–45, 47, 52, 81, 87]. Notably, researchers have developed LLMs trained on scientific corpora to support molecular and protein discovery [5, 14, 20, 54, 66, 71]. For instance, iupacGPT [14] uses IUPAC nomenclature to effectively capture relationships among atoms and chemical groups. Similarly, Progen [54] and ProtGPT2 [20] are trained for protein sequence generation and understanding. Geosciences have benefited from LLMs through their applications in ocean science, extreme weather, and remote sensing [8, 38, 41, 53, 73, 77, 83, 84]. Models like EarthGPT [83] and CLLMate [38] further integrate multimodal knowledge to support scientific question answering.

**Science Benchmarks**. With the development of large language models (LLMs), recent efforts in LLM benchmarking have increasingly focused on evaluating scientific reasoning capabilities across diverse domains and modalities. Early benchmarks [46] such as ScienceQA [48] and CMMU [27], primarily focus on science level below high school. Several college-level benchmarks [17, 70, 72,

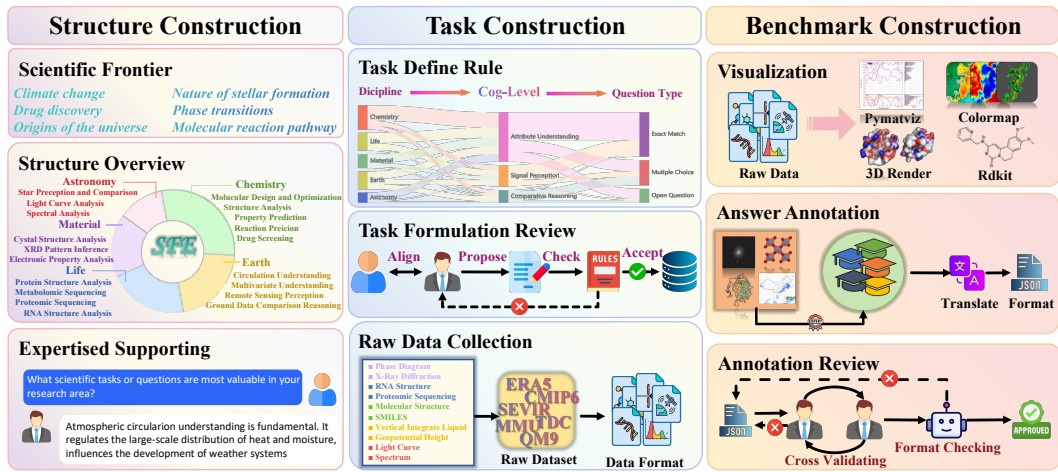

Figure 2: **Data collection framework of SFE**. First, we define 18 scientific directions based on the scientific frontier and domain experts. Building upon this structure, we invite experts to propose tasks and contribute raw data based on three cognitive levels. Finally, we employ visualization techniques and further engage experts to annotate the resulting benchmark.

78, 79] have emerged to support deeper scientific understanding. As state-of-the-art multimodal LLMs rapidly improve, many of them now achieve strong performance on these benchmarks. This has motivated the creation of more expert-level datasets that feature high-difficulty questions such as HLE [59] and CURIE [16]. However, although some benchmarks offer broad domain coverage, they often lack clearly defined tasks, making it difficult to assess specific model limitations. To address this, works like ChemLLMBench [26] and CURIE [16] have introduced smaller-scale but task-oriented benchmarks that enable more targeted evaluation of scientific capabilities. Nevertheless, nearly all of these benchmarks remain monolingual, limiting their applicability for comprehensively evaluating LLMs in global deployment contexts.

## 3 SFE Dataset and Tasks

The SFE benchmark consists of 830 multi-modal VQA pairs spanning 66 real-world scientific tasks across five disciplines: Astronomy, Chemistry, Earth, Life, and Materials Science (Fig. 1b). Each task is constructed from native disciplinary data formats such as molecular structures, spectra, and radar charts, and is carefully annotated by domain experts. On average, each question contains 2.3 scientific images (ranging from 1 to 18), and the dataset supports bilingual prompts and answers in both English and Chinese. In total, SFE spans 17 distinct scientific data formats and is motivated by scientific frontiers such as drug discovery,

Table 2: **Statistics of SFE**.

| Statistic | Number |
|---|---|
| # of VQA | 830 |
| # of Scientific Tasks | 66 |
| # of scientific signal perception VQA | 202 |
| # of scientific attribute understanding VQA | 503 |
| # of scientific comparative reasoning VQA | 125 |
| # of MCQ | 284 |
| # of Exact Match | 420 |
| # of Open Question | 126 |
| Average Question Tokens Length | 88 (en) / 86 (zh) |
| Average Answer Tokens Length | 100 (en) / 106 (zh) |
| Average # of Images | 2.3 (1-18) |
| Released # Scientific Data Format | 17 |

celestial body radial velocity estimation and peptide sequence inference, etc. Detailed dataset statistics are presented in Table 2, and a complete task list is provided in Appendix Table 6.

### 3.1 Data Collection

The construction of SFE is the result of extensive collaboration with domain experts across multiple scientific disciplines and research directions. As shown in Fig. 2, the overall data collection pipeline comprises three key stages: (1) **Structure Construction**, where we collaborate with experts to identify high-value challenges and define scientific directions; (2) **Task Construction**, where the scientific directions are instantiated into concrete tasks with specific question types and cognitive

levels through expert-driven design and review; (3) **Benchmark Construction**, where scientific raw data are rendered, visualized, and used to construct expert-authored VQA with high-quality samples.

**Structure Construction.** The SFE is initiated by identifying foundational scientific directions across five core disciplines: Astronomy, Chemistry, Earth, Life, and Materials Science. These fields are selected for their broad scientific relevance and the need for specialized knowledge. Based on the experts' consultations, we define a structured set of scientific directions, such as circulation understanding in Earth and reaction prediction in chemistry. These directions serve as the foundation for downstream task formulation and reflect current research frontiers.

**Task Construction.** Building upon the established scientific directions, domain experts collaboratively define concrete tasks. We first formalize the task definition rules by mapping each scientific direction to an appropriate cognitive level and question type as shown in Fig. 2. This taxonomy ensures that each task is aligned with a distinct level of cognitive capacity and supports evaluation diversity. For each discipline, we collaborated with 2-5 experts to define benchmark tasks that align with high-priority issues in real-world scientific research. Each expert-designed task is expected to satisfy the following core criteria: (1) It reflects a meaningful problem that requires substantial domain knowledge and is commonly encountered in real-world research. (2) It must be solvable through expert-level reasoning that necessarily integrates multimodal scientific inputs such as structured visualizations and symbolic representations. (3) It aligns with one of the proposed cognitive levels (L1-L3). To support each task, experts also provide instructions for collecting raw data that would serve as visual inputs for the VQA pairs. These data sources span a diverse range of formats, including public datasets (e.g., ERA5 [29]), existing benchmarks (e.g., MoleculeNet [75]), and domain-specific databases (e.g., RCSB PDB [7], PDBbind [69], and PubChem [33]). For each task, domain experts identify suitable data entries that are typically used to analyze the corresponding scientific problems, such as RNA sequence files (.stockholm) and protein structure files (.pdb).

**Benchmark Construction.** In the final stage, each task is instantiated into a set of VQA pairs. To ensure accessibility and visual coherence, all data formats are rendered into image form. During the answer annotation phase, experts construct each VQA pair by composing: (1) rendered multimodal input, (2) a scientifically meaningful question, and (3) an answer grounded in expert reasoning. Then, all VQA pairs are translated into both Chinese and English to provide multilingual support and formatted in JSON for standardization. Finally, a two-stage validation process is applied for quality control. Each VQA pair is first cross-reviewed by domain experts for scientific correctness, clarity, and alignment with the intended reasoning level. This is followed by rule-based validation for format checking. Only VQA pairs that pass both stages are included in the final benchmark.

## 3.2 Tasks

**Astronomy**. In Astronomy, analyzing diverse modalities such as spectra is essential for a wide range of scientific tasks, including property estimation and event detection. Therefore, we design 8 tasks to assess the cognitive abilities of MLLMs from 3 cognitive levels. L1 tasks include *galaxy morphology classification*, where MLLMs are required to differentiate perceptual features of galaxies based on the provided image. Second, L2 tasks include *surface temperature estimation*, *gravitational constant estimation*, *light curve classification*, *metallicity estimation*, etc. For example, in the *metallicity estimation* task, MLLMs need to infer the total metal abundance of a target celestial object based on its observed spectrum. Third, L3 tasks target *transient detection*. Given the pre-transient and post-transient images, along with difference images, MLLMs are expected to determine whether a transient has occurred during the process. Refer to the Appendix for more details.

**Chemistry**. We formulate 19 tasks in the Chemistry domain, spanning three cognitive levels to evaluate MLLMs' understanding of molecular structures, properties, and interactions. L1 tasks include *elemental composition recognition*, *molecular description generation*, etc. In the *molecular description generation* task, MLLMs are required to produce descriptions of specific molecules, highlighting key attributes such as types of chemical bonds and the number of carbon atoms. L2 tasks involve *Lipinski drug-likeness estimation*, *absorption property prediction*, *distribution property prediction*, etc. For example, in the *absorption property prediction* task, MLLMs need to estimate properties of molecules, such as the plasma protein binding rate. L3 tasks include *virtual screening*, *protein-ligand binding affinity prediction*, etc. In the *virtual screening* task, MLLMs are

expected to identify all molecules capable of binding to a given structure. Refer to the Appendix for more details.

**Earth**. To systematically evaluate the performance of MLLMs in the Earth science domain, we construct 14 tasks across three cognitive levels based on diverse weather variables and data sources. First, L1 tasks include *thermocline depth recognition, perception of extreme precipitation distribution*, *SAR image grounding*, etc. For example, in the *perception of extreme precipitation distribution* task, MLLMs are required to identify multiple locations of extreme precipitation. Second, L2 tasks include *moisture source understanding*, *precipitation event analysis*, *convective weather types identification*, etc. For example, the *precipitation event analysis* requires models to analyze the formation process of precipitation by using information from geopotential height, moisture flux, vertical velocity, etc. Third, L3 tasks consist of *differential prediction comparison*, *temperature sequence comparison*, etc. In the *temperature sequence comparison* task, MLLMs need to compare annual temperature series from two different time periods and describe differences in statistical characteristics. Refer to the Appendix for more details.

**Life**. In the life science domain, we construct 14 tasks across three cognitive levels to comprehensively evaluate MLLMs, focusing on modalities such as RNA structures and protein sequences. L1 tasks include *fragment ion peaks count*, *protein chain count*, *small molecule count*, etc. For example, in the *fragment ion peaks count task*, MLLMs are asked to determine the number of specific ions present in a given MS/MS spectrum. L2 tasks consist of *molecular composition inference*, *specified protein detection*, etc. For instance, in the *molecular composition inference* task, MLLMs need to infer the elemental composition of the compound based on the provided spectrum. L3 tasks include *spectrum matching*, where MLLMs are required to identify the molecular structure that best corresponds to a given mass spectrum. Refer to the Appendix for more details.

**Materials**. To evaluate MLLMs' performance in the realm of materials, we formulate 11 tasks spanning three cognitive levels. L1 tasks include *atomic composition description*, *crystal group identification*, *crystal formula determination*, etc. For example, in the *atomic composition description* task, models are required to select relevant properties of a target lattice from multiple descriptions. Second, L2 tasks cover *band gap classification*, *stability estimation*, *energy band and DOS interpretation*. For example, in the *energy band and DOS interpretation* task, MLLMs are required to infer whether a given material belongs to a metal or a semiconductor and estimate its band gap. Third, L3 tasks involve complicated comparative reasoning, such as *phase identification*. Specifically, given the XRD pattern of a composite material and candidate substances, MLLMs are required to identify three materials that form the composite material. Refer to the Appendix for more details.

## 4 Experiments and Evaluations

**General Settings.** We conduct a comprehensive evaluation of the *state-of-the-art* MLLMs on SFE. For models with open weights, we assess InternVL2.5-78B [13], InternVL3-78B [86], Qwen2.5-VL-72B [6], LLaMaVision-90B [22], and LLaVa-Onevision-72B [37]. For models with closed weights, we evaluate GPT-4o-2024-11-20 [57], GPT-4.1-2025-04-14 [57], GPT-o1-2024-12-17 [57], GPT-o3 [57], Claude 3 Opus [4], Claude 3.7 Sonnet [4], Gemini-2.0-Flash [65], Gemini-2.5-Flash [65], Gemini-2.5-Pro [65], Grok-2-Vision-12-12 [76] and Doubao-1.5-Vision-Pro [24]. When benchmarking, we configure all MLLMs' temperatures to 0 for reduced randomness and employ a standard zero-shot prompt template across all tasks. Specifically, the template begins with a description of the task assigned to the model, followed by the inclusion of question texts with interleaved images. Additionally, we fix the maximal number of generated tokens to 1024, ensuring fairness and cost-effectiveness in our evaluations.

**Metrics.** We present the BERTScore [82] and the LLM-as-a-Judge score [23] for all tasks, except for the remote sensing perception task in Earth science, where we report the execution success rate and the Intersection over Union (IoU). For BERTScore, we use the F1 score. For LLM-as-a-Judge score, we employ GPT-4o-2024-11-20 as the judge, allowing us to semantically verify the correctness of answers against model predictions. Conversely, the execution success rate is utilized to assess whether MLLMs can accurately follow prompts to produce bounding boxes in the desired format. Finally, the IoU metric evaluates the precision of these bounding boxes compared to ground truths. Without further clarification, each experiment is conducted once to obtain the final results.

Table 3: Experimental results of all models on different disciplines using different languages. The LLM-as-a-Judge score is used as the evaluation metric. 'Average' represents the mean score.

| Model | Astronomy | | Chemistry | | Earth | | Life | | Material | | Average | |
|---|---|---|---|---|---|---|---|---|---|---|---|---|
| | en | zh | en | zh | en | zh | en | zh | en | zh | en | zh |
| *Closed Weight MLLMs* | | | | | | | | | | | | |
| Grok-2-Vision-12-12 | 19.37 | 18.23 | 21.29 | 20.20 | 33.58 | 36.23 | 25.10 | 25.95 | 35.57 | 36.89 | 24.97 | 25.10 |
| GPT-4o-2024-11-20 | 20.38 | 18.04 | 21.60 | 19.25 | 32.65 | 29.93 | 30.39 | 27.58 | 49.67 | 48.36 | 27.15 | 24.72 |
| GPT-4.1-2025-04-14 | 24.18 | **25.50** | 24.01 | 22.11 | 40.40 | **44.43** | **34.90** | **33.66** | 47.70 | 48.85 | 30.88 | 31.05 |
| GPT-o1-2024-12-17 | 22.03 | 21.84 | 27.41 | 27.21 | 38.61 | 36.56 | 33.99 | 31.31 | 61.15 | **61.64** | 32.19 | 31.24 |
| GPT-o3 | 24.24 | 23.80 | **28.91** | **27.89** | **43.05** | 36.29 | 33.59 | 31.57 | **63.44** | 58.20 | **34.08** | **31.60** |
| Gemini-2.0-Flash | 16.14 | 12.78 | 27.82 | 24.69 | 34.24 | 32.91 | 32.48 | 27.32 | 52.79 | 50.49 | 29.49 | 26.33 |
| Gemini-2.5-Flash | 24.30 | 24.11 | 23.67 | 23.47 | 31.99 | 30.53 | 25.03 | 25.10 | 56.39 | 55.90 | 28.03 | 27.63 |
| Gemini-2.5-Pro | 5.13 | 6.08 | 2.07 | 2.28 | 2.52 | 3.84 | 19.73 | 22.35 | 28.69 | 27.70 | 8.04 | 8.96 |
| Claude-3-Opus | 14.68 | 16.08 | 19.63 | 17.45 | 36.62 | 32.12 | 22.55 | 23.66 | 36.72 | 32.13 | 23.64 | 22.15 |
| Claude-3.7-Sonnet | 25.89 | 22.34 | 27.79 | 25.14 | 38.21 | 37.09 | 31.24 | 29.48 | 49.51 | 46.72 | 31.62 | 29.23 |
| Doubao-1.5-vision-pro | **28.35** | 23.99 | 25.00 | 24.46 | 26.16 | 25.83 | 30.07 | 29.35 | 51.48 | 46.39 | 28.79 | 27.17 |
| *Open Weight MLLMs* | | | | | | | | | | | | |
| Qwen2.5-VL-72b | 26.46 | 20.89 | 18.33 | 15.27 | 25.03 | 24.57 | 25.29 | 22.75 | 41.47 | 42.46 | 24.17 | 21.51 |
| InternVL-2.5-78B | 15.76 | 17.09 | 18.54 | 15.58 | 34.77 | 37.22 | 27.06 | 25.49 | 43.11 | 39.84 | 24.43 | 23.54 |
| InternVL-3-78B | 27.09 | 24.62 | 19.80 | 16.29 | 28.81 | 28.94 | 30.85 | 27.58 | 40.98 | 42.30 | 26.52 | 24.30 |
| Llama-3.2-Vision-90B | 20.63 | 18.04 | 16.94 | 15.14 | 27.22 | 30.53 | 25.29 | 21.37 | 32.30 | 31.80 | 22.26 | 20.95 |
| Llava-OneVision-72B | 25.19 | 23.23 | 15.34 | 15.17 | 23.31 | 27.81 | 24.58 | 24.58 | 37.54 | 42.62 | 22.10 | 23.39 |
| Average | 21.23 | 19.79 | 21.14 | 19.51 | 31.07 | 30.93 | 28.27 | 26.82 | 45.54 | 44.52 | 26.15 | 24.93 |

Table 4: Experimental results of all models on different cognitive levels and different question types in both Chinese and English.

| Model | L1 | | L2 | | L3 | | Exact Match | | | | Open Question | | | | MCQ | |
|---|---|---|---|---|---|---|---|---|---|---|---|---|---|---|---|---|
| | | | | | | | IoU | | LLM score | | Bertscore | | LLM score | | LLM score | |
| | en | zh | en | zh | en | zh | en | zh | en | zh | en | zh | en | zh | en | zh |
| *Closed Weight MLLMs* | | | | | | | | | | | | | | | | |
| Grok-2-Vision-12-12 | 28.31 | 25.66 | 23.10 | 25.81 | 27.44 | 21.44 | 6.11 | 8.30 | 17.32 | 17.39 | 0.707 | 0.718 | 35.48 | 35.0 | 32.68 | 33.13 |
| GPT-4o-2024-11-20 | 35.50 | 32.70 | 26.46 | 23.32 | 17.28 | 18.32 | 2.66 | 2.13 | 20.42 | 17.58 | 0.698 | 0.710 | 36.74 | 39.23 | 33.87 | 30.21 |
| GPT-4.1-2025-04-14 | 35.98 | 36.61 | 30.02 | **29.64** | 26.64 | 28.32 | 5.19 | 3.47 | 22.10 | 20.14 | **0.714** | **0.723** | 46.35 | 44.13 | 38.49 | **42.74** |
| GPT-o1-2024-12-17 | **43.65** | 42.80 | 29.09 | 26.60 | 27.36 | **32.40** | 0.77 | 1.83 | 22.61 | 21.82 | 0.70 | 0.71 | 44.81 | 44.62 | 42.04 | 40.56 |
| GPT-o3 | 42.54 | 41.27 | **30.30** | 28.27 | **36.48** | 31.60 | 5.17 | 5.94 | 23.75 | 22.12 | 0.694 | 0.693 | **48.85** | 44.33 | **44.26** | 41.27 |
| Gemini-2.0-Flash | 41.43 | 36.46 | 26.86 | 23.70 | 22.00 | 21.60 | 0.91 | 0.44 | 21.38 | 19.28 | 0.70 | 0.65 | 35.67 | 25.19 | 39.47 | 37.39 |
| Gemini-2.5-Flash | 37.04 | 37.57 | 27.89 | 26.98 | 14.96 | 15.20 | 0.99 | 0.96 | **24.45** | **24.45** | 0.688 | 0.688 | 30.00 | 30.10 | 32.71 | 31.51 |
| Gemini-2.5-Pro | 17.46 | 18.78 | 6.16 | 6.94 | 1.36 | 2.24 | - | - | 8.11 | 8.60 | 0.548 | 0.558 | 12.50 | 15.96 | 6.30 | 6.94 |
| Claude-3-Opus | 25.71 | 23.54 | 23.80 | 22.09 | 19.84 | 20.32 | 0.87 | 0.08 | 15.99 | 14.71 | 0.700 | 0.708 | 30.58 | 30.87 | 32.64 | 30.21 |
| Claude-3.7-Sonnet | 37.46 | 33.39 | 29.80 | 28.15 | 30.08 | 27.28 | 2.27 | 1.38 | 21.40 | 19.09 | 0.710 | 0.671 | 46.06 | **46.73** | 41.76 | 38.13 |
| Doubao-1.5-vision-pro | 32.70 | 32.43 | 26.48 | 24.31 | 32.16 | 30.72 | 2.19 | 11.84 | 23.50 | 21.00 | 0.677 | 0.691 | 34.62 | 33.85 | 34.65 | 34.05 |
| *Open Weight MLLMs* | | | | | | | | | | | | | | | | |
| Qwen2.5-VL-72b | 31.11 | 26.08 | 23.36 | 21.91 | 16.96 | 12.96 | **24.49** | **15.37** | 20.49 | 17.55 | 0.645 | 0.667 | 20.58 | 22.12 | 31.06 | 27.25 |
| InternVL-2.5-78B | 25.08 | 22.91 | 25.33 | 24.35 | 19.84 | 21.20 | 3.79 | 3.65 | 16.53 | 14.92 | 0.690 | 0.692 | 36.06 | 36.06 | 32.11 | 31.97 |
| InternVL-3-78B | 26.67 | 27.30 | 26.10 | 22.82 | 28.00 | 25.99 | 5.25 | 7.16 | 19.39 | 15.99 | 0.69 | 0.70 | 37.5 | 38.85 | 33.27 | 31.51 |
| Llama-3.2-Vision-90B | 26.14 | 24.23 | 22.33 | 20.34 | 16.16 | 18.48 | 0.27 | 0.25 | 15.90 | 14.17 | 0.70 | 0.68 | 29.23 | 29.71 | 29.33 | 27.99 |
| Llava-OneVision-72B | 25.13 | 26.40 | 19.14 | 20.68 | 29.52 | 29.76 | 2.84 | 1.42 | 14.99 | 15.87 | 0.700 | 0.665 | 32.88 | 31.54 | 28.91 | 31.76 |
| Average | 32.00 | 30.49 | 24.76 | 23.49 | 22.89 | 22.34 | 4.26 | 4.25 | 19.27 | 17.79 | 0.689 | 0.683 | 34.87 | 34.27 | 33.35 | 32.29 |

## 4.1 Main Results

We evaluate the performance of 16 MLLMs across five scientific disciplines and two languages (English and Chinese) on SFE using the LLM-as-a-Judge score for evaluation metric. Our results in Table 3 demonstrate that SFE is capable of revealing fine-grained differences in model capabilities, visual grounding quality, and multilingual reasoning robustness. Below, we detail key observations.

**Observation 1. SFE is capable of distinguishing model capability across both proprietary and open-source MLLMs.** Among all evaluated models, GPT-o3 achieves the best overall performance, with an average score of 34.08% in English and 31.60% in Chinese. It is consistently strong across all disciplines, particularly in Earth and Materials, where scientific visual interpretation and structured multimodal reasoning are often required. Notably, Gemini-2.5-Pro registers the lowest average score among all evaluated models, with only 8.04% in English and 8.96% in Chinese

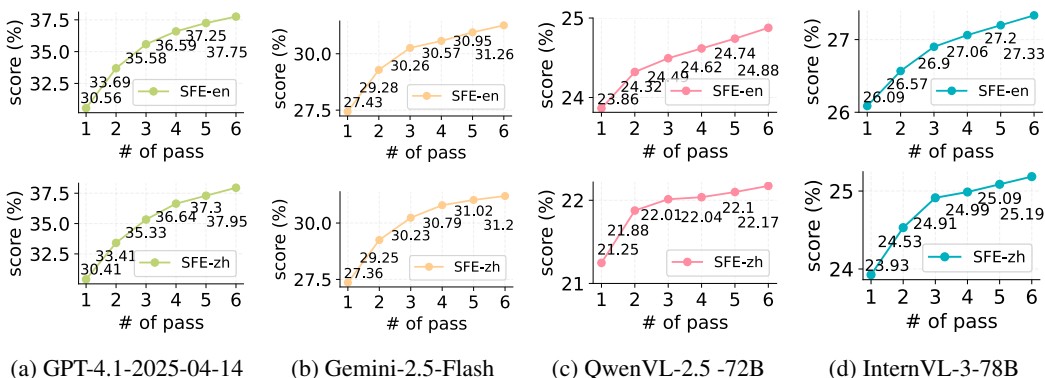

Figure 3: Pass@k scores of four *state-of-the-art* MLLMs on SFE. Closed-weight MLLMs demonstrate superior initial performance and greater scalability than open weight MLLMs.

due to excessive thinking, causing exhausted token budgets and clipped answers[2]. On the other hand, while GPT-o3 reasons as well, it balances reasoning and token usage effectively. The performance gap between GPT-o3 and Gemini-2.5-Pro exceeds 26% on average, showing SFE is comprehensive enough to differentiate the ability across the full range of models. The results also present a systematic divide between closed-weight proprietary models and open-weight models. On average, the best proprietary models (GPT-o3, GPT-o1-2024-12-17, Claude-3.7-Sonnet) outperform the strongest open alternatives (InternVL-3-78B) by 6-8%. Additionally, within SFE, the model series shows clear internal progress. For example, Claude-3.7-Sonnet outperforms Claude-3-Opus by over 7% in both English and Chinese, reflecting measurable architectural or training gains. A similar pattern is also observed within the InternVL series.

**Observation 2. SFE exhibits a clear performance gap of MLLMs between disciplines.** Our findings reveal that across nearly all models, Material Science emerges as the most tractable domain. The top model, GPT-o3, reaches 63.44% in English and 58.20% in Chinese on it. And even open models like Qwen2.5-VL-72b and InternVL-3-78B achieve over 40% in this domain. This trend reflects the relatively structured visual inputs (*e.g.*, phase diagram, X-Ray diffraction) which require the model to generate structured scientific outputs based on symbolic visual images that are aligned with current models' comparatively strong skills. In contrast, Astronomy tasks present more substantial challenges. This domain involves spectral analysis tasks, where models estimate numerical astrophysical parameters (*e.g.*, temperature and velocity) from raw or noisy spectral visualizations, which current MLLMs find challenging. This highlights SFEs role in diagnosing which types of scientific reasoning MLLMs can currently handle and which are still out of reach.

**Observation 3. SFE reveals a potential shift in MLLM capabilities from knowledge understanding to high-order reasoning.** Through SFE's three-level cognitive framework, its results provide an analysis of how recent models' cognitive improvements are distributed. Our findings (in Table 4) show that newer MLLMs exhibit notably higher performance on L3 tasks compared to earlier models, while their L2 performance remains largely similar. This trend aligns with the adoption of advanced reasoning techniques in state-of-the-art models. For example, GPT-o3 improves L3 performance from 26.64% (GPT-4.1-2025-04-14) to 36.48% without an obvious increase in L2 scores (from 30.30% to 30.02%). This pattern reflects OpenAI's reports that emphasize scaling reinforcement learning for reasoning and learning tool-use strategies rather than knowledge expansion. Similarly, InternVL-3 outperforms InternVL-2.5 by 8% in L3 performance in English, despite only marginal gains in L2. This improvement can be attributed to InternVL-3's architectural and training advances, particularly its native multimodal pretraining and Mixed Preference Optimization (MPO) to support Chain-of-Thought reasoning, which is a critical skill for success in L3 tasks such as comparative diagram analysis. The negligible gains in L2 tasks suggest that these improvements arise not from broader knowledge acquisition but from enhanced training innovations.

---

[2]Experiments with more generated tokens are conducted in Table 17 and Table 18 in the appendix.

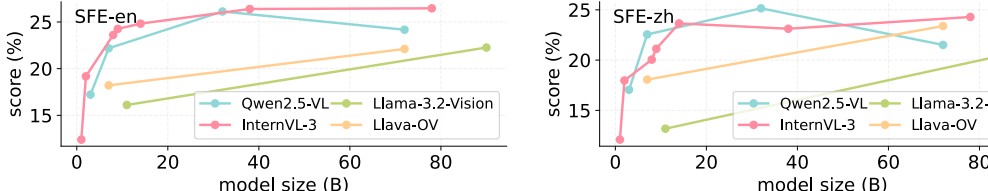

Figure 4: The scaling law of model size on SFE. Clearly, the amount of scientific data has not been scaled proportionally with the increase in model size, resulting in minor overfit for QwenVL-2.5-72B.

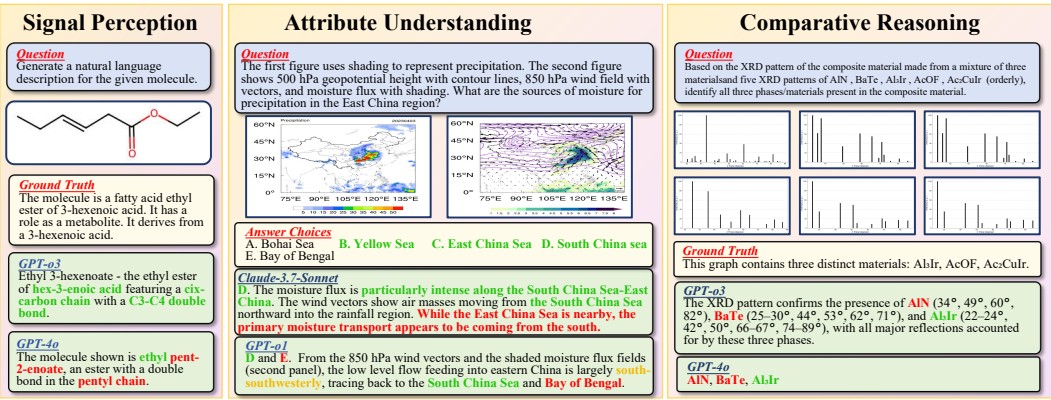

Figure 5: **Case studies** on SFE across different cognitive levels.

## 4.2 Analysis

**Pass@k Analysis.** The Pass@k metric [34] selects the highest quality answers from an MLLM as its final response to a question, indicating the models potential for improvement through post-training (*e.g.*, RLHF [58], GRPO [25], etc.). In Fig. 3, we evaluate pass@k scores of GPT-4.1-2025-04-14, Gemini-2.5-Flash, Qwen2.5-VL-72b and InternVL-3-78B on SFE, with $k$ ranging from 1 to 6. As shown, GPT-4.1-2025-04-14 and Gemini-2.5-Flash outperform *state-of-the-art* open-weight MLLMs by not only exhibiting superior initial performance (30.56% v.s. 26.09%) but also demonstrating strong scalability (30.56% → 37.75% v.s. 26.09 → 27.33%). This suggests that closed-weight MLLMs may leverage more diverse and expansive raw datasets during pre-training than open-weight MLLMs. Furthermore, their post-training phase may prioritize a balanced approach, emphasizing exploration alongside exploitation, rather than focusing exclusively on exploitation [80].

**Scaling Law of Model Size.** We benchmarked MLLMs of varying sizes to evaluate their impact on SFE, as depicted in Fig. 4. Although Llama-3.2-Vision and Llava-Onevision series improve with size, they underperform compared to the Qwen2.5-VL and InternVL series. The larger models, Qwen2.5-VL-72B and InternVL-3-78B, do not significantly surpass their smaller counterparts, indicating a lack of proportional scientific data scaling during pre-training. Additionally, Qwen2.5-VL-72Bs performance is lower than Qwen2.5-VL-7B, suggesting potential overfitting. This emphasizes the need for balanced data scaling relative to model size in the scientific domain.

**Impact on Temperatures.** We analyze the effect of temperature settings on MLLMs' performance, as shown in Table 5. The results indicate that both excessively high and low temperature values can lead to performance degradation for scientific discoveries. Empirically, maintaining the temperature within the

Table 5: Scores on different temperatures.

| Model | 0.0 | 0.2 | 0.4 | 0.6 | 0.8 | 1.0 |
|---|---|---|---|---|---|---|
| GPT-4.1-2025-04-14 | 30.88 | 31.31 | 31.59 | **32.09** | 31.97 | 31.63 |
| InternVL-3-78B | 24.43 | **26.98** | 26.93 | 26.55 | 25.86 | 23.12 |

range of 0.4 to 0.6 optimally balances the trade-off between exploration and exploitation, thereby enhancing MLLMs' efficacy.

**Case Study.** Fig. 5 presents case studies evaluating the performance of MLLMs on SFE across different cognitive levels. In the signal perception task, both GPT-4o-2024-11-20 and GPT-o3

effectively recognize functional groups within molecular images. However, GPT-o3, equipped with reasoning capabilities, offers more precise results for fine-grained tasks such as counting carbon atoms. For attribute understanding, Claude-3.7-Sonnet and GPT-o1-2024-12-17 accurately identify colormaps and wind vectors in geographical images. Despite achieving approximately one-third accuracy in their responses, GPT-o1-2024-12-17 falsely identifies the `Bay of Bengal` with its wind strength misinterpreted, suggesting an incomplete geographical understanding. In comparative reasoning, GPT-o3 integrates information from multiple images but struggles with comparative selection, revealing an incomplete reasoning chain. Conversely, across similar tasks, MLLMs like GPT-4o-2024-11-20, Grok-2-Vision-12-12, and others primarily focus on the initial four images, indicating limitations in processing long visual contexts.

## 5 Conclusion

In this paper, we propose the Scientists First Example (SFE) benchmark, aiming to provide a granular assessment of MLLMs' scientific cognitive capabilities from perception to reasoning. SFE includes 830 expert-verified VQA pairs across 66 tasks in five high-value disciplines, addressing critical needs for more rigorous and diverse evaluation tools for scientific MLLMs. Extensive experiments reveal insights that could contribute to advancements in AI-driven scientific discoveries.

However, the benchmarks scope and depth could be further enhanced, which we intend to improve in future work. Furthermore, while SFE has the potential to significantly advance these discoveries by offering a robust evaluation framework, it also raises concerns about increasing reliance on AI in scientific research. This might inadvertently undermine the value of human intuition and creativity.

## Acknowledgments and Disclosure of Funding

This work is supported by Intern Discovery.

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

# A    PrismaX Team

We provide the full list of PrismaX team members as follows:

- Yuhao Zhou[3*] (Shanghai Artificial Intelligence Laboratory & Engineering Research Center of Machine Learning and Industry Intelligence, Ministry of Education & Sichuan University)
- Yiheng Wang[3*] (Shanghai Artificial Intelligence Laboratory & Shanghai Jiao Tong University)
- Xuming He[3*] (Shanghai Artificial Intelligence Laboratory & Zhejiang University)
- Ao Shen[4] (Shanghai Artificial Intelligence Laboratory & Fudan University)
- Ruoyao Xiao (Chinese Academy of Meteorological Sciences)
- Zhiwei Li (Shanghai Artificial Intelligence Laboratory)
- Qiantai Feng (Shanghai Artificial Intelligence Laboratory)
- Zijie Guo (Shanghai Artificial Intelligence Laboratory & Fudan University)
- Yuejin Yang (Shanghai Artificial Intelligence Laboratory & Research Institute of Intelligent Complex Systems, Fudan University)
- Hao Wu (Shanghai Artificial Intelligence Laboratory)
- Wenxuan Huang (Shanghai Artificial Intelligence Laboratory)
- Jiaqi Wei (Shanghai Artificial Intelligence Laboratory & Zhejiang University)
- Dan Si (Sichuan University)
- Xiuqi Yao (Shanghai Artificial Intelligence Laboratory)
- Jia Bu (East China Normal University)
- Haiwen Huang (East China Normal University)
- Tianfan Fu (Shanghai Artificial Intelligence Laboratory & Nanjing University)
- Shixiang Tang (Shanghai Artificial Intelligence Laboratory)
- Ben Fei (Shanghai Artificial Intelligence Laboratory)
- Dongzhan Zhou (Shanghai Artificial Intelligence Laboratory)
- Fenghua Ling (Shanghai Artificial Intelligence Laboratory)
- Yan Lu (Shanghai Artificial Intelligence Laboratory)
- Siqi Sun (Fudan University & Shanghai Artificial Intelligence Laboratory )
- Chenhui Li (East China Normal University)
- Guanjie Zheng (Shanghai Jiao Tong University)
- Jiancheng Lv (Sichuan University)
- Wenlong Zhang[†] (Shanghai Artificial Intelligence Laboratory)
- Lei Bai[†] (Shanghai Artificial Intelligence Laboratory)

[†] Corresponding author: zhangwenlong@pjlab.org.cn, bailei@pjlab.org.cn

---

[3]This work was primarily conducted during the authors internship at the Shanghai Artificial Intelligence Laboratory.

[4]Ao Shen is served as an external advisor only.

# B Tasks Overview

Table 6: Details of tasks in SFE.

| Discipline | Task | TaskID | Subtask | Question Type |
|---|---|---|---|---|
| **Astronomy (8)** | Star Perception and Comparison | A001 | Galaxy morphology classification | MCQ |
| | | A011 | Transient detection | MCQ |
| | Light Curve Analysis | A003 | Light curve classification | MCQ |
| | Spectral Analysis | A006 | Surface temperature estimation | Exact Match |
| | | A007 | Gravitational constant estimation | Exact Match |
| | | A008 | Metallicity estimation | Exact Match |
| | | A009 | Alpha-element abundance estimation | Exact Match |
| | | A010 | Radial velocity estimation | Exact Match |
| **Chemistry (19)** | Structure Analysis | C001 | Elemental composition recognition | Exact Match |
| | | C003 | IUPAC name recognition | Exact Match |
| | | C004 | Molecular description generation | Open Question |
| | Property Prediction | C005 | Lipinski drug-likeness estimation | Exact Match |
| | | C007 | Topological polar surface area calculation | Exact Match |
| | | C023 | Absorption property prediction | Exact Match |
| | | C024 | Distribution property prediction | Exact Match |
| | | C025 | Metabolism property prediction | Exact Match |
| | | C026 | Excretion property prediction | Exact Match |
| | | C027 | Toxicity property prediction | MCQ |
| | | C028 | QM9 quantum chemical property prediction | Exact Match |
| | | C029 | Protein-ligand binding affinity prediction | Exact Match |
| | Reaction Prediction | C009 | Product SMILES prediction | Exact Match |
| | | C010 | Reaction classification | MCQ |
| | | C012 | Reactant molecular recognition | Exact Match |
| | | C013 | Reaction condition and catalyst prediction | MCQ |
| | Molecular Design and Optimization | C006 | Synthetic accessibility estimation | Exact Match |
| | | C018 | Molecular property optimization | MCQ |
| | Drug screening | C030 | Virtual screening | MCQ |
| **Earth (14)** | Circulation Understanding | E004 | perception of extreme precipitation distribution | MCQ |
| | | E006 | Moisture source attribution | MCQ |
| | | E007 | subtropical high ridge control region | MCQ |
| | | E008 | thermocline depth recognition | MCQ |
| | | E017 | Outlier analysis | MCQ |

| Discipline | Task | Subtask ID | Subtask | Question Type |
|---|---|---|---|---|
| | Multivariate Understanding | E005 | Precipitation event analysis | Open Question |
| | | E021 | Convective weather types identification | MCQ |
| | | E022 | Convective influence regions identification | MCQ |
| | Remote Sensing Perception | E011 | SAR image grounding | Exact Match |
| | | E012 | Infrared image grounding | Exact Match |
| | Ground Data Comparison Reasoning | E014 | temperature sequence comparison | MCQ |
| | | E016 | Vertical profile comparison | MCQ |
| | | E018 | differential prediction comparison | Open Question |
| | | E001 | Satellite-radar matching | MCQ |
| **Life Science (14)** | Proteomic Sequencing | L001 | Fragment ion peaks count | Exact Match |
| | | L003 | De novo peptide sequence | Exact Match |
| | Metabolomic Sequencing | L005 | Atom count inference | Exact Match |
| | | L006 | Molecular composition inference | Exact Match |
| | | L007 | Spectrum Matching | MCQ |
| | Protein Structure Analysis | L008 | Protein chain count | Exact Match |
| | | L009 | Small molecule count | Exact Match |
| | | L010 | Specified protein detection | Exact Match |
| | | L019 | Protein structure feature analysis | Open Question |
| | RNA Structure Analysis | L014 | Structural domains identification | MCQ |
| | | L015 | Structural domain count | Exact Match |
| | | L016 | RNA type identification | Exact Match |
| | | L017 | RNA secondary structure inverse folding | Exact Match |
| | | L020 | Structural motifs and positions description | Open Question |
| **Materials Science (11)** | Crystal Structure Analysis | M001 | Atomic composition description | MCQ |
| | | M002 | Crystal group identification | Exact Match |
| | | M003 | Crystal formula determination | Exact Match |
| | | M004 | Elemental valence state prediction | Exact Match |
| | | M006 | Stability estimation | MCQ |
| | Electronic Property Analysis | M008 | Energy band and DOS interpretation | Open Question |
| | | M010 | Band gap classification | Open Question |
| | | M018 | Valence state and electronic orbital analysis | Open Question |
| | XRD Pattern Inference | M020 | Phase identification | Open Question |

| Discipline | Task | Subtask ID | Subtask | Question Type |
|---|---|---|---|---|
| | | M021 | Lattice constant estimation | Open Question |
| | | M022 | Crystal grain size estimation | Open Question |

## C  Tasks Description

**Astronomy**. In the field of astronomy, raw astronomical observations span multiple modalities, including images, time series, spectra, and hyperspectral observations [1, 32, 2]. Specifically, data collected for celestial objects often covers multiple bands and includes rich attribute information. Therefore, inspired by [3], we divide astronomical tasks into three categories based on data modalities, including *Celestial Image Understanding and Comparison*, *Time-series Light Curve Analysis*, and *Spectral Analysis*, each of which is detailed below.

- *Star Perception and Comparison*. The morphology and brightness of celestial objects provide crucial insights into galaxy formation, which is essential for studying the evolutionary history of stars. For instance, in the *transient detection* sub-task, given the pre-transient and post-transient images, as well as the difference between the two, MLLMs are required to determine whether a transient event has occurred during this process.

- *Light Curve Analysis*. Photometric light curves record the brightness of celestial objects over time, and are crucial for estimating the intrinsic properties and predicting the dynamic behavior of stars. For example, in the *Light Curve Classification* sub-task, MLLMs are provided with time-series photometric data and are required to classify the type of celestial object.

- *Spectral Analysis* Spectral data enables the inference of physical properties such as surface temperature and chemical composition. Therefore, we have designed 5 sub-tasks based on different physical properties. For example, in the *Surface Temperature Estimation* task, MLLMs are required to estimate the surface temperature of a target celestial object based on its spectral profile.

**Chemistry**. To systematically evaluate MLLMs performance in the field of chemistry, we organize our benchmark around **four major capability categories**: *Structure Analysis, Molecular Property Prediction, Chemical Reaction Prediction, and Molecular Design and Optimization*. These capabilities reflect fundamental scientific questions central to chemical research and application.

- *Structure Analysis*. Structure Analysis tests a models grasp of symbolic chemistry, such as identifying elemental composition, describing molecules using natural language, and generating IUPAC (International Union of Pure and Applied Chemistry) names (*e.g.*, methane, ethane) [55]. It examines the foundational ability to ensure that the model correctly recognizes the chemical structure and elements of the molecules.

- *Property Prediction*. Molecular Property Prediction aims to assess a models ability to infer chemical and biological properties from molecular structures. It focuses on the properties of chemical molecules that scientists are most interested in, such as drug molecules' ADMET (*i.e.*, absorption, distribution, metabolism, excretion, toxicity) [19], synthetic accessibility, and quantum mechanics energy, etc.

- *Reaction Prediction*. Chemical Reaction Prediction assesses the MLLM's understanding of chemical reactions, including the products of the reaction, reaction centers (the region within a molecule where bond breaking and forming occur), and required reaction conditions (*e.g.*, catalyst).

- *Molecular Design and Optimization*. Molecular Design and Optimization evaluates the MLLMs capacity to generate molecules with desirable chemical properties (*e.g.*, pharmaceutical properties). These tasks focus on identifying novel and diverse molecules with desirable molecular properties (*e.g.*, druglikeness, solubility, binding affinities to the target proteins), which are fundamental tasks in drug and material discovery [18].

- *Drug Screening*. Drug Screening evaluates the MLLMs ability to identify biologically relevant candidate molecules based on protein sequences and molecular structure information [30].

**Earth**. The Earth science domain includes various data modalities. First, variables can be either temporal or static. Meteorological time series are often used to analyze strong convective weather events [85, 21, 74]. Static attributes, such as long-term average temperature maps, are typically used to capture climate patterns. Second, strong correlations often exist between variables. For example, the position of the western Pacific subtropical high (*i.e.*, the 5880 hPa line) is positively correlated with the coverage range of precipitation belts. Third, variables may originate from diverse sources, including ground-based weather radar [62], multi-spectral satellite imagers [61], and various remote sensing instruments. To cater to such diverse scenarios, we establish a comprehensive Earth science task paradigm that focuses on 4 tasks, each comprising multiple sub-tasks.

- *Task1: Circulation Understanding*. In atmospheric physics, large-scale circulation patterns play a central role in modulating regional weather phenomena, such as precipitation anomalies. For example, in the *moisture source attribution* sub-task, MLLMs are required to reason the origin of moisture contributing to the regional rainfall by integrating information from moisture fluxes and wind fields.

- *Task2: Multivariate Understanding*. Complex weather phenomena often arise from the interplay among multiple meteorological variables. For example, in the *precipitation event analysis* sub-task, MLLMs are required to consider precipitation, temperature, wind direction, moisture flux, and geopotential height collectively to infer the underlying causes of rainfall events.

- *Task3: Remote Sensing Perception*. Remote sensing imagery provides multispectral observations of Earth's surface, enabling comprehensive monitoring of environmental and urban conditions. To assess the capability of MLLMs in identifying object categories and localizing their spatial positions from satellite data, we design two sub-tasks using different image modalities. MLLMs are required to first count the number of objects belonging to a specified category, and then identify the spatial location of each instance using the bounding box.

- *Task4: Ground Data Comparison Reasoning*. Ground observational data have been widely used in scenarios such as global climatological analysis and extreme precipitation monitoring [68, 9]. To evaluate the capability of MLLMs in comparing and reasoning meteorological terminology and data distributions, we define 4 tasks based on different variables. For example, in the *temperature sequence comparison task*, MLLMs need to compare annual temperature series from two different time periods and describe differences of statistical characteristics

**Life**. To access the capabilities of MLLMs in the domain of Life Science, we define **three core categories**: *Biomolecular Profiling, Sequence Reasoning, and Structure Interpretation*. These capabilities reflect core scientific practices in the Life Science, where understanding arises from linking molecular measurements, symbolic sequences, and structural representations to functional biological meaning.

- *Biomolecular Profiling*. Biomolecular Profiling involves extracting quantitative and compositional features of biological molecules. These profiling skills are fundamental in high-throughput biological workflows such as mass spectrometry and molecular diagnostics, where precise characterization of biomolecular species is critical for downstream analysis.

- *Sequence Reasoning*. Sequence Reasoning addresses the functional interpretation of biological sequences, including proteins and RNAs. Such reasoning requires models to understand sequence-function relationships, modularity, and constraints that are central to biological information flow and molecular design.

- *Structure Interpretation*. Structure Interpretation targets the analysis of spatial and symbolic representations of biological macromolecules. These tasks emphasize spatial abstraction and multi-view understanding, as biological structures are often represented through hybrid formats that combine 2D, symbolic, and textual information.

**Materials Science**. Materials science presents unique challenges grounded in structural periodicity, quantum-level electronic properties, and empirical characterization signals. To reflect these, we design tasks around **three core capabilities**: *Crystal Structure Analysis, Electronic Property Analysis, and XRD Pattern Inference*.

- *Crystal Structure Analysis*. Crystal Structure Analysis involves symbolic, compositional, and descriptive reasoning over crystalline systems. These tasks reflect how materials scientists represent, classify, and formally describe the structural foundations of solids, where a deep

understanding of symmetry operations, periodicity, and the thermodynamic and structural stability conditions is required.

- *Electronic Property Analysis.* Electronic Property Analysis aims at interpreting the electronic behavior of a material through visual and symbolic representations. These tasks require models to bridge diagrammatic understanding with functional inference, emulating expert workflows in electronic materials analysis and property prediction.

- *XRD Pattern Inference.* XRD Pattern Inference focuses on scientific reasoning grounded in experimental diffraction signals. These tasks require models to connect peak pattern distributions with structural parameters through symbolic and numerical reasoning.

# D  Question Format

For multiple-choice questions, the question is formatted as:

> **Multiple-choice questions format example**
>
> You are an expert in `discipline` and need to solve the following question. The question is a multiple-choice question. Answer with the option letter from the given choices.
>
> {task prompt}
>
> {question}
>
> {options}

For exact match questions, the question is formatted as:

> **Exact match questions format example**
>
> You are an expert in `discipline` and need to solve the following question. The question is an exact match question. Answer the question using a single word or phrase.
>
> {task prompt}
>
> {question}

For open-ended questions, the question is formatted as:

> **Exact match questions format example**
>
> You are an expert in `discipline` and need to solve the following question. The question is an open-ended question. Answer the question using a phrase.
>
> {task prompt}
>
> {question}

Finally, for bounding boxes extraction questions, the question is formatted as:

> **bounding boxes extraction questions format example**
>
> You are an expert in `discipline` and need to solve the following question. The question is an open-ended question. Answer the question using a phrase.
> Each bounding box is represented by four numbers, corresponding to the positions `x_min`, `y_min`, `x_max`, and `y_max`. The coordinate origin is located at the top-left corner of the image, and the bottom-right corner has coordinates (1, 1). Therefore, both `x` and `y` range from 0 to 1. In the final output, two bounding boxes are separated by a semicolon, and all bounding boxes are enclosed in square brackets. Here is an example output: "[x1_min, y1_min, x1_max, y1_max; x2_min, y2_min, x2_max, y2_max]" (no quotation marks).
>
> {task prompt}

```
{question}
```

# E    Responses Evaluation

Given the MLLM's responses, the ground truth answer and the problem, we ask GPT-4o to judge the correctness with the following prompt.

---

**LLM-as-a-Judge prompt**

You are a strict evaluator assessing answer correctness. You must score the model's prediction on a scale from 0 to 10, where 0 represents an entirely incorrect answer and 10 indicates a highly correct answer.

# Input

Question:

```
{question}
```

Ground Truth Answer:

```
{answer}
```

Model Prediction:

```
{prediction}
```

# Evaluation Rules
- The model prediction may contain the reasoning process, you should spot the final answer from it.
- For multiple-choice questions: Assign a higher score if the predicted answer matches the ground truth, either by option letters or content. Include partial credit for answers that are close in content.
- For exact match and open-ended questions:
   * Assign a high score if the prediction matches the answer semantically, considering variations in format.
   * Deduct points for partially correct answers or those with incorrect additional information.
- Ignore minor differences in formatting, capitalization, or spacing since the model may explain in a different way.
- Treat numerical answers as correct if they match within reasonable precision
- For questions requiring units, both value and unit must be correct

# Scoring Guide
Provide a single integer from 0 to 10 to reflect your judgment of the answer's correctness.

# Strict Output format example
4

---

