# A   Subtask Description

## A.1   Astronomy

### A.1.1   Galaxy Morphology Classification (A001)

In this subtask, we address the problem of galaxy morphology classification from single-band optical imagery. The goal is to determine the structural category of a galaxy—such as merging, spiral, or edge-on—based on its visual appearance. This task poses significant challenges due to the complex and diverse nature of galactic structures, along with variations in scale, orientation, and observational noise. We adopt a multi-class classification formulation with four expert-defined morphological labels. Accurate recognition requires capturing both global shape and fine-grained features, which motivates the need for models with strong spatial sensitivity and robustness to astrophysical variations.

---

**Galaxy Morphology Classification (A001)**

**Images**:

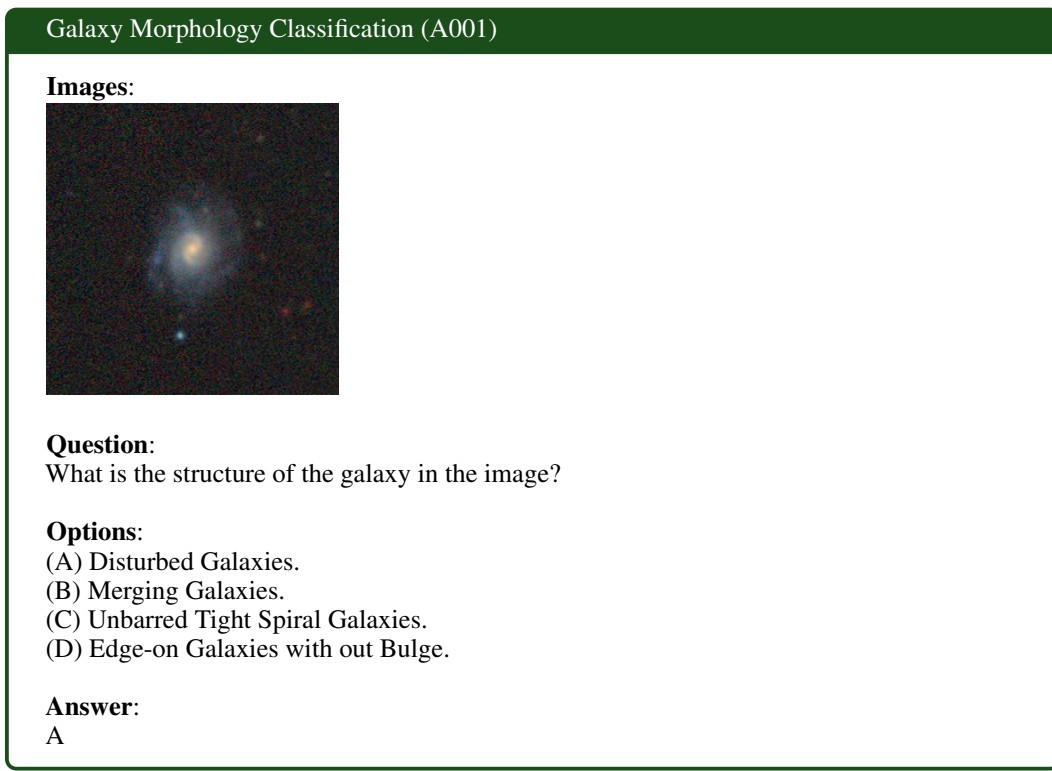

**Question**:
What is the structure of the galaxy in the image?

**Options**:
(A) Disturbed Galaxies.
(B) Merging Galaxies.
(C) Unbarred Tight Spiral Galaxies.
(D) Edge-on Galaxies with out Bulge.

**Answer**:
A

---

### A.1.2   Light Curve Classification

This subtask focuses on the classification of astronomical light curves across multiple photometric bands. Each instance represents the temporal flux variation of a celestial source, visualized as band-wise curves. The objective is to identify the physical origin of the variability—ranging from active galactic nuclei (AGN) to microlensing or tidal disruption events. The problem is inherently noisy and sparse, with irregular sampling and band-dependent features. We pose this as a multi-class classification task, where models must learn to align temporal dynamics with astrophysical signatures. This setting tests the model's capacity to extract semantic patterns from structured time-series representations under observational constraints.

---

**Light Curve Classification (A003)**

**Images**:

---

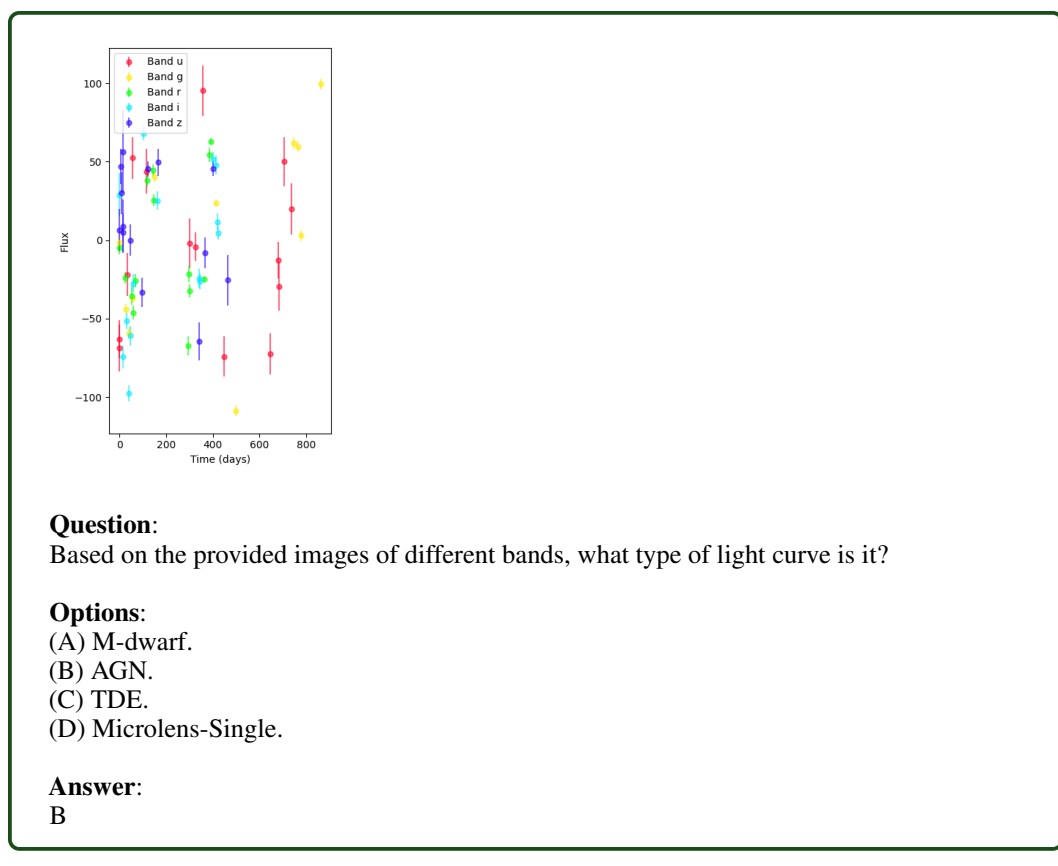

**Question**:
Based on the provided images of different bands, what type of light curve is it?

**Options**:
(A) M-dwarf.
(B) AGN.
(C) TDE.
(D) Microlens-Single.

**Answer**:
B

### A.1.3 Surface Temperature Estimation

This subtask addresses the estimation of stellar surface temperature from spectral observations. Each input is a flux-calibrated spectrum plotted over wavelength, with red markers denoting the Line Spread Function sigma. The task is to predict the effective temperature (Teff) of the star, formulated as a regression problem. Accurate estimation relies on modeling subtle variations in spectral line profiles and continuum shapes, which encode temperature-sensitive features. The challenge lies in handling high-dimensional, noisy spectral data while preserving physically meaningful cues. This task benchmarks a model's ability to extract quantitative astrophysical parameters from structured scientific measurements under realistic observational conditions.

**Surface Temperature Estimation (A006)**

**Images**:

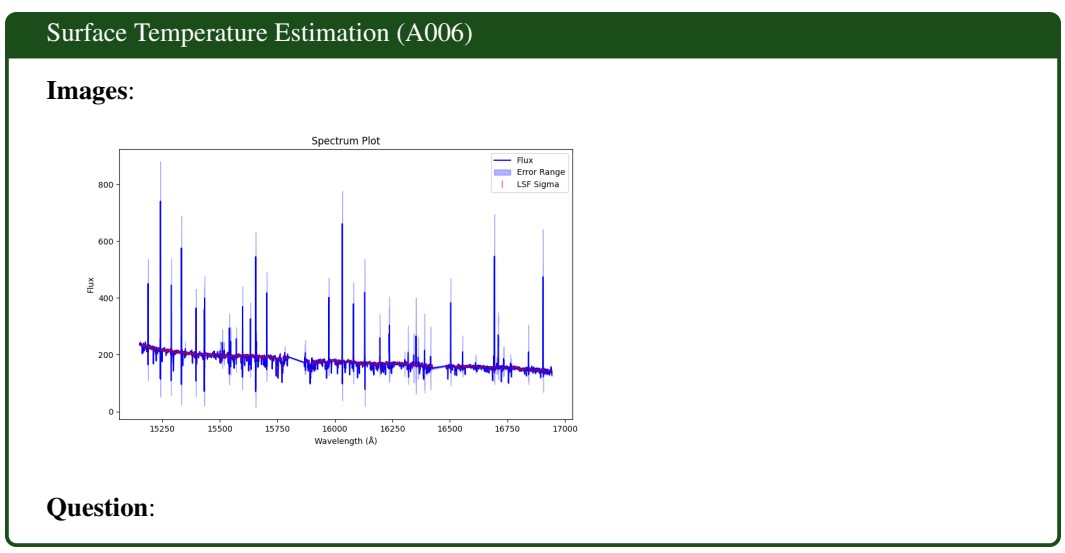

**Question**:

The image provided is a spectrum of a star. The X-axis represents wavelength in Angstroms (\u00c5), and the Y-axis represents flux. The red spots indicate the Line Spread Function Sigma. What is the estimated effective temperature(Teff, in Kelvin) of the star based on the spectrum? You only need to respond with a float value.

**Answer**:
5402.044921875

### A.1.4 Gravitational Constant Estimation

This subtask focuses on estimating the surface gravity (log g) of a star from its observed spectrum. The input is a one-dimensional flux curve over wavelength, with red markers indicating the Line Spread Function sigma. Surface gravity estimation is formulated as a regression task, requiring precise inference from subtle line-broadening features and continuum patterns. The complexity arises from the interplay of noise, spectral resolution, and astrophysical diversity. Robust models must learn to extract informative cues while generalizing across varying stellar types. This task provides a controlled benchmark for evaluating a model's ability to recover physical parameters from high-dimensional spectral data.

---

**Gravitational Constant Estimation (A007)**

**Images**:

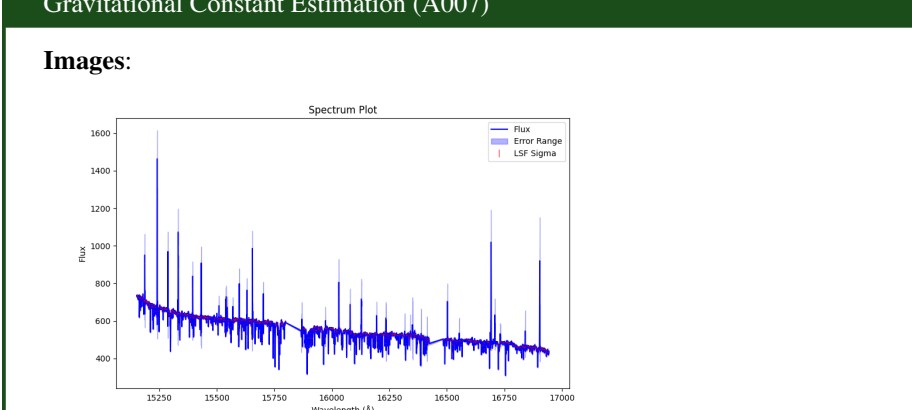

**Question**:
The image provided is a spectrum of a star. The X-axis represents wavelength in Angstroms (\u00c5), and the Y-axis represents flux. The red spots indicate the Line Spread Function Sigma. What is the estimated surface gravity (log g) of the star based on the spectrum? You only need to respond with a float value.

**Answer**:
4.1094794273376465

---

### A.1.5 Metallicity Estimation

This subtask targets the estimation of stellar metallicity ([M/H]) from spectroscopic data. Each input is a high-resolution stellar spectrum, where the flux is plotted against wavelength, and red markers indicate the Line Spread Function sigma. The goal is to regress the overall metal abundance, a key indicator of stellar composition and evolution. Metallicity signatures manifest as fine absorption features, often shallow and blended, making the task sensitive to spectral resolution and noise. We cast this as a regression problem that requires models to extract and integrate subtle chemical patterns. This benchmark emphasizes precision in learning astrophysical properties from dense observational data.

### A.1.6 Alpha-element Abundance Estimation

This subtask targets the estimation of stellar metallicity ([M/H]) from spectroscopic data. Each input
is a high-resolution stellar spectrum, where the flux is plotted against wavelength and red markers
indicate the Line Spread Function sigma. The goal is to regress the overall metal abundance, a key
indicator of stellar composition and evolution. Metallicity signatures manifest as fine absorption
features, often shallow and blended, making the task sensitive to spectral resolution and noise. We
cast this as a regression problem that requires models to extract and integrate subtle chemical patterns.
This benchmark emphasizes precision in learning astrophysical properties from dense observational
data.

spectrum? You only need to respond with a float value.

**Answer**:
0.12627530097961426

### A.1.7 Radial Velocity Estimation

This subtask aims to estimate the radial velocity of a star from its observed spectrum. The input is a flux-versus-wavelength curve, with red markers indicating the Line Spread Function sigma. Radial velocity manifests as Doppler-induced shifts in spectral lines, often subtle and entangled with instrumental effects and stellar variability. We cast this as a regression problem that requires precise modeling of line positions under varying noise and resolution. Success in this task demands both global pattern recognition and fine-grained alignment sensitivity. This benchmark probes a model's capability to infer stellar motion from spectral displacement, under realistic observational constraints.

---

**Radial Velocity Estimation (A010)**

**Images**:

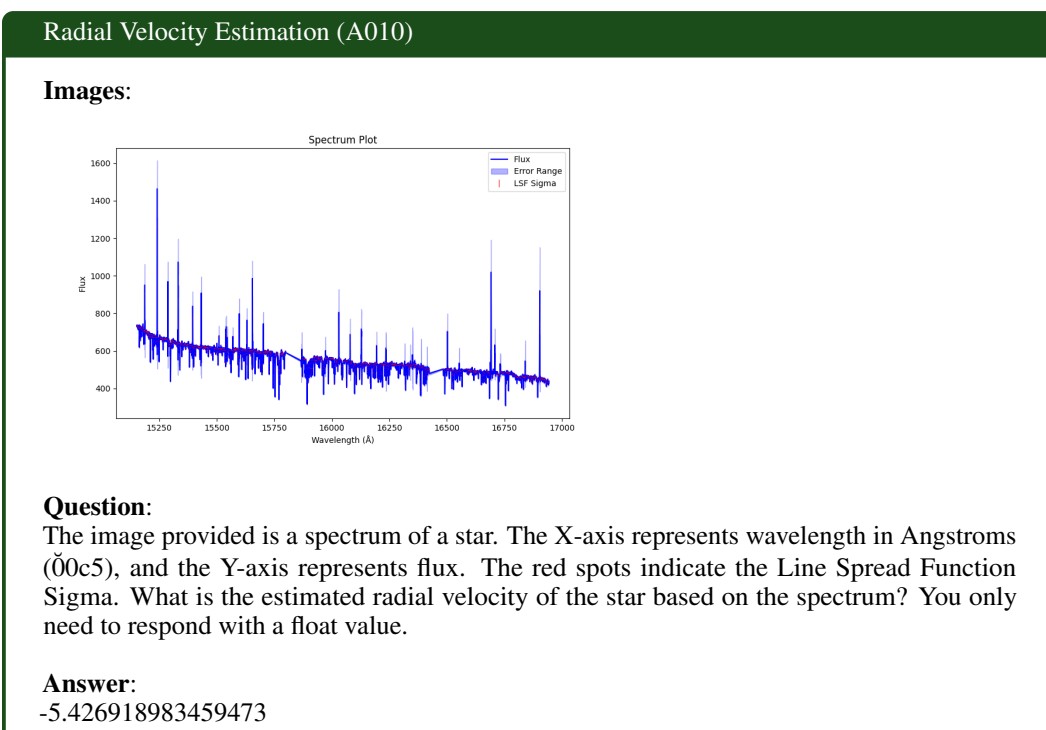

**Question**:
The image provided is a spectrum of a star. The X-axis represents wavelength in Angstroms (Ŏ0c5), and the Y-axis represents flux. The red spots indicate the Line Spread Function Sigma. What is the estimated radial velocity of the star based on the spectrum? You only need to respond with a float value.

**Answer**:
-5.426918983459473

---

### A.1.8 Transient Detection

This subtask targets the detection of astrophysical transients through image differencing. Given a triplet of images—current observation, historical reference, and their pixel-wise difference—the objective is to determine the presence of a transient event, such as a supernova or variable star. The challenge arises from subtle photometric variations and observational artifacts that may obscure true events or generate false positives. We formulate the task as a binary classification problem, demanding models that can effectively learn discriminative temporal cues while suppressing noise and instrumental distortions. This setting provides a controlled yet realistic benchmark for evaluating temporal sensitivity in astronomical image analysis.

---

**Transient Detection (A011)**

**Images**:

---

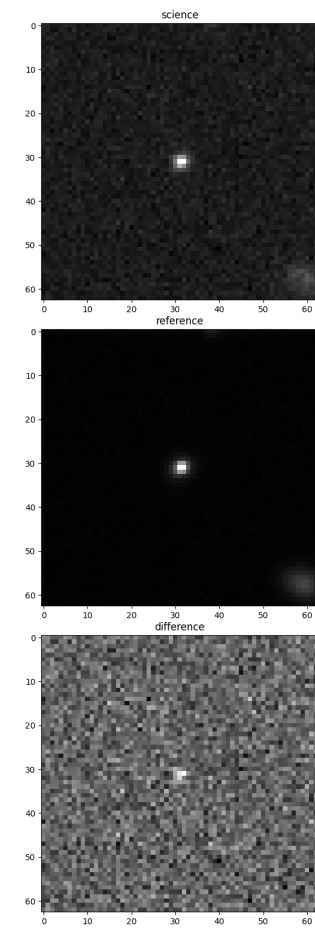

**Question**:
The first image is the latest reference image, the second image is the historical reference image, and the third image shows the difference between the two. Please determine whether a transient event has occurred based on these three images.

**Options**:
(A) Yes, a transient event has occurred.
(B) No, a transient event has not occurred.
(C) I don't know.
**Answer**:
B

## A.2 Chemistry

### A.2.1 Elemental Composition Recognition (C001)

The Elemental Composition Recognition task focuses on identifying and quantifying the atomic constituents in a given molecular structure image. Given a 2D chemical diagram, the task requires the model to enumerate all unique elements present and tally the number of atoms for each element. This subtask demands a comprehensive understanding of chemical notation and spatial relationships, challenging the system's capability to parse complex visual and symbolic information. Precision in recognizing overlapping or ambiguous atoms is vital. This task serves as a fundamental benchmark for evaluating visuo-semantic reasoning in machine perception frameworks and sets a strong baseline for molecular understanding.

**Images**:

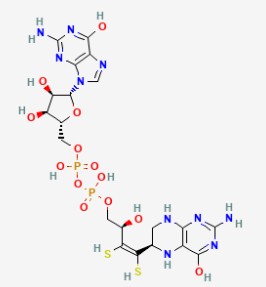

**Question**:
From the molecular structure <image>, list all the unique elements present in the compound and the number of atoms of each.

**Answer**:
"C": 20, "H": 28, "N": 10, "O": 13, "P": 2, "S": 2

### A.2.2 IUPAC Name Recognition (C003)

The IUPAC Name Recognition subtask aims to bridge molecular structure identification with systematic chemical nomenclature. Given a molecular diagram, the model is required to generate its precise IUPAC name, demonstrating both structural understanding and proficiency in chemical linguistics. This task rigorously tests the model's ability to parse complex visual information and translate it into standardized nomenclature, reflecting the fundamentals of computational chemistry and pattern recognition. In essence, this subtask challenges the model to establish a robust mapping from visually represented molecules to their unique textual identifiers, a critical step toward comprehensive chemical informatics.

**Images**:

**Question**:
From the molecular structure <image>, provide the molecular IUPAC name.

**Answer**:
N-[(5S,5aS,8aR,9R)-9-(4-hydroxy-3,5-dimethoxyphenyl)-8-oxo-5a,6,8a,9-tetrahydro-5H-[2]benzofuro[5,6-f][1,3]benzodioxol-5-yl]-2-hydroxybenzamide

### A.2.3 Molecular Description Generation (C004)

The subtask centers on the automatic generation of natural language descriptions for molecular structures from their graphical representations. Given an image depicting a chemical compound,

the model is expected to output a coherent and accurate textual summary detailing the molecule's key structural features, constituent components, and functional groups. This task challenges the model's understanding of visual molecular notation and its chemical semantics, thereby facilitating applications ranging from automated database curation to assistive chemical education. By bridging vision and language in the context of cheminformatics, this subtask provides a rigorous benchmark for evaluating multimodal reasoning and generation capabilities.

---

**Molecular Description Generation (C004)**

**Images**:

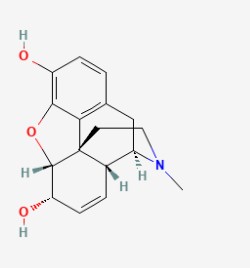

**Question**:
Generate a natural language description for the given molecule <image>.

**Answer**:
The molecule is a branched amino tetrasaccharide comprising N-acetyl-beta-D-glucosamine at the reducing end with a N-acetyl-alpha-D-galactosaminyl-(1->3)-[alpha-L-fucosyl-(1->2)]-beta-D-galactosyl moiety attached at the 4-position. It has a role as an epitope.

---

### A.2.4   Lipinski Drug-likeness Estimation (C005)

In this subtask, we evaluate the ability of models to estimate key drug-likeness properties according to Lipinski's Rule of Five, a fundamental heuristic in medicinal chemistry for assessing oral bioavailability. Given a molecular structure image, the model is required to compute five physicochemical descriptors: molecular weight, XLogP (an estimate of octanol-water partition coefficient), hydrogen bond donor count, hydrogen bond acceptor count, and rotatable bond count. The outputs are standardized as a JSON object for consistency. This task demands not only a robust understanding of chemical structures but also precise quantitative reasoning, analogous in rigor to structure-property prediction challenges.

---

**Lipinski Drug-likeness Estimation (C005)**

**Images**:

**Question**:
From the molecular structure <image>, calculate the five Lipinski rule indicators with values rounded to one decimal place: molecular weight, LogP, number of hydrogen bond donors,

number of hydrogen bond acceptors, and number of rotatable bonds. Please output as a JSON dict with these exact keys (no units):

```
{
"Molecular Weight": ,
"XLogP": ,
"Hydrogen Bond Donor Count": ,
"Hydrogen Bond Acceptor Count": ,
"Rotatable Bond Count": ,
}
```

**Answer**:

```
{
"Molecular Weight": 285.3,
"XLogP": 0.8,
"Hydrogen Bond Donor Count": 2.0,
"Hydrogen Bond Acceptor Count": 4.0,
"Rotatable Bond Count": 0.0
}
```

### A.2.5 Topological Polar Surface Area Calculation (C007)

In this subtask, we focus on the quantitative assessment of a compound's polarity via its Topological Polar Surface Area (TPSA). Given a molecular structure in image format, participants are required to accurately deduce and compute the TPSA, a widely adopted descriptor for predicting drug absorption and permeability. The answer, precise to one decimal place, reflects the sum of surface contributions from polar atoms, and is to be reported without units. This task demands not only rigorous chemical understanding but also proficiency in molecule-feature interpretation, mirroring real-world scenarios in cheminformatics and drug discovery pipelines.

---

**Topological Polar Surface Area Calculation (C007)**

**Images**:

**Question**:
Calculate the Topological Polar Surface Area (TPSA) of the compound from the molecular structure <image>. Report the value in square angstroms (Ŏ0c5Ŏ0b2), rounded to one decimal place. Please omit the unit (u00c5 u00b2) in your answer.

**Answer**:
38.3

---

### A.2.6 Absorption Property Prediction (C023)

In this subtask, we investigate the prediction of key absorption-related properties directly from molecular structure images. The task requires estimating multiple pharmacokinetic attributes, including lipophilicity (logP), aqueous solubility, Caco-2 permeability, and hydration free energy, as

well as binary classifications on human intestinal absorption (HIA), bioavailability, and P-glycoprotein (Pgp) inhibition. Given a compound's structure, the model is prompted to output a precise JSON dictionary with these properties. This design emphasizes exact-match evaluation, reflecting practical drug discovery settings where comprehensive absorption profiling from molecular information is essential for candidate selection and prioritization.

---

**Absorption Property Prediction (C023)**

**Images**:

**Question**:

This is an exact-match question. Based on the molecular structure shown in <image>, please predict the following absorption-related properties for the molecule. For each property, provide a value or a 'yes' / 'no' answer where applicable. The properties are as follows: Lipophilicity, which refers to the Logarithm of the partition coefficient (logP); Solubility, which is the aqueous solubility of the molecule (mg/L); Hydration Free Energy, which is the free energy of hydration (kcal/mol); Caco-2 Permeability, which is the effective permeability of the molecule across Caco-2 cells (cm/s); Is HIA Activity, which indicates whether the molecule shows high human intestinal absorption (yes/no); Is Bioavailable, which indicates whether the molecule is bioavailable (yes/no); and Is Pgp Inhibitor, which indicates whether the molecule inhibits P-glycoprotein (yes/no). Please return your predictions in a JSON dictionary using exactly the following keys. Do not include units or additional descriptions in your output:

```
{
"lipophilicity": ,
"solubility": ,
"caco2_permeability": ,
"is_hia_activity": ,
"is_bioavailable": ,
"is_pgp_inhibitor":
}
```

**Answer**:

```
{
"lipophilicity":1.08,
"solubility":-4.6202577876,
"caco2_permeability":-4.460681,
"is_bioavailable":"yes",
"is_pgp\_inhibitor":"no",
}
```

---

### A.2.7 Distribution Property Prediction (C024)

In this subtask, we focus on the prediction of pharmacokinetic distribution properties for small molecules by analyzing their chemical structures. Specifically, the challenge requires the model to infer whether a given molecule can penetrate the blood-brain barrier (is_bbb), as well as to estimate its plasma protein binding rate (ppbr, in percent) and volume of distribution at steady state (vdss, in

L/kg). The model must accurately extract salient molecular features from visualized structures and reason about their impact on distribution metrics, returning a tightly constrained JSON output for downstream evaluation. This design emphasizes precise, structure-informed reasoning and format adherence.

---

**Distribution Property Prediction (C024)**

**Images**:

**Question**:

This is an exact-match question. Based on the molecular structure shown in <image>, please predict the following distribution-related properties for the molecule. For each property, provide a value or a 'yes' / 'no' answer where applicable. The properties are as follows: Is BBB, which indicates whether the molecule can cross the Blood-Brain Barrier (yes/no); PPBR, which refers to the Plasma Protein Binding Rate (unit: %); and VDss, which is the Volume of Distribution at steady state (unit: L/kg). Please return your predictions in a JSON dictionary using exactly the following keys. Do not include units or additional descriptions in your output:

```
{
"is_bbb": ,
"ppbr": ,
"vdss": ,
}
```

**Answer**:

```
{
"is_bbb":"no",
"ppbr":99.6,
"vdss":1,
}
```

---

### A.2.8 Metabolism Property Prediction (C025)

The Metabolism Property Prediction subtask focuses on the systematic evaluation of xenobiotic biotransformation potential by analyzing molecular structures for their interaction with cytochrome P450 isoforms. Specifically, the task requires the model to assess a given molecule's inhibitory effects on CYP2C19, CYP2D6, CYP3A4, CYP1A2, and CYP2C9, as well as to determine substrate specificity for CYP2C9, CYP2D6, and CYP3A4. Formulated as an exact-match classification, the subtask demands precision in capturing molecular features governing P450 metabolism, and supports downstream drug design by enabling robust prediction of drug-drug interaction risks and metabolic liabilities.

---

**Metabolism Property Prediction (C025)**

**Images**:

---

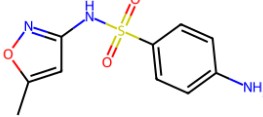

**Question**:
This is an exact-match question. Based on the molecular structure shown in <image>, please predict the following metabolism-related properties for the molecule. For each property, provide a 'yes' or 'no' answer. The properties are as follows: whether the molecule inhibits CYP P450 enzymes including 2C19, 2D6, 3A4, 1A2, and 2C9; and whether it is a substrate for CYP2C9, CYP2D6, and CYP3A4. Please return your predictions in a JSON dictionary using exactly the following keys. Do not include any additional descriptions in your output:

```
{
"CYP2C19_inhibition": ,
"CYP2D6_inhibition": ,
"CYP3A4_inhibition": ,
"CYP1A2_inhibition": ,
"CYP2C9_inhibition": ,
"CYP2C9_substrate": ,
"CYP2D6_substrate": ,
"CYP3A4_substrate": ,
}
```
**Answer**:
```
{
"CYP2C19_inhibition":"no",
"CYP2D6_inhibition":"no",
"CYP3A4_inhibition":"no",
"CYP1A2_inhibition":"no",
"CYP2C9_inhibition":"no",
"CYP2C9_substrate":"yes",
"CYP2D6_substrate":"no",
"CYP3A4_substrate":"yes"
}
```

### A.2.9   Excretion Property Prediction (C026)

In this subtask, the model is presented with a molecular structure image and tasked to predict key excretion-related pharmacokinetic properties, specifically half-life and clearance. The model must analyze the given molecular structure and infer the half-life (in hours) and the intrinsic clearance rate (in mL/min/kg), returning the results as precise numerical values in a standardized JSON format. This requires the model to establish a strong correspondence between molecular structural features and ADME property prediction, ensuring both robustness and reliability in the estimation of pharmacokinetic profiles from purely visual molecular information.

Excretion Property Prediction (C026)

**Images**:

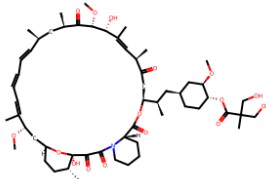

**Question**:
This is an exact-match question. Based on the molecular structure shown in <image>, please predict the following excretion-related properties for the molecule. For each property, provide a numerical value. The properties are as follows: Half-life, which is the half-life duration of the drug (unit: hours); and Clearance, which refers to the intrinsic clearance rate of the drug (unit: mL/min/kg). Please return your predictions in a JSON dictionary using exactly the following keys. Do not include units or additional descriptions in your output:

```
{
"half_life": ,
"clearance": ,
}
```

**Answer**:

```
{
"half_life":18,
"clearance":150,
}
```

### A.2.10 Toxicity Property Prediction (C027)

In this subtask, we address the challenging problem of toxicity property prediction for chemical compounds, a key concern in drug discovery and environmental safety. Given the molecular structure of a compound, the objective is to systematically assess its potential toxic effects across diverse biological dimensions, such as cellular processes, organ specificity, metabolic interactions, and environmental risks. Participants are required to analyze each molecule and select all relevant toxicity aspects from a predefined set of possible outcomes. This formulation enables a comprehensive evaluation of toxicity profiles, facilitating downstream applications and risk assessment in both pharmaceutical and ecological contexts.

**Toxicity Property Prediction (C027)**

**Images**:

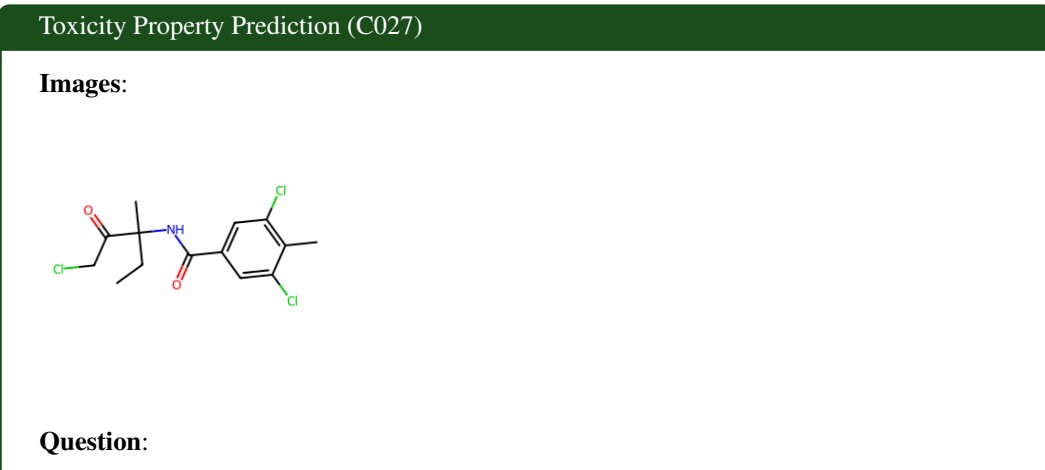

**Question**:

Please analyze the toxicity of this molecule. Which of the following aspects might be affected? (Multiple selections possible)

**Options**:
(A) Potential impact on cell proliferation, apoptosis, and cell cycle.
(B) Potential induction of oxidative stress, impact on metabolism, or mitochondrial function.
(C) Potential effect on enzyme activity, receptor activation, or inhibition.
(D) Potential impact on gene expression, transcription factor activation, and other gene-level regulation.
(E) Potential influence on immune responses, cytokine release, etc.
(F) Potential toxicity to specific organs (e.g., liver, heart, kidney).
(G) Potential impact on development or reproduction processes.
(H) Potential carcinogenicity or environmental toxicity to aquatic life, plants, soil, etc.
(I) Other (please specify).

**Answer**:
C, F

### A.2.11 QM9 Quantum Chemical Property Prediction (C028)

In this subtask, we assess a model's ability to predict quantum chemical properties for small organic molecules from QM9, given only their 2D structural images. The task is challenging, as it requires translating visual molecular representations into accurate estimations of key quantum mechanical quantities, such as dipole moment, HOMO-LUMO energies, and atomization energies. Participants are required to output predictions for sixteen specified properties in a strict JSON format. This subtask evaluates both the holistic and fine-grained chemical understanding of deep models, bridging vision and quantum chemistry, and highlights the capacity for end-to-end property inference from purely visual cues.

QM9 Quantum Chemical Property Prediction (C028)

**Images**:

**Question**:
This is an exact-match question. Based on the molecular structure shown in <image>, please predict the following quantum mechanical properties for the molecule. For each property, provide a numerical value. The properties are as follows: mu, which is the dipole moment (unit: Debye); alpha, the isotropic polarizability (unit: Bohr$^3$); homo, the energy of the highest occupied molecular orbital (unit: Hartree); lumo, the energy of the lowest unoccupied molecular orbital (unit: Hartree); gap, the energy gap between HOMO and LUMO (unit: Hartree); r2, the electronic spatial extent (unit: Bohr$^2$); zpve, the zero point vibrational energy (unit: Hartree); u0, the internal energy at 0K (unit: Hartree); u298, the internal energy at 298.15K (unit: Hartree); h298, the enthalpy at 298.15K (unit: Hartree); g298, the free energy at 298.15K (unit: Hartree); cv, the heat capacity at 298.15K (unit: cal/(mol*K)); u0_atom, the atomization energy at 0K (unit: kcal/mol); u298_atom, the atomization energy at 298.15K (unit: kcal/mol); h298_atom, the atomization enthalpy at 298.15K (unit: kcal/mol); and g298_atom, the atomization free energy at 298.15K (unit: kcal/mol). Please return your

predictions in a JSON dictionary using exactly the following keys. Do not include units or additional descriptions in your output:

```
{
"mu": ,
"alpha": ,
"homo": ,
"lumo": ,
"gap": ,
"r2": ,
"zpve": ,
"u0": ,
"u298": ,
"h298": ,
"g298": ,
"cv": ,
"u0_atom": ,
"u298_atom": ,
"h298_atom": ,
"g298_atom": ,
}
```

**Answer**:

```
{
"mu": 3.9822,
"alpha": 67.58,
"homo": -0.2636,
"lumo": 0.0037,
"gap": 0.2673,
"r2": 973.0186,
"zpve": 0.136173,
"u0": -458.980805,
"u298": -458.973242,
"h298": -458.972298,
"g298": -459.012433,
"cv": 30.273,
"u0_atom": -1696.9424682681,
"u298_atom": -1707.3057794031,
"h298_atom": -1716.788695411,
"g298_atom": -1580.3933391711,
}
```

### A.2.12   Protein-ligand Binding Affinity Prediction (C029)

The Protein-ligand Binding Affinity Prediction subtask evaluates the capability of computational models to quantitatively infer the binding affinity between a given small molecule and a target protein, based on molecular structure images and protein FASTA sequences. Accurate prediction of binding constants (Kd, Ki, or IC50) is critical in rational drug design and virtual screening. This task presents both the molecular structure and the amino acid sequence, requiring the model to comprehensively analyze intermolecular interactions and sequence-specific features. The outcome is measured by the precision of predicted affinity values, reflecting the model's understanding of protein-ligand recognition mechanisms.

## Protein-ligand Binding Affinity Prediction (C029)

**Images**:

**Question**:
This question is based on the molecular structure <image> and the protein FASTA sequence of 1a30(>1A30_1|Chains A, B|HIV-1 PROTEASE|Human immunodeficiency virus 1 (11676) PQITLWKRPLVTIKIGGQLKEALLDTGADDTVIEEMSLPGRWKPKMIGGIG-GFIKVRQYDQIIIEICGHKAIGTVLVGPTPVNIIGRNLLTQIGCTLNF >1A30_2|Chain C|TRIPEPTIDE GLU-ASP-LEU| EDL
Please predict the binding affinity (Kd/Ki/IC50), which represents the molecule's binding affinity to the target protein (format: Kd/Ki/IC50 = <value> <unit>).

**Answer**:
Ki=50uM

### A.2.13 Product SMILES Prediction (C009)

Product SMILES Prediction is designed to evaluate a model's chemical reasoning in reaction product prediction. Each instance presents a reactant structure as an image, deliberately omitting all auxiliary components such as catalysts and solvents, thereby restricting the available information to the core reactant(s). The model is tasked to generate the canonical SMILES string corresponding to the most probable product that arises from the depicted structure. This setting challenges models to capture fundamental mechanistic knowledge and to generalize beyond rote memorization, paralleling the rigorous demand for abstraction prevalent in chemical synthesis planning.

## Product SMILES Prediction (C009)

**Images**:

**Question**:
Generate the SMILES representation of the most likely product based on the given reactant structure <image>. The image only contains reactants, excluding catalysts, reagents, or solvents.

**Answer**:
CCOC(=O)C1=CC2=C(C=CC=C2O1)N3CCN(CC3)CC4=CC=CC=C4

### A.2.14 Reaction Classification (C010)

The Reaction Classification subtask aims to evaluate the model's ability to accurately identify and categorize organic reactions based on presented molecular structures. Each instance provides a reactant's chemical diagram and a multiple-choice list of plausible reaction types, often involving subtle distinctions such as variations in leaving groups or coupling partners. The task presents significant challenges, requiring both recognition of functional groups and a nuanced understanding of common reaction mechanisms. Correctly classifying these transformations is essential for predicting chemical reactivity and synthetic planning, thereby serving as a robust benchmark for the model's chemical reasoning and expert-level pattern recognition capabilities.

---

**Reaction Classification (C010)**

**Images**:

**Question**:
Based on the given reactant structure <image>, classify the most likely type of reaction.

**Options**:
(A) Iodo Buchwald-Hartwig amination
(B) Bromo N-arylation
(C) Chloro Buchwald-Hartwig amination
(D) Bromo Buchwald-Hartwig amination
(E) Chloro N-arylation

**Answer**:
C

---

### A.2.15 Reactant Molecular Recognition (C012)

The Reactant Molecular Recognition subtask targets a fundamental challenge in reaction informatics: determining molecular identities from structural representations. Given an image capturing the reactant structures in a chemical reaction, the objective is to accurately extract and enumerate the molecular formula for each reactant, preserving their left-to-right order as depicted. This subtask emphasizes both precise chemical perception and the translation of visual molecular information into standard chemical formulas, bridging the gap between image-based molecular recognition and structured chemical knowledge. The task demands high accuracy and robustness against diverse molecular drawings, providing rigorous benchmarks for visual reasoning models in the chemistry domain.

---

**Reactant Molecular Recognition (C012)**

**Images**:

**Question**:
From the reactants structure <image>, identify the molecular formula of each reactant involved in the reaction, in left-to-right order.

**Answer**:
C5H2Cl2IN, C8H10N2O

### A.2.16 Reaction Condition and Catalyst Prediction (C013)

In this subtask, the objective is to predict the required catalysts, solvents, or reagents for a given organic transformation, based solely on the reactant structures. Participants are presented with a visual depiction of the reactants alongside a set of multiple-choice options, each consisting of chemical structures representing possible conditions. The task may have one or more correct answers, mirroring the ambiguity and diversity frequently encountered in chemical synthesis. This problem formulation emphasizes both chemical knowledge and reasoning, challenging models to bridge the gap between structural input and operational context, much like practical decision-making in synthetic chemistry.

**Reaction Condition and Catalyst Prediction (C013)**

**Images**:

**Question**:
This is a multiple-choice question with one or more correct answers. Based on the reactant structures <image>, select all catalysts, solvents, or reagents required for the reaction.

**Options**:

(A) solvent    (B) catalyst    (C) reagent

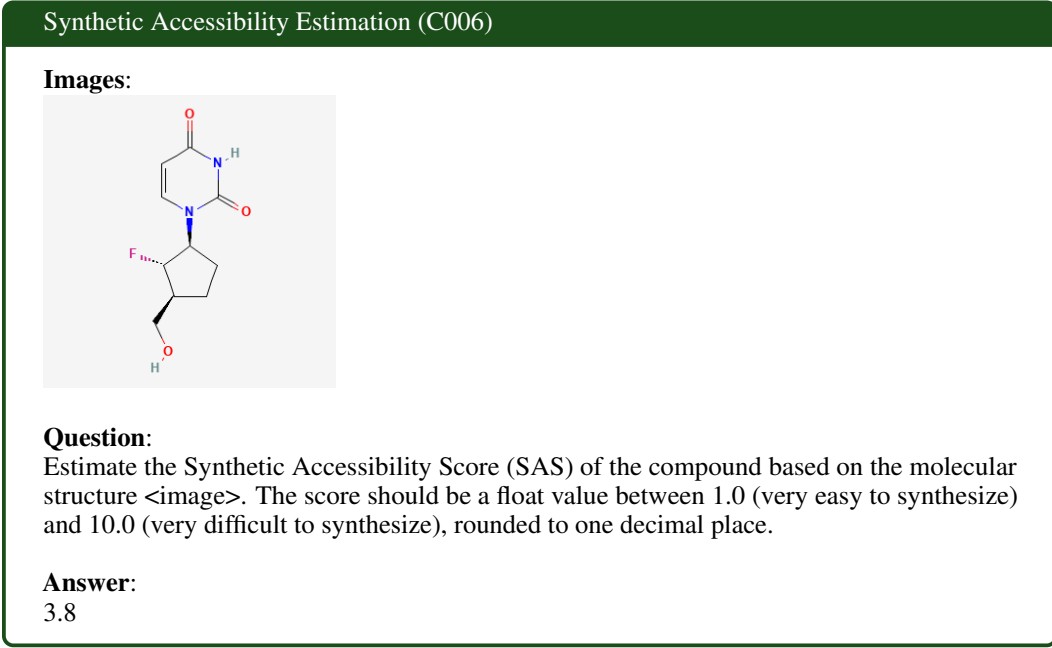

**Answer**:
C, E, F, G

### A.2.17 Synthetic Accessibility Estimation (C006)

The Synthetic Accessibility Estimation subtask aims to evaluate a model's capability in assessing the practical feasibility of chemical synthesis from molecular structure images. Each instance presents a single compound as a 2D structural diagram, prompting the model to predict a Synthetic Accessibility Score (SAS) within the standardized 1.0–10.0 range, where lower scores denote higher synthetic tractability. This task reflects the essential integration of cheminformatics perception and domain reasoning, requiring the system to accurately distill both structural complexity and functional group context. Performance in this subtask is critical for downstream applications in de novo molecular design and automated retrosynthetic planning.

---

**Synthetic Accessibility Estimation (C006)**

**Images**:

**Question**:
Estimate the Synthetic Accessibility Score (SAS) of the compound based on the molecular structure <image>. The score should be a float value between 1.0 (very easy to synthesize) and 10.0 (very difficult to synthesize), rounded to one decimal place.

**Answer**:
3.8

---

### A.2.18 Molecular Property Optimization (C018)

The Molecular Property Optimization (C018) subtask is designed to evaluate models' ability to discern and select molecular structures exhibiting superior performance in annotated chemical properties.

In each instance, participants are presented with a set of diverse molecules, each paired with a specific property value or descriptor. The task requires selecting all molecules that demonstrate optimal or above-threshold performance for the given properties. This setting examines not only the recognition of molecular structures but also a nuanced understanding of structure-property relationships. Ultimately, the subtask serves as a rigorous benchmark for algorithmic reasoning in molecular optimization scenarios.

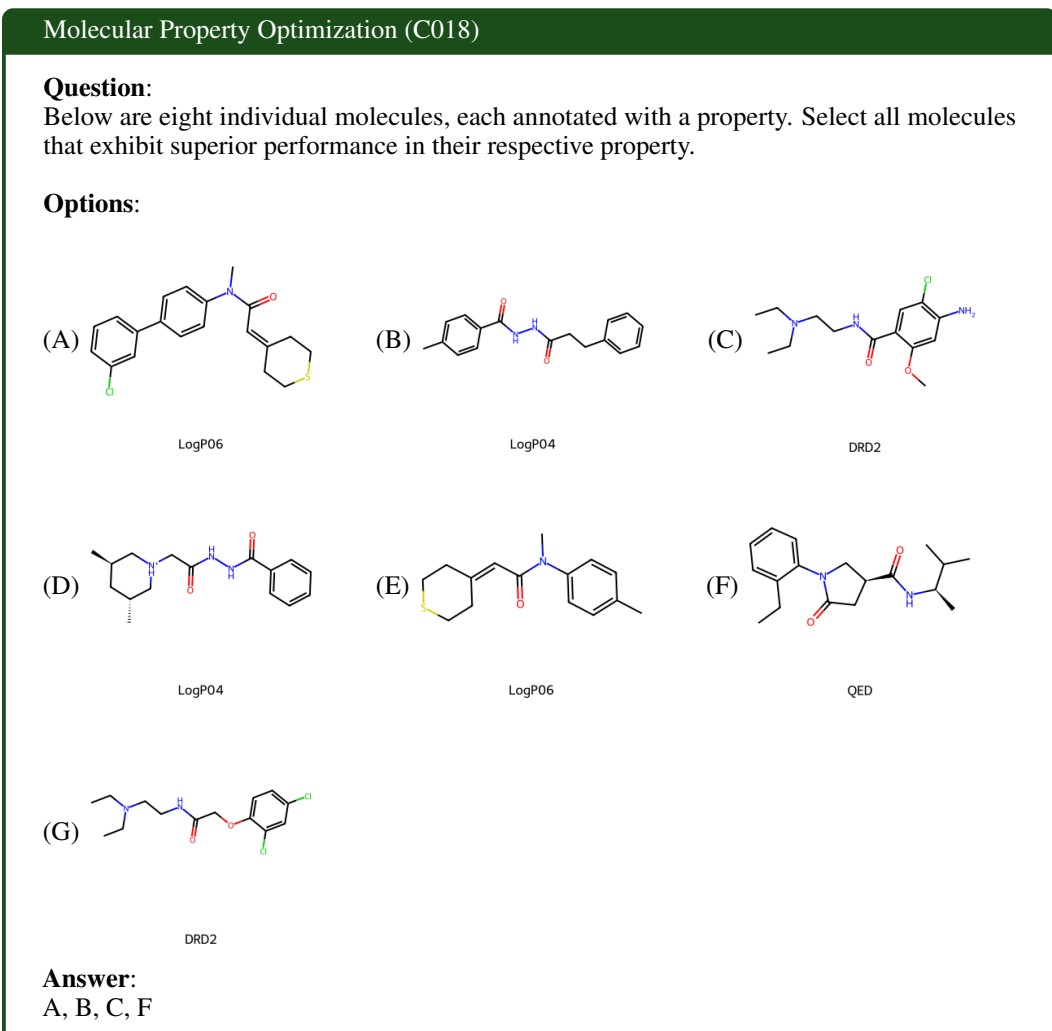

**Molecular Property Optimization (C018)**

**Question**:
Below are eight individual molecules, each annotated with a property. Select all molecules that exhibit superior performance in their respective property.

**Options**:

(A) LogP06

(B) LogP04

(C) DRD2

(D) LogP04

(E) LogP06

(F) QED

(G) DRD2

**Answer**:
A, B, C, F

## A.2.19   Virtual Screening (C030)

Virtual screening is a critical subtask in computational drug discovery, aiming to efficiently identify active compounds with high binding affinity toward a specific biological target. In our benchmark, we present a series of molecular candidates alongside structural representations of the target protein. The task requires the model to discern, among a provided set of molecules, those capable of binding to the target. This subtask evaluates the model's ability to reason about 3D molecular interactions, recognize chemical compatibilities, and generalize knowledge across diverse molecular structures, forming a key foundation for subsequent lead optimization and further experimental validation.

**Virtual Screening (C030)**

**Images**:

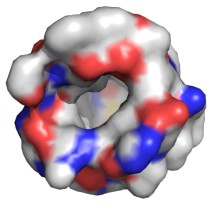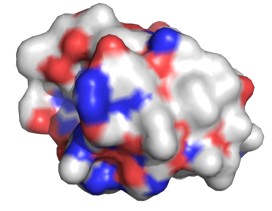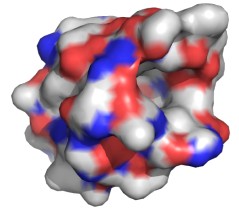

**Question**:
This is a multiple-choice question with one or more correct answers. Based on the target structure <image><image><image>, select all molecules that can bind to it.

**Options**:

(A) 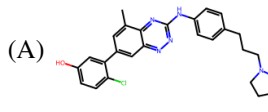

(B) 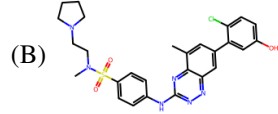

(C) 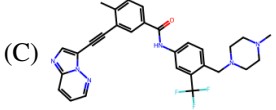

(D) 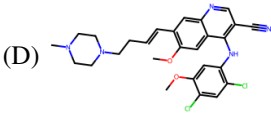

(E) 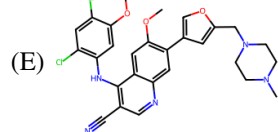

(F) 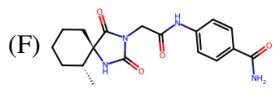

(G) 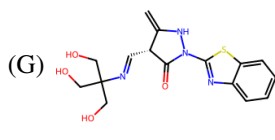

(H) 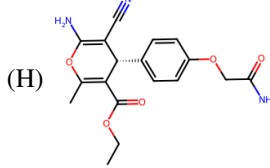

(I) 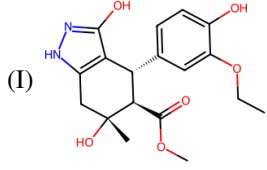

(J) 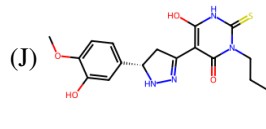

(K) 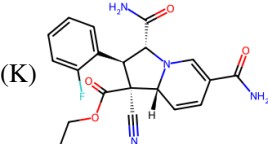

(L) 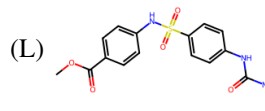

(M) 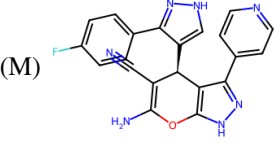

(N) 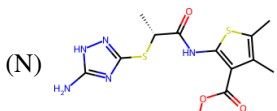

(O) 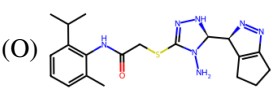

**Answer**:

A, B, C, D, E

## A.3 Earth

### A.3.1 Satellite-radar matching

This subtask evaluates a model's ability to match geostationary satellite imagery with corresponding radar-based VIL (Vertically Integrated Liquid) measurements. Given two infrared satellite channels (IR069 and IR107) capturing cloud-top properties, the objective is to identify the VIL image that best corresponds to the observed convective structures. The challenge lies in learning the non-trivial spatial and radiometric mappings between satellite brightness temperatures and radar reflectivity proxies, which vary across cloud regimes and precipitation intensities. This task benchmarks a model's capacity for multi-modal image understanding, emphasizing physical consistency and pattern recognition across heterogeneous atmospheric sensing modalities.

---

**Satellite-radar matching (E001)**

**Images**:

**Question**:
The first image is the IR069 image, the second image is the IR107 image, and the third to eighth images are VIL images, corresponding to options A to F. Which VIL image best matches the content shown in the IR069 and IR107 images?

**Options**:

---

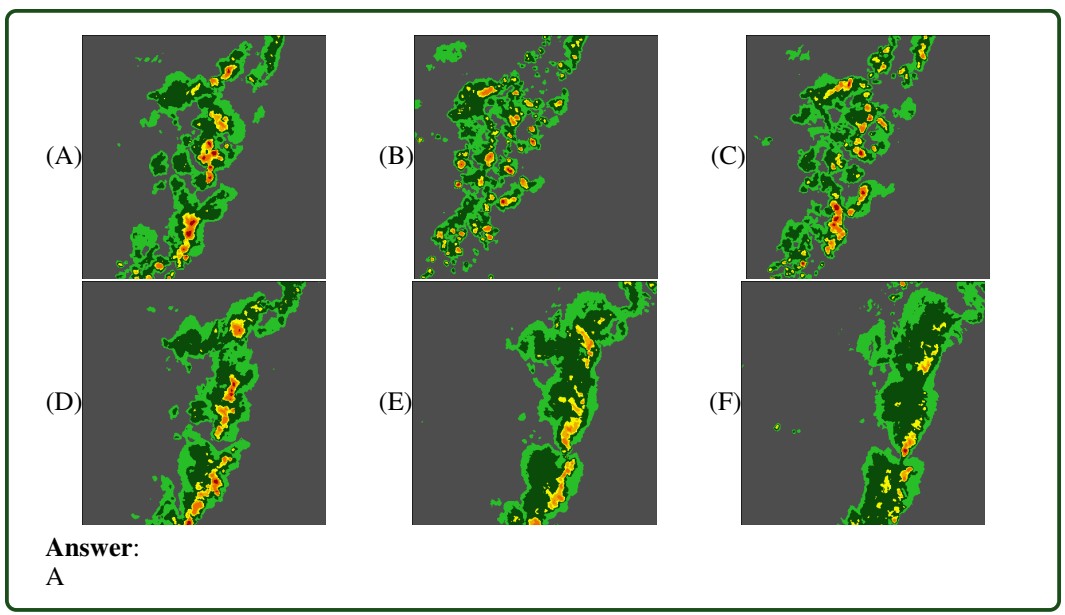

**Answer**:
A

### A.3.2 Perception of Extreme Precipitation Distribution

This subtask focuses on localized heavy precipitation detection from gridded satellite-based rainfall products. Given a daily accumulated precipitation map over East Asia, the goal is to identify specific geographic regions experiencing intense rainfall (greater than 50 mm). The task requires precise spatial reasoning and pattern recognition under varying rainfall intensities and distributions. Successful models must exhibit fine-grained geospatial understanding and demonstrate robust threshold-based detection under noisy meteorological conditions. By emphasizing accurate identification of rainfall extremes, this task serves as a critical benchmark for evaluating models in climate-aware perception and disaster risk assessment.

**Perception of Extreme Precipitation Distribution (E004)**

**Images**:

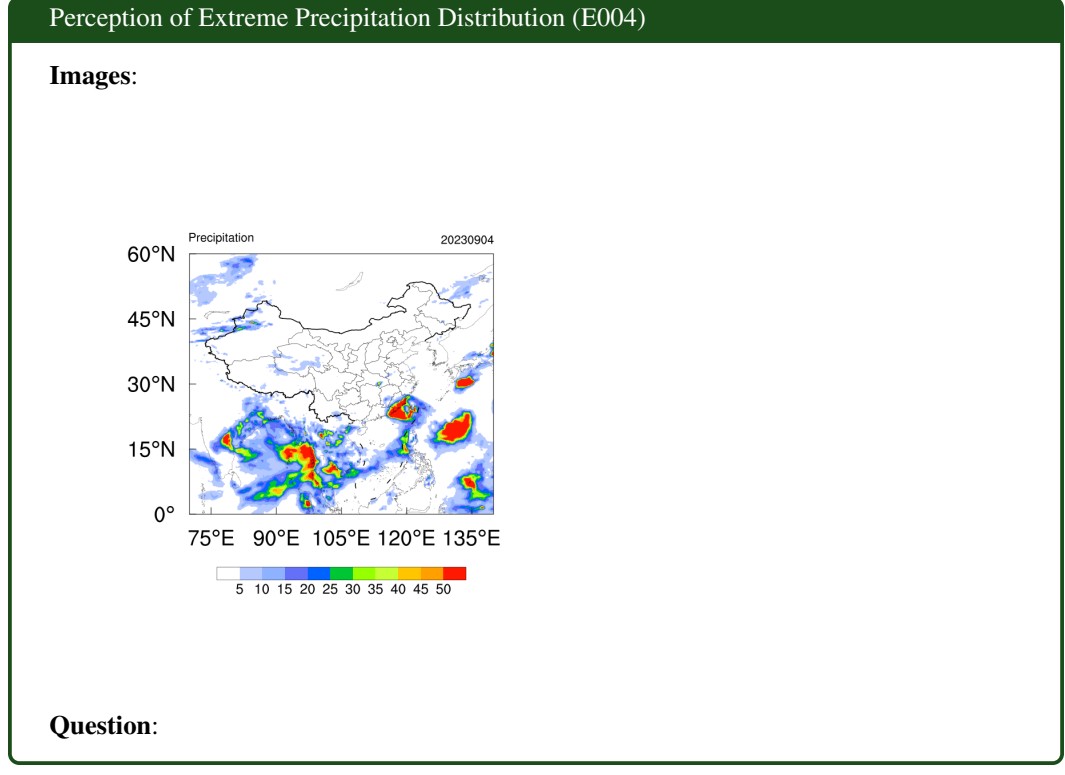

**Question**:

This is a map showing the daily accumulated precipitation in East Asia. Please select the precipitation area that includes regions with over 50 mm of rainfall from the following options.

**Options**:
(A) Southeast coastal region in China.
(B) The southern waters of Japan.
(C) Southern Philippines.
(D) West coast of India.
(E) Hainan Island.
(F) Southern Thailand.
(G) Korea.
(H) North China region.
**Answer**:
A, B, D, F

### A.3.3 Precipitation Event Analysis

This subtask targets causal reasoning in synoptic-scale meteorological diagnostics. Given multi-modal meteorological charts—including precipitation distribution, geopotential height fields, wind vectors, moisture flux, and vertical motion—the objective is to infer the underlying physical mechanisms driving precipitation events. The specific case focuses on eastern China and demands an accurate interpretation of trough positioning, moisture transport pathways, and dynamical lifting conditions. Successful solutions require a holistic understanding of atmospheric dynamics and their spatial interactions. This task evaluates the model's ability to integrate heterogeneous data modalities and produce structured, domain-specific explanatory analyses—a key step toward trustworthy AI in meteorological forecasting.

### Precipitation Event Analysis (E005)

**Images**:

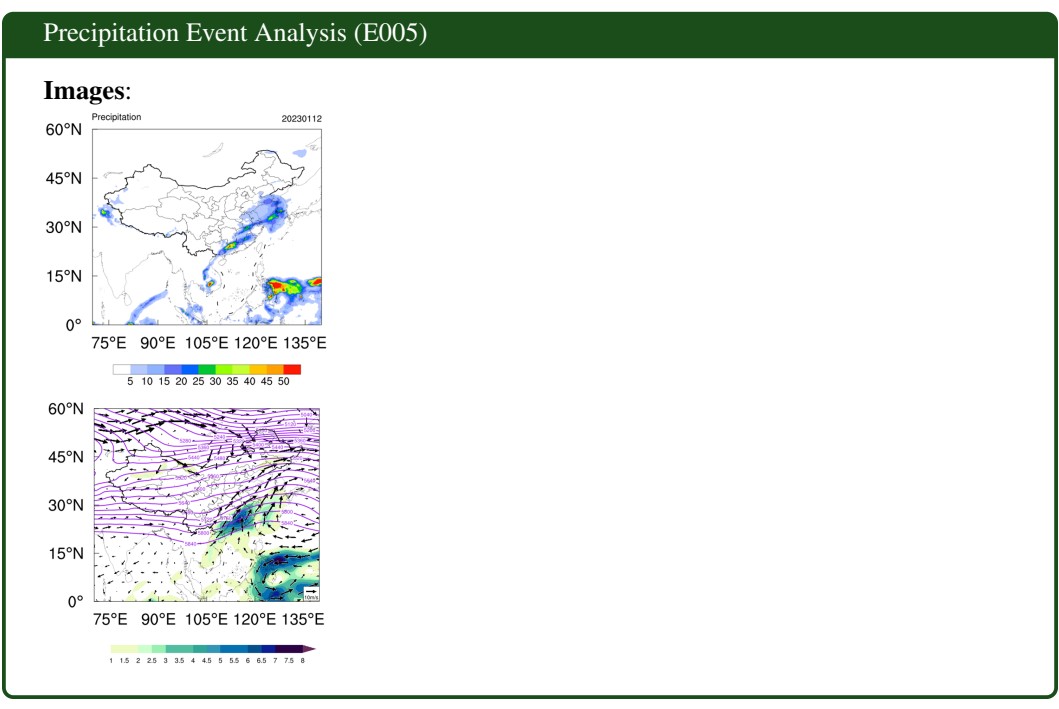

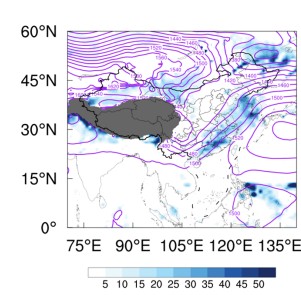

**Question**:
The first figure uses shading to represent precipitation. The second figure shows 500 hPa geopotential height with contour lines, 850 hPa wind field with vectors, and moisture flux with shading. The third figure displays 850 hPa geopotential height as contour lines and 500 hPa vertical velocity as shading. Please analyze the precipitation formation process in eastern China.

**Answer**:
The precipitation area is located ahead of a westerly trough, which extends southward from Gansu to Yunnan. The region is influenced by warm advection ahead of the trough and receives moisture transported from the South China Sea. The subtropical high over the northwestern Pacific is situated over the ocean east of the Philippines, steering moisture from the South China Sea toward the eastern coastal areas of China. In the precipitation zone, strong upward vertical motion leads to low-level atmospheric convergence, continuously supplying moisture.

### A.3.4 Moisture Source Distribution

This subtask focuses on identifying the dominant moisture source regions contributing to precipitation events, based on multi-variable meteorological analysis. Given composite plots of precipitation distribution and low-level atmospheric dynamics—including wind vectors and moisture flux—the task requires reasoning over spatial patterns to infer the most plausible moisture origins. The example centers on East China, where the South China Sea is revealed as the key moisture supplier through southerly transport. This task challenges models to associate physical variables across domains and to recognize geophysical transport pathways, encouraging the development of systems that understand and reason about real-world Earth system dynamics.

Moisture Source Distribution (E006)

**Images**:

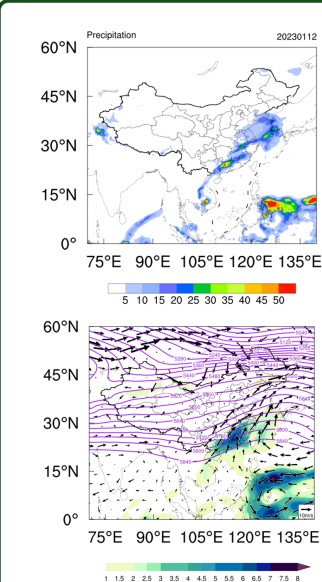

**Question**:
The first figure uses shading to represent precipitation. The second figure shows 500 hPa geopotential height with contour lines, 850 hPa wind field with vectors, and moisture flux with shading. What are the sources of moisture for precipitation in the East China region?

**Options**:
(A) Bohai Sea.
(B) Yellow Sea.
(C) East China Sea.
(D) South China Sea.
(E) Bay of Bengal.

**Answer**:
D

### A.3.5   Subtropical High Ridge Control Region

This subtask investigates the spatial extent of the Western Pacific Subtropical High Ridge through the analysis of synoptic-scale meteorological fields. The figure presents a combination of 500 hPa geopotential height, low-level wind vectors, and moisture flux shading. Participants are required to identify the geographic regions under the influence of the subtropical high, marked by anticyclonic circulation and suppressed moisture transport. The task demands spatial reasoning over multivariable meteorological patterns, reinforcing the model's understanding of atmospheric circulation systems. It emphasizes the capability to localize semi-permanent pressure systems and their dynamic control zones, crucial for modeling large-scale weather and climate variability.

Subtropical High Ridge Control Region (E007)

**Images**:

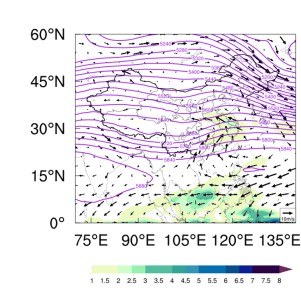

**Question**:
The figure shows 500 hPa geopotential height with contour lines, 850 hPa wind field with vectors, and moisture flux with shading. Please identify regions controlled by Western Pacific Subtropical High Ridge from the following options.

**Options**:
(A) Thailand.
(B) Eastern China.
(C) Southern China.
(D) Northern Philippines.
(E) India.
(F) Sri Lanka.
(G) Malaysia.
(H) Nepal

**Answer**:
E, F

### A.3.6 Thermocline Depth Recognition

This subtask targets the recognition of thermocline structures from vertical ocean temperature profile visualizations. Given a contour plot of ocean temperature variation with depth, models are tasked with identifying the depth range corresponding to the maximum vertical temperature gradient—an indicator of the thermocline. This requires spatial interpretation of temperature isolines and their compression zones. The task emphasizes fine-grained understanding of ocean stratification and vertical mixing processes, critical for climate dynamics and marine forecasting. By evaluating the ability to locate subtle thermal transitions, the subtask probes models' capacity to extract physical meaning from complex geophysical patterns.

---

Thermocline Depth Recognition (E008)

**Images**:

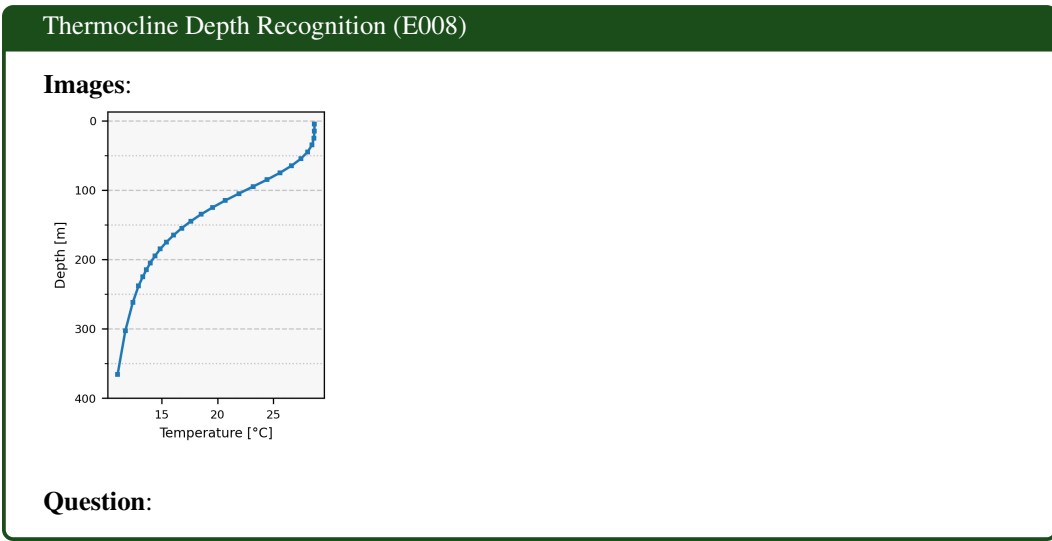

**Question**:

---

What is the approximate depth range of the maximum vertical temperature gradient in the figure?

**Options**:
(A) 50-150m.
(B) 100-200m.
(C) 150-250m.
(D) 200-300m.

**Answer**:
A

### A.3.7 SAR Image Grounding

This subtask centers on object grounding in synthetic aperture radar (SAR) imagery, specifically the detection and localization of maritime vessels. Given a high-resolution SAR image, models are required to identify all visible ships and annotate their positions using horizontal bounding boxes in normalized coordinates. The challenge lies in recognizing targets with diverse scales, orientations, and backscatter intensities under varying sea surface conditions. Unlike optical imagery, SAR data demands proficiency in interpreting texture and structure under coherent illumination. This task evaluates a model's capacity to robustly ground semantically meaningful objects in complex geospatial scenes beyond the visible spectrum.

---

**SAR Image Grounding (E011)**

**Images**:

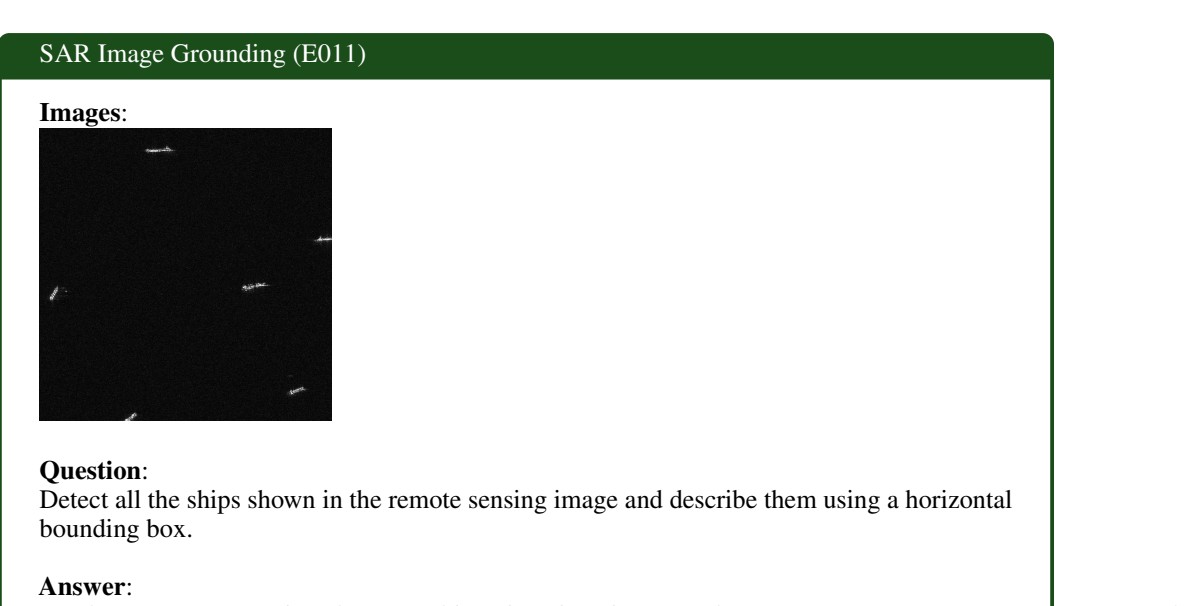

**Question**:
Detect all the ships shown in the remote sensing image and describe them using a horizontal bounding box.

**Answer**:
[0.95,0.37,1.00,0.39;0.69,0.53,0.79,0.56;0.36,0.06,0.46,0.09;0.85,0.88,0.91,0.91;0.29,0.97,0.34,1.00;0.04,0.54,0.07,0.59]

---

### A.3.8 Infrared Image Grounding

This subtask focuses on object grounding in thermal infrared imagery, targeting the detection of vehicles in low-light or thermally dominant environments. Given an infrared image, the model must localize all visible cars using horizontal bounding boxes in normalized coordinates. Unlike RGB images, infrared data emphasizes temperature contrasts, posing unique challenges for distinguishing objects with similar thermal signatures or partially occluded forms. Accurate detection under these conditions reflects a model's robustness to modality shifts and its ability to exploit spatial and thermal cues. This task is a critical benchmark for grounding algorithms in non-visible spectra under real-world conditions.

### A.3.9 Temperature Sequence Comparison

This subtask evaluates the model's capacity for fine-grained temporal reasoning over climate data. Given a global annual temperature series spanning 1981 to 2000, the task requires analyzing two consecutive decades to extract key statistical characteristics—value range, extremes, and temporal trends—followed by comparative insights between periods. Successful completion demonstrates the model's ability to interpret dense visual time series and capture nuanced variations in climate dynamics. By grounding numeric trends in specific temporal intervals, this task benchmarks a model's capability for structured temporal analysis, a core skill for scientific understanding and decision-making in environmental monitoring and forecasting.

Temperature Sequence Comparison (E014)

**Images**:

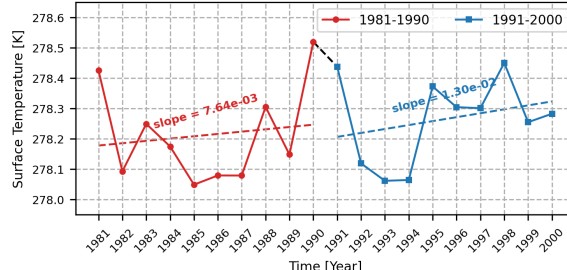

**Question**:
The figure below is a global annual average temperature series from 1981 to 2000, describing the temperature information (range, extreme values and trends) for the time periods 1981-1990 and 1991-2000, as well as the difference between the two time periods.

**Answer**:
The temperature range in 1981-1990 is from 278.0 to 278.5, with the minimum value occurring in 1985 and the maximum value occurring in 1990, showing an overall increasing trend. The

temperature range in 1991-2000 is from 278.1 to 278.5, with the minimum value occurring in 1993 and the maximum value occurring in 1998, showing an overall increasing trend. The minimum temperature in 1981-1990 is lower, the maximum temperature in 1981-1990 is higher, and the growth trend in 1991-2000 is greater.

### A.3.10 Vertical Profile Comparison

This subtask investigates the model's ability to compare and analyze vertical oceanographic profiles. The input consists of a plot showing two temperature (or other scalar) profiles across varying ocean depths. The model is required to identify the depth intervals where the discrepancy between the two profiles is the most significant. This tests the model's proficiency in recognizing depth-wise gradient differences and isolating meaningful regions of divergence. Accurate performance in this task reflects the model's capacity for precise visual differentiation in scientific plots, a foundational skill for downstream applications such as anomaly detection, climate diagnostics, and ocean monitoring.

---

**Vertical Profile Comparison (E016)**

**Images**:

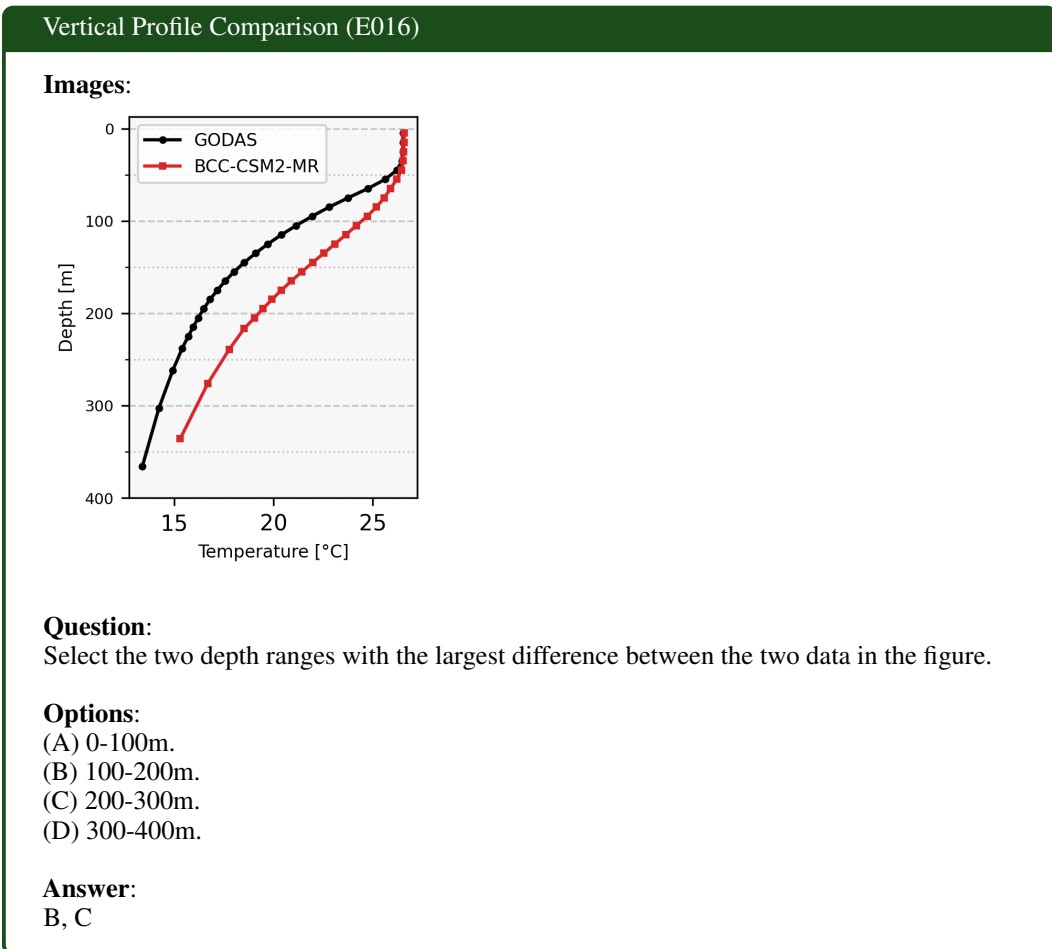

**Question**:
Select the two depth ranges with the largest difference between the two data in the figure.

**Options**:
(A) 0-100m.
(B) 100-200m.
(C) 200-300m.
(D) 300-400m.

**Answer**:
B, C

---

### A.3.11 Outlier Analysis

This subtask evaluates the model's capability in identifying regional outliers within a global temperature anomaly map. The input is a 2D spatial distribution of surface temperature anomalies, where the model must pinpoint the regions exhibiting the most extreme warm and cold deviations. This task necessitates nuanced spatial pattern recognition and comparative reasoning to distinguish subtle yet climatically significant anomalies. The correct identification of both hot and cold outliers demonstrates the model's effectiveness in high-resolution geospatial anomaly detection, an essential competency for climate diagnostics, extreme event attribution, and model evaluation in Earth system science.

**Images**:

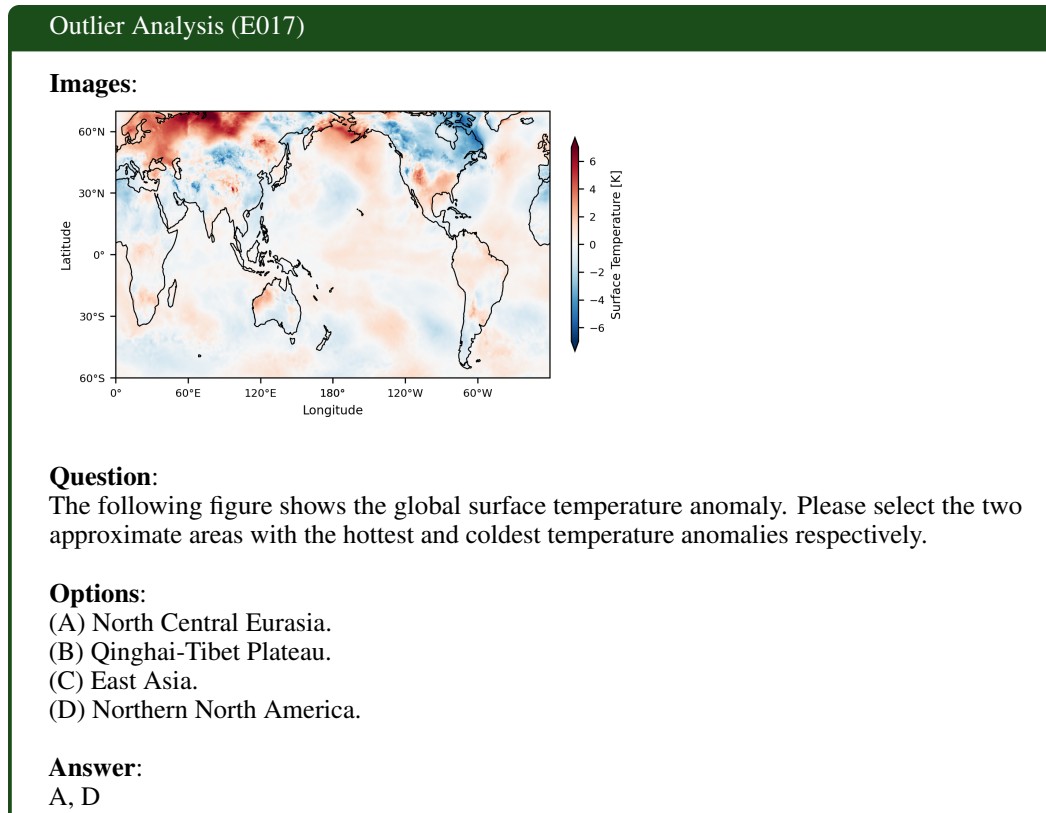

**Question**:
The following figure shows the global surface temperature anomaly. Please select the two approximate areas with the hottest and coldest temperature anomalies respectively.

**Options**:
(A) North Central Eurasia.
(B) Qinghai-Tibet Plateau.
(C) East Asia.
(D) Northern North America.

**Answer**:
A, D

### A.3.12 Differential Prediction Comparison

This subtask challenges the model to perform spatiotemporal comparison between observed and forecasted radar reflectivity sequences, focusing on Vertical Integrated Liquid (VIL) as an indicator of precipitation intensity. The model must identify key discrepancies in intensity evolution, spatial distribution, and movement patterns over time. Successfully capturing differences such as forecast underestimation of scattered echoes, intensity weakening, and deviation in convective system trajectories demonstrates the model's ability to assess forecast accuracy and temporal dynamics. This task is critical for advancing precipitation nowcasting and improving understanding of model limitations in representing complex atmospheric convection processes.

**Images**:

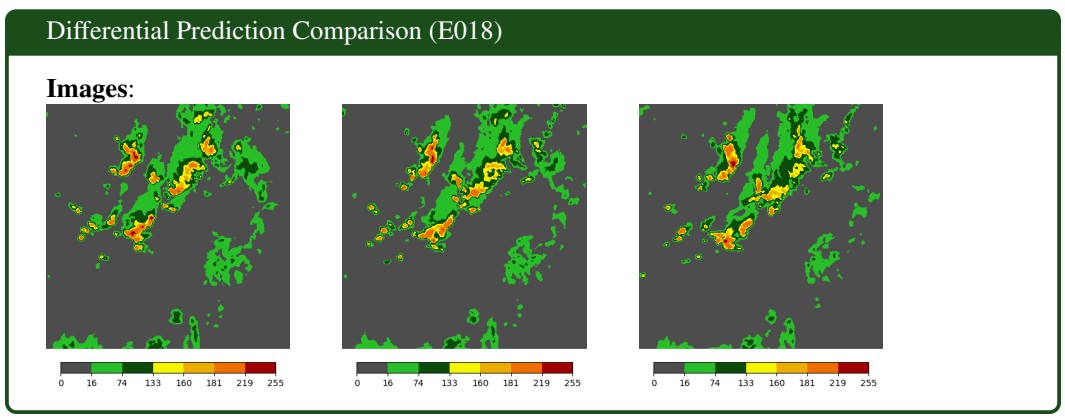

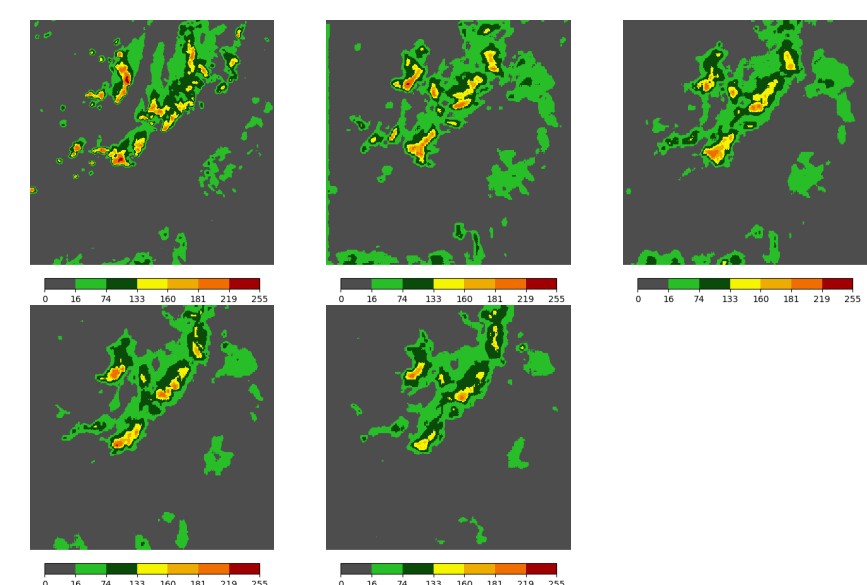

**Question**:
The first 4 images illustrate the observed radar reflectivity sequence, and the last 4 images present the forecasted sequence. The variable visualized in both sequences is the Vertical Integrated Liquid (VIL), which represents the total amount of liquid water contained in a vertical column of the atmosphere. value between 0 and 16 indicates high probability of a sunny weather; value between 16 and 74 indicates high probability of light rain; value between 74 and 133 indicates high probability of moderate rain; value between 133 and 160 indicates high probability of heavy rain; value between 160 and 181 indicates high probability of very heavy rain; value between 181 and 219 indicates high probability of intense rain; value above 219 indicates high probability of extreme rain .Please highlight the major discrepancies between the forecasted and observed fields.

**Answer**:
The forecast sequence exhibits a rapid weakening of intensity, with an overall trend of strengthening first and then weakening, while the observed sequence maintains a relatively steady intensity. Small scattered echoes are not captured in the forecast sequence. Additionally, the convective systems in the observed sequence show a southeastward movement, whereas the forecasted systems remain largely stationary.

## A.3.13 Convective Weather Types Identification

This subtask requires the model to integrate multi-parameter meteorological data—such as wind shear, supercell composite indices, temperature advection, and reflectivity mosaics—to identify geographic regions at highest risk of severe convective weather. By synthesizing dynamic atmospheric instability, moisture, and storm maintenance probabilities across spatially distributed observations, the model must pinpoint the area most susceptible to severe weather impacts. This task evaluates the model's capacity to fuse heterogeneous weather indicators for robust convective event prediction, advancing interpretability and accuracy in severe weather nowcasting and supporting timely hazard mitigation efforts.

Convective Weather Types Identification (E021)

**Images**:

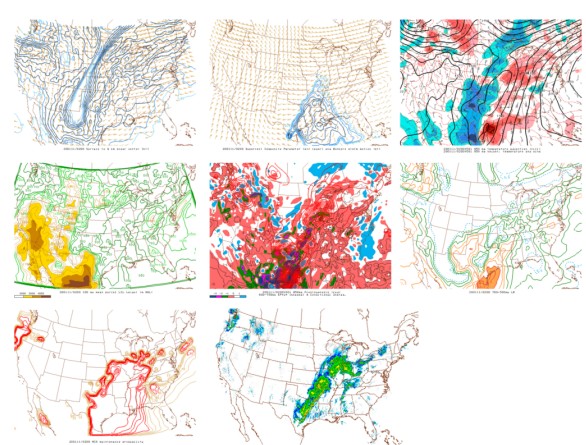

**Question**:
The following 8 figures represent weather conditions and each figure contains multiple weather parameters. The most important parameters in each figure is provided as follow:Figure 1: 0-6km wind shear magnitude. Figure 2: Supercell Composite Parameter. Figure 3: 850 mb Temperature Advection. Figure 4: Lifting Condensation Level height. Figure 5: Geostrophic equivalent potential vorticity. Figure 6: 700-500mb lapse rate instability. Figure 7: Mesoscale Convective System maintenance probability. Figure 8: Base Reflectivity Mosaic. Choose one geographical area(s) most likely to be impacted by the severe weather event from the four options provided.

**Options**:
(A) east Texas.
(B) east-central Minnesota into northwest Wisconsin.
(C) southern Indiana into Kentucky.
(D) eastern and middle Tennessee into northwestern Alabama and northern Mississippi.

**Answer**:
A

### A.3.14  Convective Influence regions Identification

This subtask challenges the model to assess severe weather potential by synthesizing multi-dimensional meteorological parameters, including wind shear, temperature advection, convective instability, and reflectivity patterns. The objective is to determine the likelihood and timing of convective watches or relevant winter weather advisories. This requires a nuanced interpretation of probabilistic indicators and dynamic atmospheric conditions to classify the scenario into distinct watch categories or winter weather events. The task evaluates the model's ability to integrate complex weather data and uncertainty estimates, advancing predictive precision for operational forecasting and early warning decision-making in severe weather contexts.

Convective Influence regions Identification (E022)

**Images**:

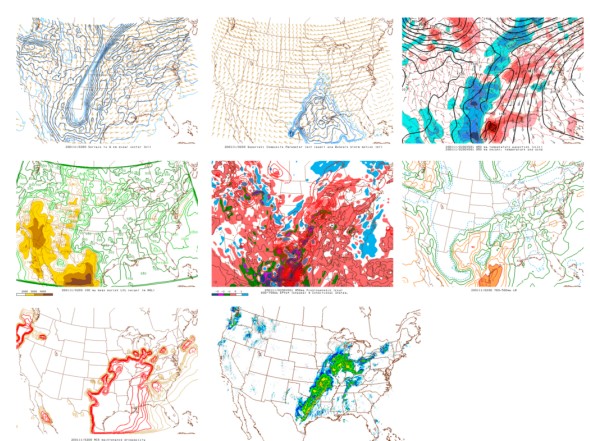

**Question**:
The following 8 figures represent weather condition,s and each figure contains multiple weather parameters. The most important parameters in each figure is provided as follow:Figure 1: 0-6km wind shear magnitude. Figure 2: Supercell Composite Parameter. Figure 3: 850 mb Temperature Advection. Figure 4: Lifting Condensation Level height. Figure 5: Geostrophic equivalent potential vorticity. Figure 6: 700-500mb lapse rate instability. Figure 7: Mesoscale Convective System maintenance probability. Figure 8: Base Reflectivity Mosaic. Choose the most likely scenario regarding severe weather concerns from the options provided. Options include whether a convective watch has been issued, the probability of a future watch, or, in winter, potential weather phenomena related to winter storms.

**Options**:
(A) Severe potential...Watch unlikely (confidence level on the expectation of a watch is 5 or 20%).
(B) Severe potential...Watch possible (confidence level on the expectation of a watch is 40 or 60%).
(C) Severe potential...Watch likely (confidence level on the expectation of a watch is 80 or 95%).
(D) Heavy snow.
(E) Severe potential...tornado watch likely (confidence level on the expectation of a watch is 80 or 95%)
(F) Severe potential...severe thunderstorm watch likely (confidence level on the expectation of a watch is 80 or 95%)
(G) Winter mixed precipitation.
(H) Freezing rain.
(I) Severe potential...watch needed soon (confidence level on the expectation of a watch is 95%).
(J) Blizzard.
(K) Snow squall.

**Answer**:
A

## A.4 Life Science

### A.4.1 Fragment Ion Peaks Count (L001)

In this subtask, we task the model with the quantitative analysis of MS/MS spectra for peptide sequencing, focusing specifically on the identification and enumeration of fragment ion peaks. Given a peptide sequence and its associated spectrum, the model must accurately count the number of observed b ion and y ion peaks. This requires precise understanding of peptide fragmentation patterns

and careful interpretation of spectral data. The model's output is formatted as a structured JSON object, enabling direct assessment of its peptide sequencing and mass spectrometry expertise. This subtask tests fundamental capabilities vital for automated proteomics data analysis.

---

**Fragment Ion Peaks Count (L001)**

**Images**:

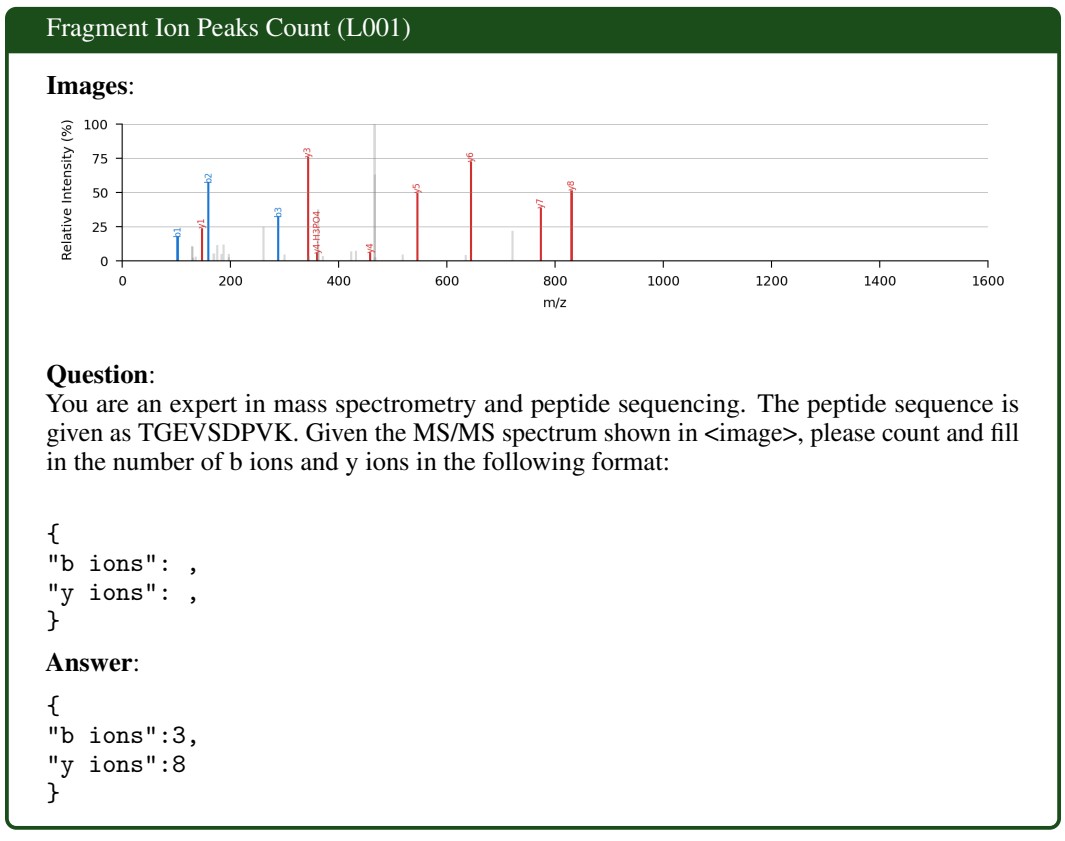

**Question**:
You are an expert in mass spectrometry and peptide sequencing. The peptide sequence is given as TGEVSDPVK. Given the MS/MS spectrum shown in <image>, please count and fill in the number of b ions and y ions in the following format:

```
{
"b ions": ,
"y ions": ,
}
```

**Answer**:

```
{
"b ions":3,
"y ions":8
}
```

---

### A.4.2 De Novo Peptide Sequence (L003)

For this subtask, we focus on the challenging problem of de novo peptide sequencing from tandem mass spectrometry (MS/MS) data. Unlike database search approaches, de novo sequencing requires inferring the peptide sequence directly from the observed spectrum, relying solely on fragmentation patterns and precise mass measurements. We provide spectra alongside an explicit set of amino acid and modification masses, removing ambiguity in interpretation. The task emphasizes expert-level reasoning, demanding accurate mapping from spectral peaks to sequence candidates, and evaluation of the most plausible peptide given all evidence. This subtask provides a rigorous benchmark for de novo sequencing algorithms and human expertise.

---

**De Novo Peptide Sequence (L003)**

**Images**:

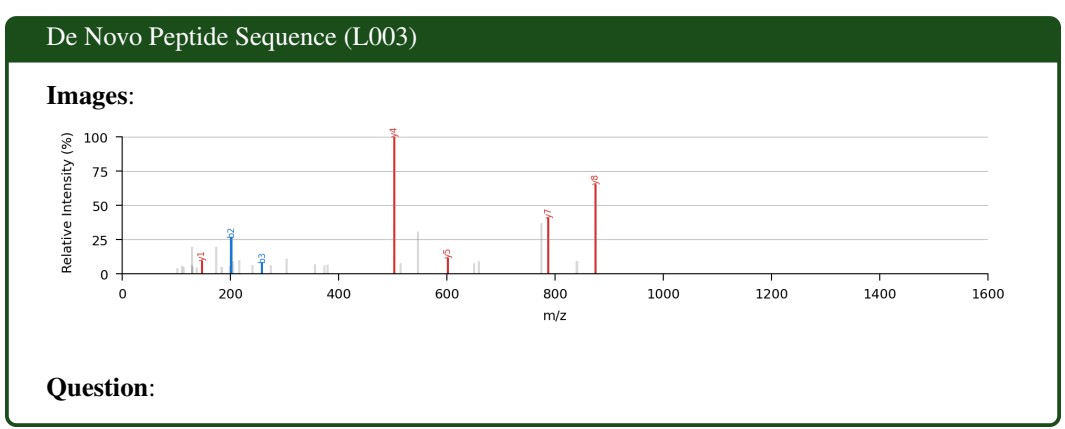

**Question**:

---

You are an expert in mass spectrometry and peptide sequencing. Given an MS/MS spectrum, your task is to predict the amino acid sequence of the peptide that generated the spectrum using your expert knowledge of peptide fragmentation patterns and *de novo* sequencing principles. The masses of amino acids and modifications are as follows:
G: 57.021464, A: 71.037114, S: 87.032028, P: 97.052764, V: 99.068414, T: 101.047670, C+57.021: 160.030649, L: 113.084064, I: 113.084064, N: 114.042927, D: 115.026943, Q: 128.058578, K: 128.094963, E: 129.042593, M: 131.040485, H: 137.058912, F: 147.068414, R: 156.101111, Y: 163.063329, W: 186.079313, M+15.995: 147.035400, N+0.984: 115.026943, Q+0.984: 129.042594, +42.011: 42.010565, +43.006: 43.005814, -17.027: -17.026549, +43.006-17.027: 25.980265. Given the MS/MS spectrum shown in <image>, determine the most likely peptide sequence. Please write your answer as a single peptide sequence.

**Answer**:
ISGEVPEEK

### A.4.3   Atom Count Inference (L005)

The Atom Count Inference subtask is designed to evaluate a model's capability in deducing specific elemental content—such as the number of sulfur atoms—using MS/MS spectrometric data. Each instance presents a compound's spectrum and prompts the model to infer element counts, leveraging isotope distributions and spectral evidence. This task challenges the model's understanding of mass spectrometry, particularly its ability to interpret subtle isotopic features, and requires integration of both quantitative peak information and qualitative spectral patterns. Through this setup, we systematically probe reasoning on elemental composition given real-world instrument data.

## Atom Count Inference (L005)

**Images**:

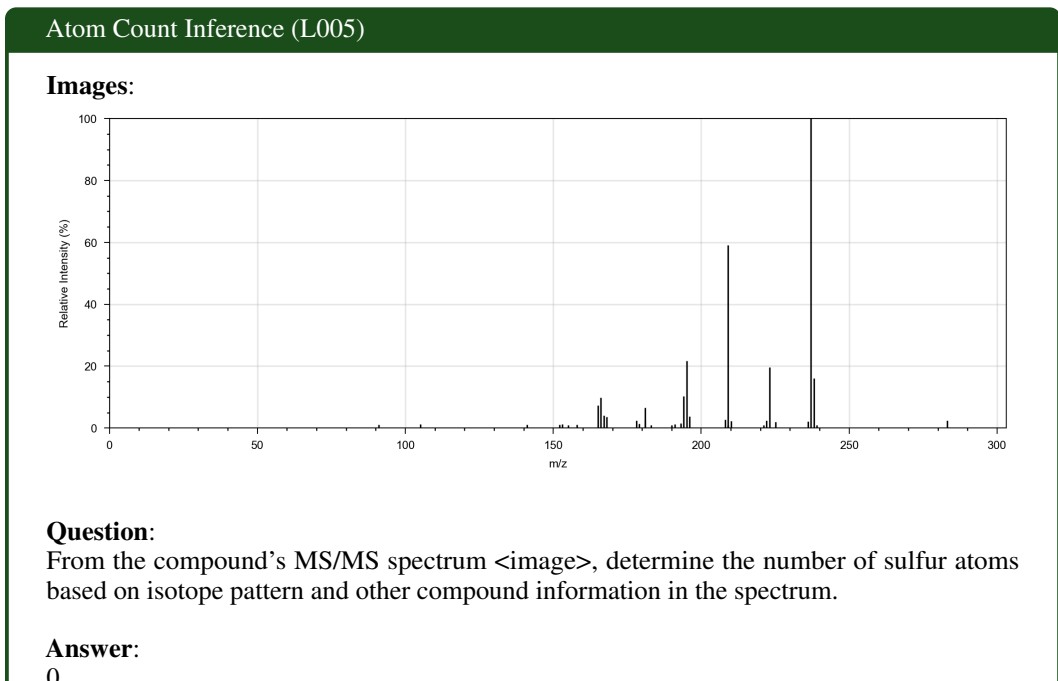

**Question**:
From the compound's MS/MS spectrum <image>, determine the number of sulfur atoms based on isotope pattern and other compound information in the spectrum.

**Answer**:
0

### A.4.4   Molecular Composition Inference (L006)

The Molecular Composition Inference subtask evaluates a model's capability to deduce the elemental makeup of a compound directly from its MS/MS spectral data. This is a crucial step in metabolomics analysis, where complex spectra encode rich information about molecular structure. By examining isotope patterns and fragmentation signals, the model is expected to identify which elements (such as C, H, O, N, S, etc.) constitute the parent compound. This subtask focuses on the essential challenge

of abstracting key chemical information from raw experimental data, providing a fundamental basis for subsequent molecular structure elucidation.

---

**Molecular Composition Inference (L006)**

**Images**:

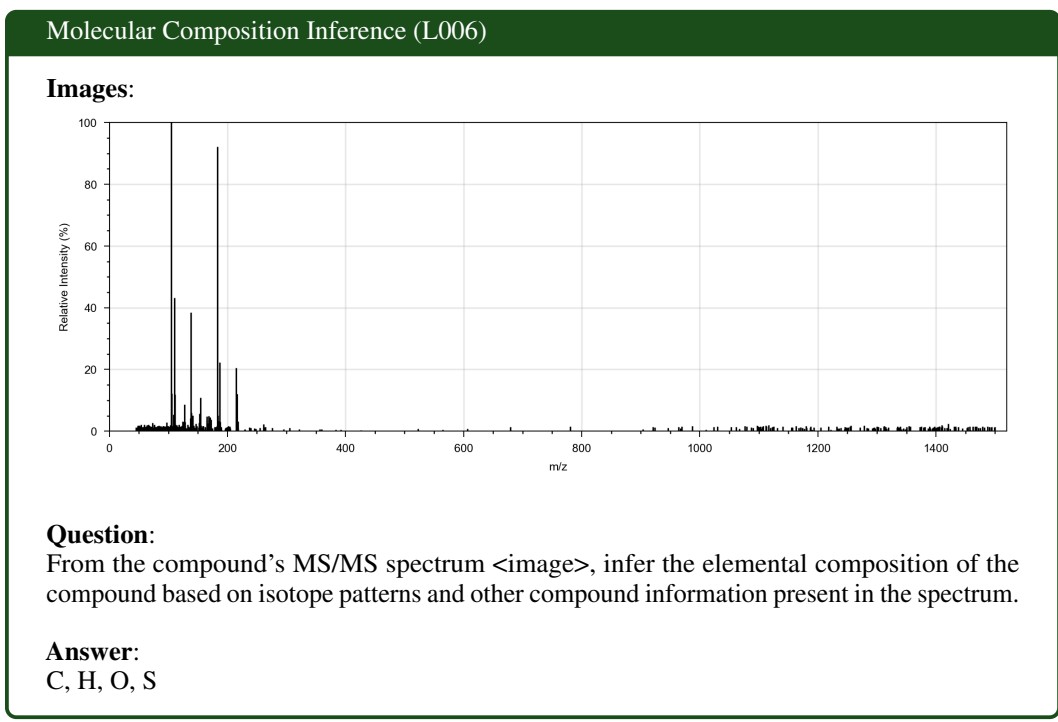

**Question**:
From the compound's MS/MS spectrum <image>, infer the elemental composition of the compound based on isotope patterns and other compound information present in the spectrum.

**Answer**:
C, H, O, S

---

### A.4.5  Spectrum Matching (L007)

The Spectrum Matching subtask aims to evaluate the ability of models to interpret mass spectra and associate them with their corresponding molecular structures. Given a mass spectrum as input, the model must select the correct structure from a set of plausible candidates, each rendered as a chemical diagram. This task requires not only low-level pattern recognition in spectra but also high-level understanding of molecular fragmentation pathways. Accurate matching necessitates reasoning over both spectral peaks and chemical features, thus offering a rigorous and realistic testbed for machine understanding in the intersection of cheminformatics and machine learning.

---

**Spectrum Matching (L007)**

**Images**:

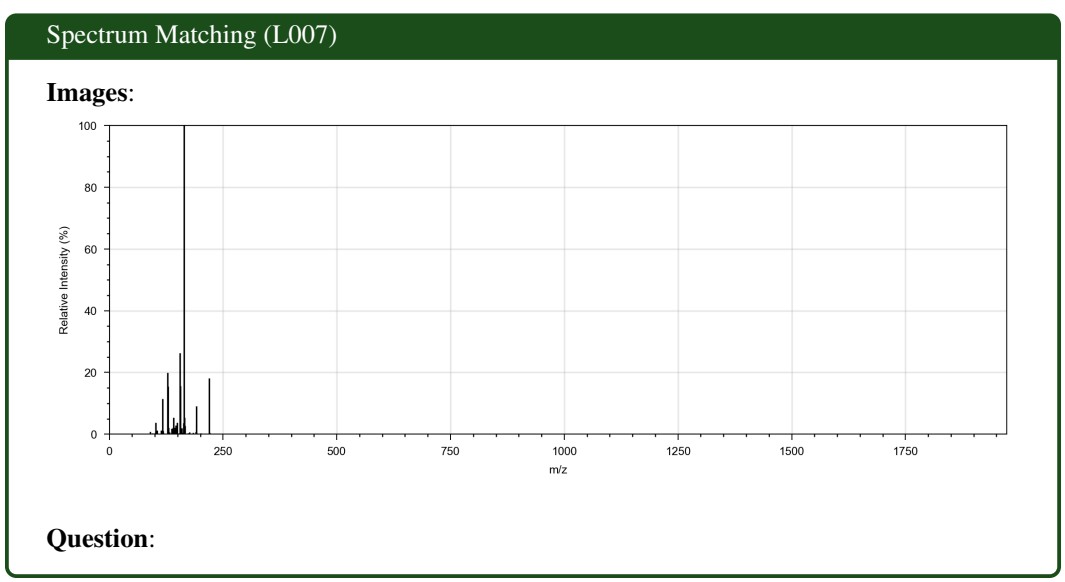

**Question**:

---

This is a multiple-choice question with only one correct answer. Based on the given mass spectrum <image>, select the molecular structure that best matches it.

**Options**:

(A) 

(B) 

(C) 

(D) 

(E) 

(F) 

(G) 

**Answer**:
B

### A.4.6  Protein Chain Count (L008)

The Protein Chain Count subtask is designed to evaluate the model's capability of understanding and interpreting molecular structural data as visualized in PDB format images. Specifically, given a protein's structural image where hydrogen atoms are removed for clarity, the objective is to accurately enumerate the distinct protein chains present. This requires the model to correlate visual clusters or segments within the image to underlying biomolecular structures, demonstrating an integration of visual perception and biochemical knowledge. Such a task serves as a fundamental assessment of the model's proficiency in recognizing discrete biological entities from complex visual input, echoing challenges in real-world scientific analysis.

Protein Chain Count (L008)

**Images**:

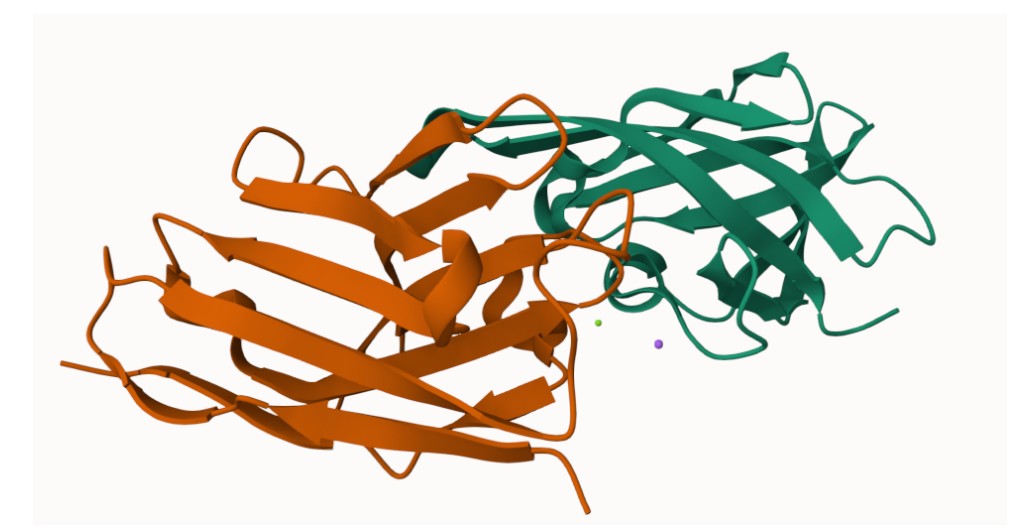

**Question**:
From the molecular structure PDB image <image>, (remove hydrogen atoms), please answer the number of protein chains with integers.

**Answer**:
2

### A.4.7 Small Molecule Count (L009)

The Small Molecule Count (L009) subtask focuses on the quantitative analysis of biomolecular structures by requiring models to accurately discern and enumerate ligand chains present in PDB images, excluding hydrogen atoms. Given a molecular visualization, the task challenges algorithms to capture subtle structural cues indicative of ligand entities, testing both spatial understanding and chemical knowledge. This subtask is pivotal for evaluating a model's ability to interpret complex biochemical illustrations, further advancing automated structural biology. Our formulation aligns with real-world use cases in molecular docking and drug discovery, where correct ligand identification underpins successful downstream applications.

Small Molecule Count (L009)

**Images**:

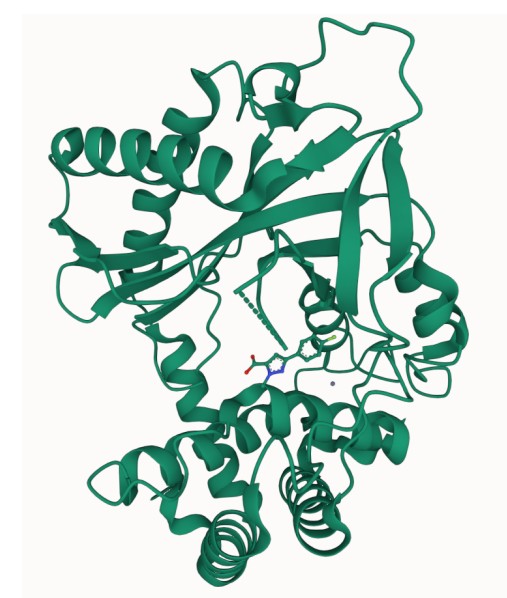

**Question**:
From the molecular structure PDB image <image>, (remove hydrogen atoms), please answer the number of ligand chains with integers.

**Answer**:
2

### A.4.8 Specified Protein Detection (L010)

The Specified Protein Detection subtask is designed to evaluate a model's capability in recognizing and classifying protein structures from visual representations. Given an input image depicting a protein's tertiary or quaternary structure, the model is required to identify the functional class the protein belongs to among several biologically relevant categories, such as enzymes, structural proteins, or receptors. This task assesses both the fine-grained visual understanding and the domain-specific reasoning of the model, mirroring real-world scenarios in computational biology where structural inference is pivotal for downstream applications, such as drug discovery or pathway analysis.

Specified Protein Detection (L010)

**Images**:

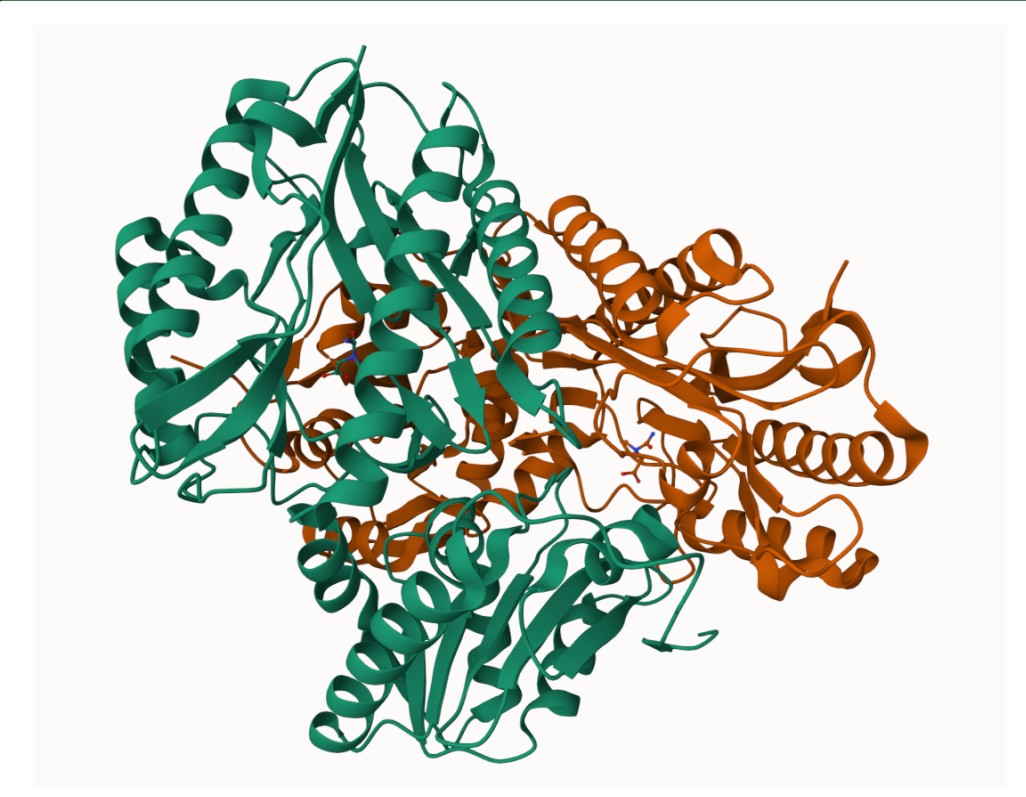

**Question**:
Based on the structures <image>, choose the most suitable description of the protein structure.

**Options**:
(A) Enzymes
(B) Storage Proteins
(C) Structural Proteins
(D) Transport Proteins
(E) Signaling Proteins
(F) Regulatory Proteins
(G) Motor Proteins
(H) Receptor Proteins
(I) Defensive Proteins

**Answer**:
A

### A.4.9 Protein Structure Feature Analysis (L019)

In this subtask, we evaluate the model's capacity to interpret and describe high-resolution molecular structures from PDB crystal images. The task requires the model to identify the protein's overall fold architecture, recognize salient secondary structure elements such as $\alpha$-helices and $\beta$-sheets, and locate bound ligands or cofactors within the protein's interior. By analyzing the spatial arrangement of atoms and surface features, the model is expected to provide concise, structurally grounded summaries that reflect an understanding of protein-ligand interactions. This setting emphasizes precise perception and reasoning over intricate spatial features essential in molecular biology and rational drug design.

**Images**:

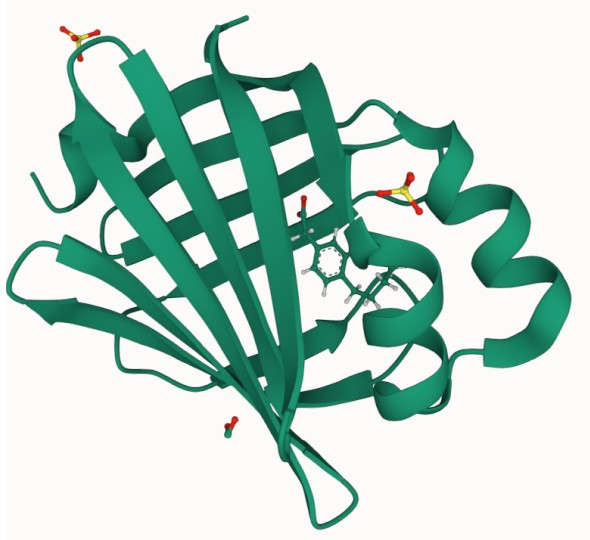

**Question**:
From the molecular structure PDB image <image>, which is from Crystal Structure of human FABP4 in complex with 2-(5,6,7,8,9,10-hexahydrobenzo[8]annulen-3-yl)acetic acid, please give description of this structure in one paragraph.

**Answer**:
The provided image displays the crystal structure of human Fatty Acid Binding Protein 4 (FABP4) in complex with the ligand 2-(5,6,7,8,9,10-hexahydrobenzoannulen-3-yl)acetic acid. The FABP4 protein, depicted in teal, adopts a characteristic $\beta$-barrel fold, composed of ten antiparallel $\beta$-strands that form a clam-shell like structure, creating an internal cavity. This cavity is where the ligand, shown in a ball-and-stick representation with grey/white carbons and red oxygens, is bound, indicating its role in sequestering and transporting fatty acids or related hydrophobic molecules. Short $\alpha$-helical segments cap the ends of the $\beta$-barrel. Additionally, sulfate ions, likely from the crystallization buffer, are visible on the protein's surface, represented by yellow sulfur atoms and red oxygen atoms.

### A.4.10 Structural Domains Identification (L014)

The Structural Domains Identification subtask aims to assess the model's capacity for fine-grained recognition of canonical and non-canonical RNA secondary structure elements from diagrammatic representations. Given a visualized RNA fold, the task requires discerning a range of features, including stems, hairpin,s and interior loops, bulges, multi-branched loops, and complex topologies such as pseudoknots. Precise identification of these motifs is crucial for understanding RNA's functional and structural diversity. This subtask provides a rigorous benchmark for evaluating both domain-specific pattern recognition and the broader generalization ability of the model in RNA structure analysis.

## Structural Domains Identification (L014)

**Images**:

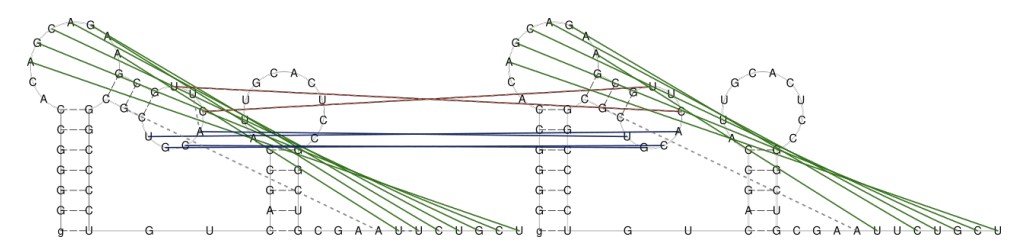

**Question**:
Based on the RNA secondary structure diagram <image>, which of the following structures might the RNA contain? (Multiple choices).

**Options**:
(A) stem
(B) Hairpin loop
(C) Interior loop
(D) bulge loop
(E) Multi-branched loop
(F) Pseudoknot

**Answer**:
A, B, C, F

### A.4.11 Structural Domain Count (L015)

The Structural Domain Count subtask is designed to assess the ability of models to interpret RNA secondary structure diagrams and accurately quantify the number of distinct stem regions present. Each instance presents a clear visual representation of an RNA structure, enabling the systematic evaluation of structural comprehension. This subtask emphasizes precise recognition of canonical stem features, free from confounding sequence information, and focuses on developing visual reasoning skills in a biological context. By reporting the count of stem regions, the task provides a straightforward yet challenging benchmark for assessing the model's proficiency in structural annotation, a critical component for molecular understanding.

**Structural Domain Count (L015)**

**Images**:

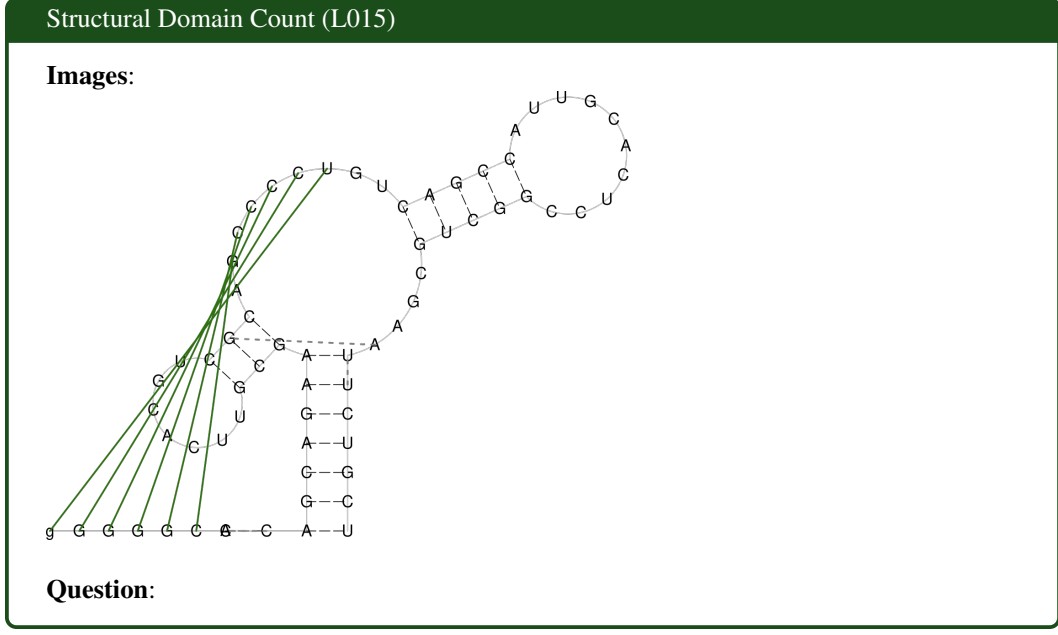

**Question**:

Examine the given RNA secondary structure diagram <image>. How many stem regions are present in the structure?

**Answer**:
3

## A.4.12 RNA Type Identification (L016)

This subtask focuses on RNA secondary structure inverse folding: given a diagram of an RNA secondary structure, the model must accurately classify the type of RNA it represents. The task requires the model to reason about spatial arrangements and base-pairing patterns from structural images, rather than relying on primary sequence information. This challenging setting tests the model's ability to recognize and understand biologically meaningful motifs visualized within complex structures. Solving this subtask sheds light on the model's capacity for visual pattern recognition in molecular biology, bridging the gap between image-based data and biomolecular function classification.

## A.4.13 RNA Secondary Structure Inverse Folding (L017)

The RNA Secondary Structure Inverse Folding subtask requires participants to infer the precise nucleotide sequence corresponding to a given RNA secondary structure image. Expert-level reading skills are essential, as the task demands meticulous extraction and transcription of the displayed bases, adhering strictly to biological conventions—notably, replacing thymine (T) with uracil (U) where applicable. Sequences are reported in the 5' to 3' direction, formatted as specified JSON outputs. This task rigorously tests both structural comprehension and attention to detail, providing a challenging benchmark for evaluating the capability of models in sequence-structure reasoning within biomolecular domains.

## RNA Secondary Structure Inverse Folding (L017)

**Images**:

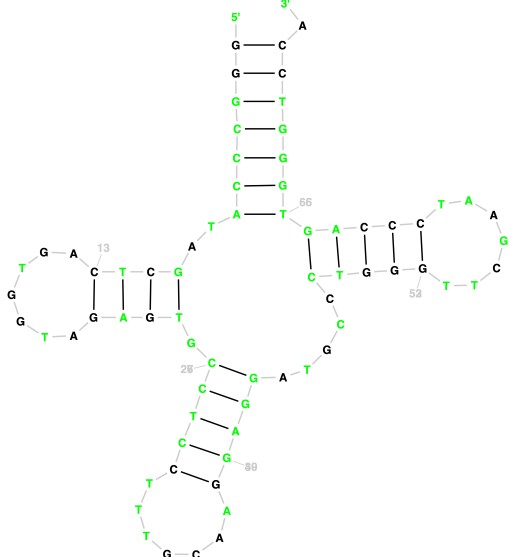

**Question**:

You are a professional biologist, and you are skilled at reading RNA sequences. From the RNA secondary structure <image>, give the exact RNA sequence. Answer in capital letters only, from 5' to 3'. If there exists any T base in the picture, replace them into U base. Please directly output the sequence.

**Answer**:

GGGCCCAUAGCUCAGUGGUAGAGUGCCUCCUUUGCAAGGAGGAUGCCCUGGGUUCGAAUCCCAGUGGGUCCA

## A.4.14 Structural Motifs and Positions Description (L020)

## Structural Motifs and Positions Description (L020)

**Images**:

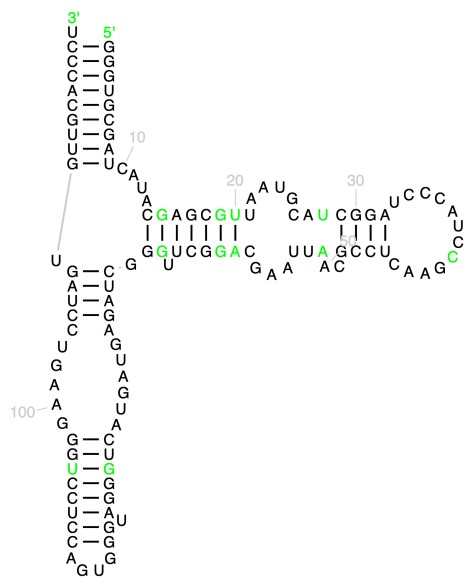

**Question**:
You are a professional biologist, andy you are skilled at reading and analyzing RNA secondary structure pictures. Please generate a natural language description for the given RNA <image>, find the length, describe the internal loops and hairpin loops of the RNA and point out their start and terminate position.

**Answer**:
The RNA is comprised of 119 bases in total, which has two internal loops from position U21 to C26, position U53 to C57; from position A71 to A78 and from position G9 to C104, and it also has two hairpin loops from position U33 to C44 and from position G85 to A90

## A.5 Materials Science

### A.5.1 Atomic Composition Description (M001)

This subtask evaluates the capability to interpret and reason about the atomic-scale structure of crystalline materials, as represented by lattice diagrams. The model is required to analyze detailed crystallographic features—including anion and cation arrangements, coordination environments, and stacking sequences—by selecting all correct statements from a set of options. The format enforces concise, decisive outputs, mirroring critical reading and matching of visual and textual information. In alignment with rigorous scientific inquiry, candidates are discouraged from speculation; only substantiated facts, as visually and structurally evidenced in the lattice, are to be identified.

Atomic Composition Description (M001)

**Images**:

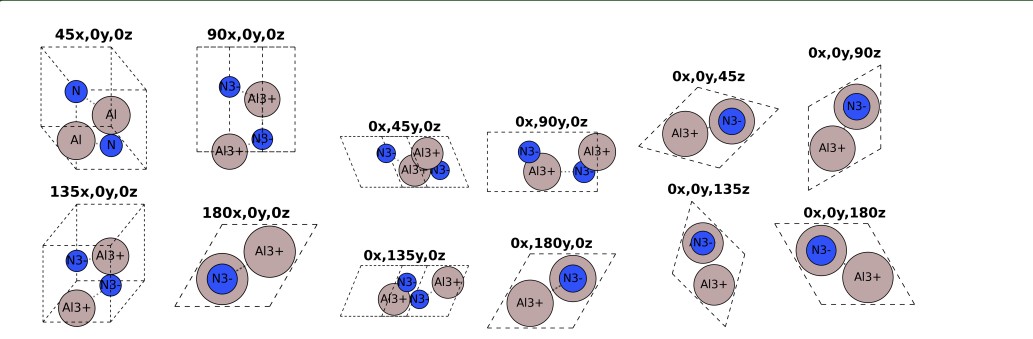

**Question**:
Please strictly follow the format—no additional text or explanation may be provided.
You should only reply with the letter(s) of the correct option(s) (e.g., A, C) corresponding to the descriptions that match the diagram.
Based on the different views of a material: front view <image>, side view <image> and top view <image>

**Options**:
(A) The chemical formula of the lattice is AlN.
(B) The $N^{3-}$ anions form a hexagonal close-packed sublattice.
(C) Each $N^{3-}$ is tetrahedrally coordinated by four $Al^{3+}$ cations.
(D) Each $Al^{3+}$ is octahedrally coordinated by six $N^{3-}$ anions.
(E) $Al^{3+}$ cations occupy exactly half of the tetrahedral voids in the $N^{3-}$ hcp lattice.
(F) The conventional hexagonal unit cell contains four AlN formula units.
(G) The anion stacking sequence along the c-axis is ABAB.

**Answer**:
A, B, C, E, F, G

## A.5.2 Crystal Group Identification (M002)

The Crystal Group Identification subtask aims to rigorously test the model's understanding of symmetry in crystallographic contexts. Given a lattice diagram of a material, the model is prompted to directly infer the most likely space group symbol representative of the structure's symmetry and lattice parameters, such as $Pn\bar{3}m1$, without supplementary justification. This task is minimal and unambiguous, focusing the model's attention on essential visual cues within the diagram. By constraining the output to a standard space group notation, we ensure objective evaluation, mirroring practical challenges faced by experts in structural materials analysis.

### Crystal Group Identification (M002)

**Images**:

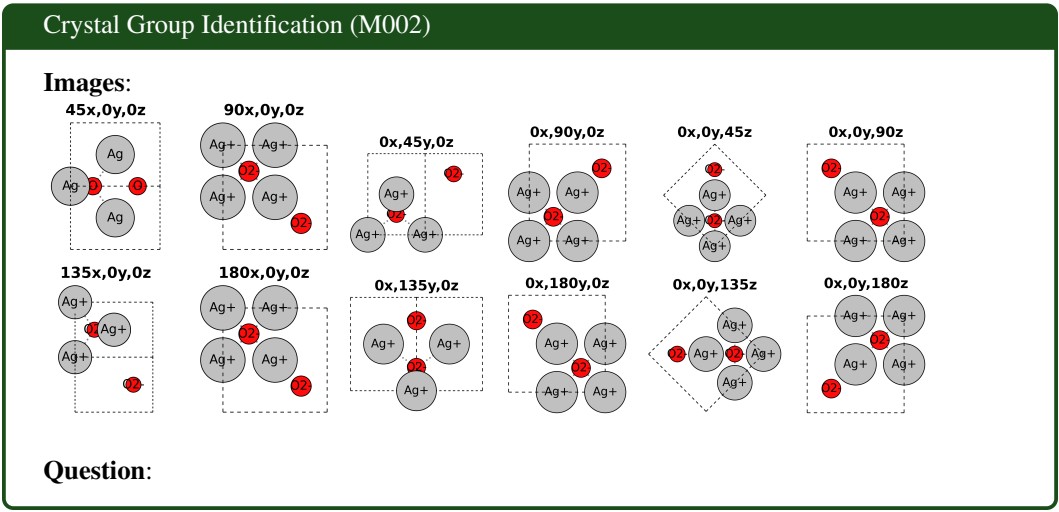

**Question**:

Given different views of the material, determine the space-group symbol that best matches the symmetry and lattice features shown, and reply only with that symbol (e.g., $Fm\bar{3}m$), without any additional text, explanation, or commentary.
Based on the different views of a material: front view <image>, side view <image> and top view <image>, determine the most likely space group of the material.

**Answer**:
$Pn\bar{3}m1$

### A.5.3  Crystal Formula Determination (M003)

Crystal Formula Determination introduces a focused subtask designed to assess the model's capability to infer chemical composition from visual lattice representations. Given a lattice-diagram of a crystalline material, the model is required to extract atomic identities, count stoichiometric ratios, and succinctly output the correct molecular formula (e.g., $Al_3Ir$), without any extraneous explanation or content. This task evaluates whether models possess the fine-grained visual reasoning and domain-specific knowledge necessary to interpret crystal structures in a manner analogous to trained materials scientists. The subtask thus provides a controlled benchmark for assessing structural understanding and symbolic reasoning within the materials science domain.

---

**Crystal Formula Determination (M003)**

**Images**:

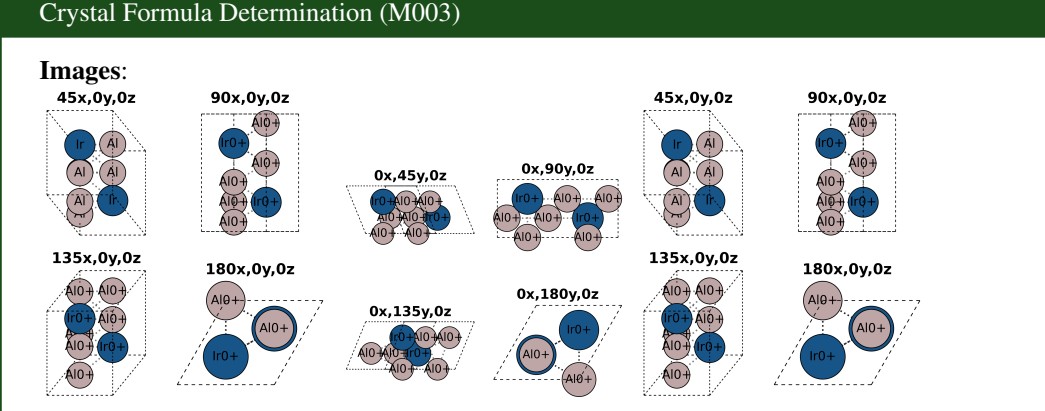

**Question**:
Given different views of the material, infer the molecular formula based solely on the lattice composition and reply only with that formula (e.g., $Ag_2O$), without any additional text, explanation, or commentary.
Based on the different views of a material: front view <image>, side view <image> and top view <image>, determine the molecular formula of the material.

**Answer**:
$Al_3Ir$

---

### A.5.4  Elemental Valence State Prediction (M004)

The Elemental Valence State Prediction subtask focuses on inferring the oxidation states of constituent elements in crystalline materials solely from their lattice diagrams. Given an image representing the atomic arrangement, the model must accurately identify and output the oxidation state for each element present. The annotation is strictly formatted as 'Element: Oxidation state' pairs, separated by commas, without additional explanation. This task evaluates the model's ability to interpret crystallographic structures and deduce fundamental chemical properties, mirroring the visual reasoning capabilities essential for automated scientific discovery in materials science.

## Elemental Valence State Prediction (M004)

**Images**:

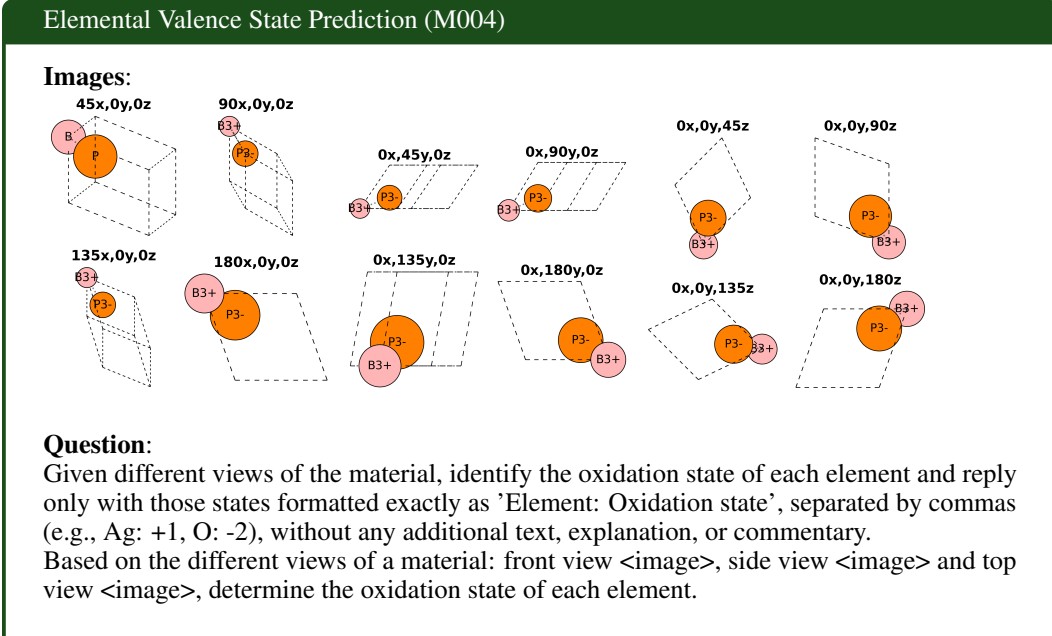

**Question**:
Given different views of the material, identify the oxidation state of each element and reply only with those states formatted exactly as 'Element: Oxidation state', separated by commas (e.g., Ag: +1, O: -2), without any additional text, explanation, or commentary.
Based on the different views of a material: front view <image>, side view <image> and top view <image>, determine the oxidation state of each element.

**Answer**:
B: +3, P: -3

### A.5.5 Stability Estimation (M006)

We introduce the Stability Estimation subtask, which aims to probe a model's capacity for extracting and reasoning over thermodynamic phase diagrams. Given a compositional phase diagram and multiple-choice options, the model must identify the composition ratio corresponding to the most thermodynamically stable phase, formulating its response as a single letter without supplementary text. This formulation encourages precise reading comprehension and domain-specific reasoning by constraining output format. The task is critical in materials science applications, where correct phase identification underpins the rational design of compounds. Our setup rigorously tests both image interpretation and scientific decision-making abilities.

## Stability Estimation (M006)

**Images**:

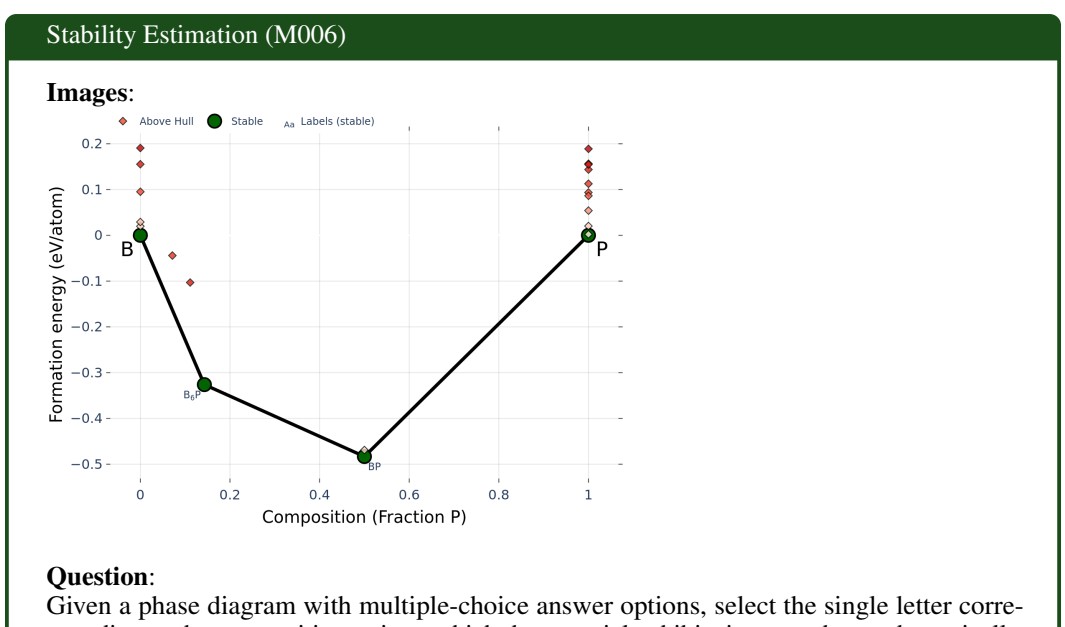

**Question**:
Given a phase diagram with multiple-choice answer options, select the single letter corresponding to the composition ratio at which the material exhibits its most thermodynamically

stable phase and reply only with that letter (e.g., C), without any additional text, explanation, or commentary.
Based on the given phase diagram <image>, at what composition ratio does the material exhibit its most thermodynamically stable phase.

**Options**:
(A) B (P fraction = 0.00)
(B) $B_6P$ (P fraction $\approx$ 0.14)
(C) $B_3P$ (P fraction = 0.25)
(D) BP (P fraction = 0.50)
(E) P (P fraction = 1.00)
**Answer**:
D

## A.5.6 Energy Band and DOS Interpretation (M008)

This subtask focuses on interpreting electronic band structures and density of states (DOS) plots to deduce critical material properties. Given graphical data for specific materials, participants are required to ascertain whether the system exhibits metallic or semiconducting behavior, based on the presence or absence of a band gap at the Fermi level. Further, they must extract additional insights, such as identifying atomic contributions to states near the Fermi level or noting significant features in the band topology. The task demands precise visual analysis and concise summarization, resembling the interpretive rigor employed in modern condensed matter research.

### Energy Band and DOS Interpretation (M008)

**Images**:

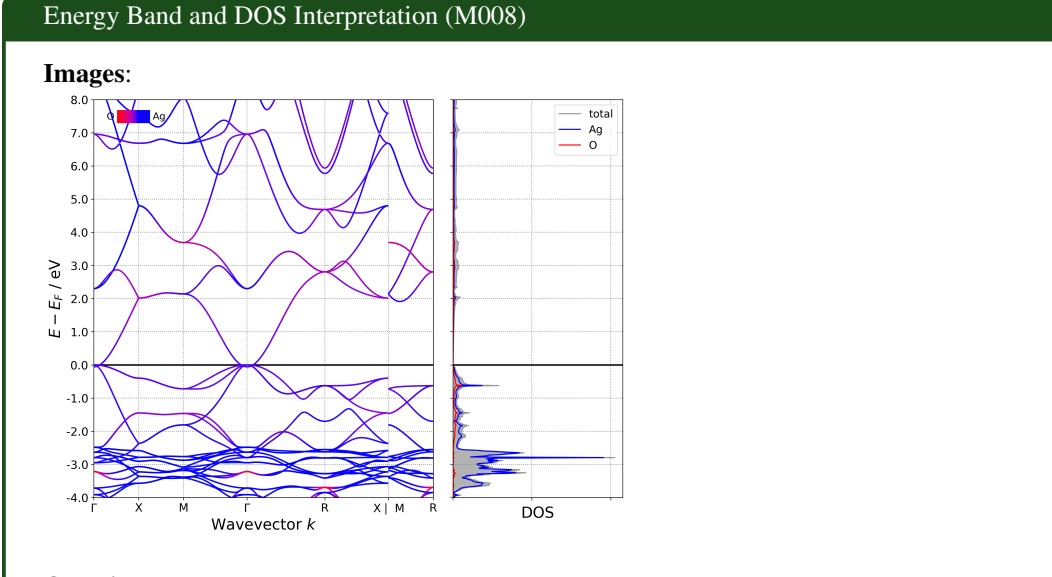

**Question**:
Based on the given band structure and DOS <image>, is the material metallic or semiconducting? What could be concluded more from the graph?

**Answer**:
The material is metallic (bands cross the Fermi level). Electronic states near the Fermi level dominated by Ag atoms.

## A.5.7 Band Gap Classification (M010)

We formulate the Band Gap Classification subtask to determine the electronic nature of a material from its computed band structure. Given a plot of energy versus k-point, the objective is to analyze the positions of the valence band maximum (VBM) and conduction band minimum (CBM). The core decision is whether the bandgap is direct, with VBM and CBM at the same high-symmetry

k-point, indirect, with these features at different k-points, or absent, indicating a metallic state. This classification provides insight into the fundamental optoelectronic properties of the material, facilitating materials discovery and characterization.

---

### Band Gap Classification (M010)

**Images**:

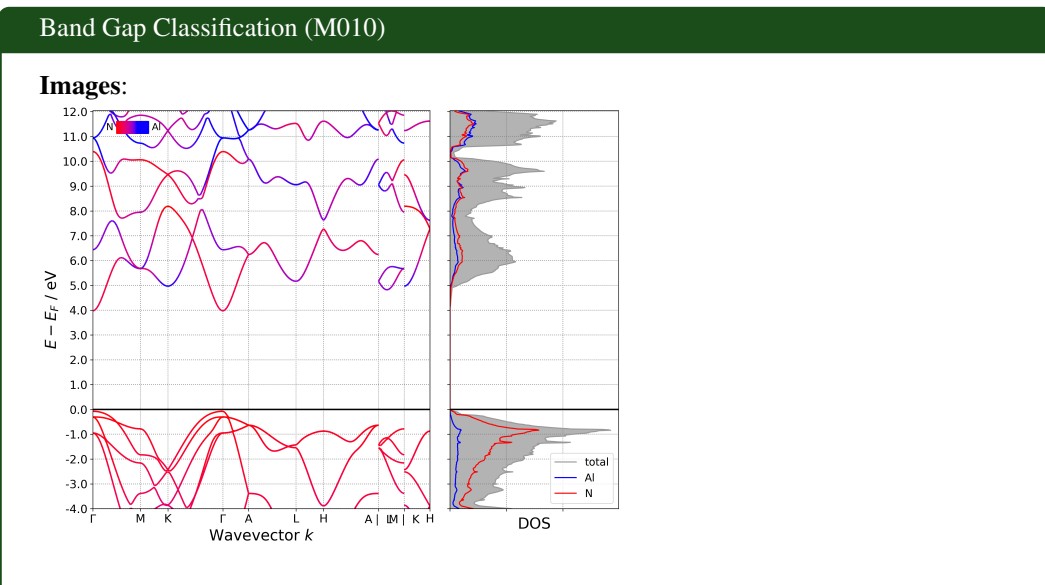

**Question**:

Given an electronic band structure plot, analyze the positions of the valence band maximum (VBM) and conduction band minimum (CBM) to decide if the material has a direct bandgap (VBM and CBM at the same high-symmetry k-point, which you must specify), an indirect bandgap (VBM and CBM at different high-symmetry k-points, which you must specify both), or if band crossings at the Fermi level indicate a metallic state with no bandgap. Reply with a single concise narrative paragraph stating your conclusion, including the relevant high-symmetry k-points for direct or indirect cases, or for a metallic case briefly explaining that band crossings preclude any bandgap, without bullet points, lists, or additional commentary.

Based on the provided band structure <image>, determine whether the material exhibits a direct or indirect bandgap.

**Answer**:

VBM and CBM are located at same k-points, the material exhibits direct bandgap

---

### A.5.8 Valence State and Electronic Orbital Analysis (M018)

This subtask focuses on rigorous interpretation of XPS spectra, requiring the analyst to accurately resolve and assign elemental signatures, oxidation states, and electronic environments present in complex material samples. The task demands a close examination of core-level binding energies, consideration of possible satellite peaks, and differentiation between multiple chemical states or phases when coexisting species are detected. By systematically associating observed peaks with established reference values, participants are expected to clearly articulate the composition and chemical identity of the sample, ensuring precise and unambiguous material characterization consistent with high standards of scientific reporting.

---

### Valence State and Electronic Orbital Analysis (M018)

**Images**:

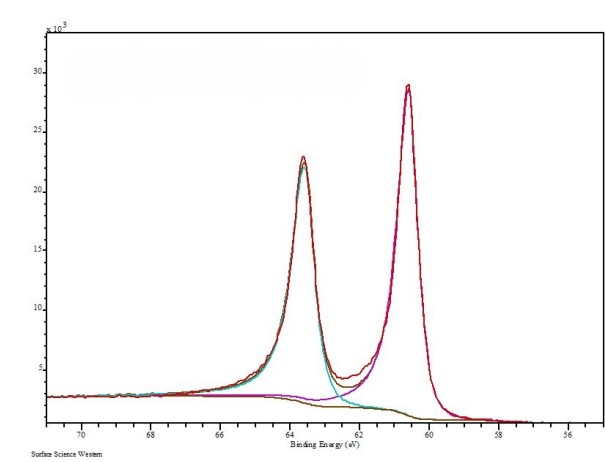

**Question**:
Given the XPS spectrum of the sample, analyze the spectral features to determine which elements and compounds are present and assign their core-level peaks (including any satellite or oxidation-state-specific signatures), and reply with a single descriptive paragraph that mirrors the example style—no bullet points, lists, or additional commentary.
Based on the provided XPS spectra <image>, please specify the element species and identify the characteristic peaks. If multiple materials are present, analyze and assign the peaks for each materials.

**Answer**:
The depicted XPS spectrum showed Ir $4f_{7/2}$ and $4f_{5/2}$ peaks, with a binding energy of 60.59 eV and 63.59 eV respectively.

### A.5.9 Phase Identification (M020)

In this subtask, we construct a phase identification scenario that simulates a real-world X-ray diffraction (XRD) analysis workflow. The participant is presented with the XRD pattern of a composite material alongside the reference XRD patterns for a set of candidate phases. The objective is to determine, through pattern comparison, the exact combination of phases present in the mixture. This task not only assesses the ability to recognize subtle similarities in peak positions and intensities but also emphasizes robust reasoning under ambiguity as occurs in practical material characterization, which is critical for developing automated, generalizable scientific AI models.

Phase Identification (M020)

**Images**:

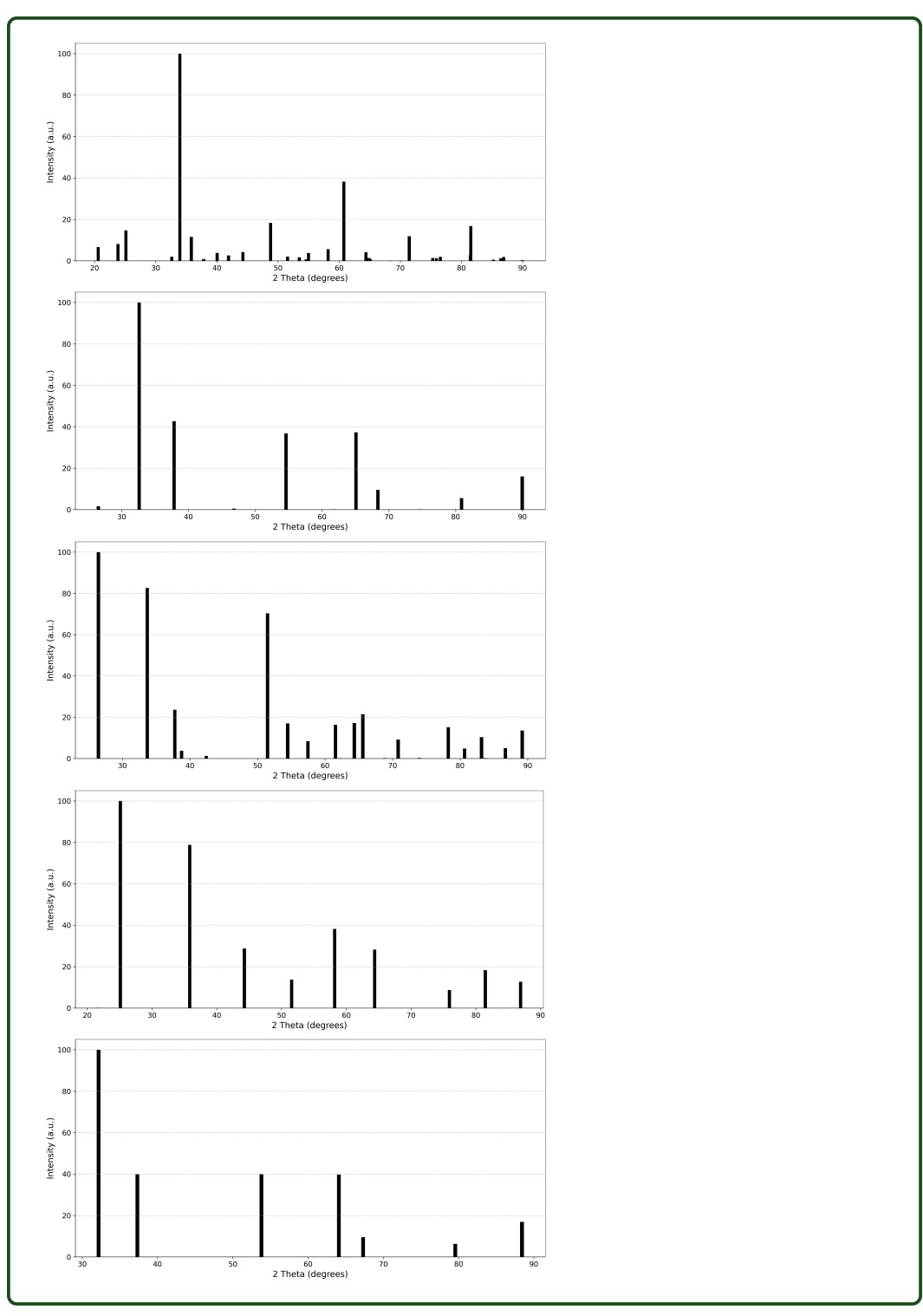

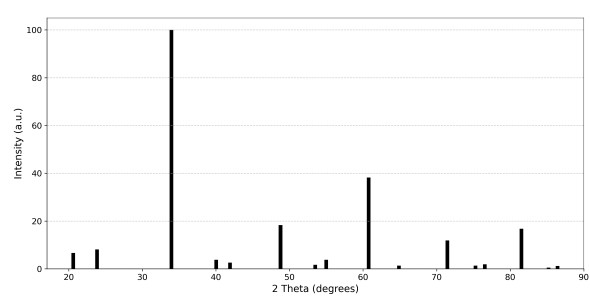

**Question**:
Given a combined XRD pattern and several materials' individual XRD patterns, analyze the set of diffraction peaks to determine which crystalline phases are present and provide a concise narrative explanation of your reasoning in a single paragraph, avoiding bullet points, lists, or any additional commentary.
Based on the XRD pattern of the composite material made from a mixture of three materials <image> and five XRD patterns of $Ag_2O$ <image>, $SnO_2$ <image>, $BaTe$ <image>, $BAs$ <image> and $Ac_2CuIr$ <image> (orderly), identify all three phases/materials present in the composite material.

**Answer**:
This graph contains three distinct materials: $Ag_2O$, $BaTe$, and $Ac_2CuIr$.

## A.5.10 Lattice Constant Estimation (M021)

In this subtask, we focus on the extraction of fundamental crystallographic information from experimental X-ray diffraction (XRD) patterns. The objective is to interpret a given XRD spectrum by first identifying the material's crystalline phase and then systematically assigning Miller indices (hkl) to the primary diffraction peaks observed. Building on the assigned peaks, precise lattice parameters are estimated through established diffraction geometry. By requiring concise, integrated responses, this subtask evaluates the ability to connect peak analysis with physical structure, encouraging rigorous yet efficient crystallographic reasoning akin to practices in real-world materials research.

---

**Lattice Constant Estimation (M021)**

**Images**:

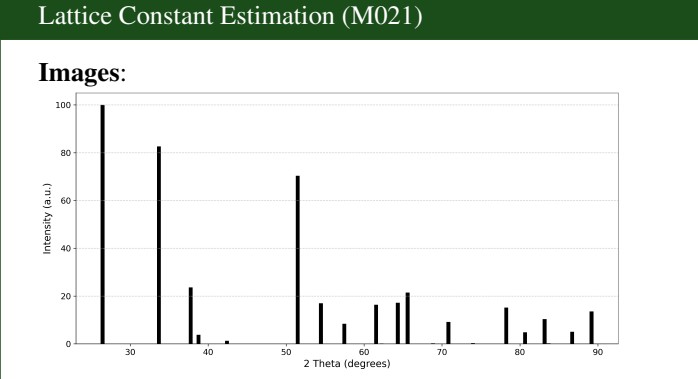

**Question**:
Given an XRD pattern of a single-phase material, identify the crystalline phase, assign the obvious primary diffraction peaks to their Miller indices (hkl), calculate the lattice parameters from those peak positions, and reply with a single concise narrative paragraph integrating these findings and calculations, without bullet points, lists, or additional commentary.
Based on the provided XRD pattern <image>, please identify the crystalline phase(s) present, assign Miller indices (hkl) to the distinct primary diffraction peaks (note that, you only need to specify the obvious peaks), and use those peak positions to estimate the lattice parameters.

---

> **Answer**:
> The main diffraction peaks at $2\theta = 26.5°$, $33.7°$, $38.8°$, $51.5°$, $54.5°$, $62.2°$, $65.6°$, $70.8°$, $78.2°$ and $89.2°$ correspond exactly to the (110), (101), (111), (211), (220), (221), (301), (202), (321) and (312) reflections of $SnO_2$ (Space group $P4_2/mnm$). No impurity phase is found in the XRD pattern. The calculated lattice parameters are: a = b = 4.76Å, c = 3.21Å, $\alpha = \beta = \gamma = 90°$.

### A.5.11 Crystal Grain Size Estimation (M022)

This subtask focuses on leveraging X-ray diffraction (XRD) patterns to quantitatively estimate the grain size of polycrystalline materials. By isolating the most intense diffraction peak and rigorously applying the Scherrer equation, the objective is to derive a reliable measurement of average crystallite dimensions. The setup emphasizes precision in extracting critical parameters such as peak position and full width at half maximum (FWHM), thus ensuring consistency and reproducibility. This procedure is essential in characterizing the microstructural evolution of materials, directly correlating XRD features with underlying nanoscale attributes, and ultimately bridging experimental observables with materials science fundamentals.

> ### Crystal Grain Size Estimation (M022)
>
> **Images**:
>
> 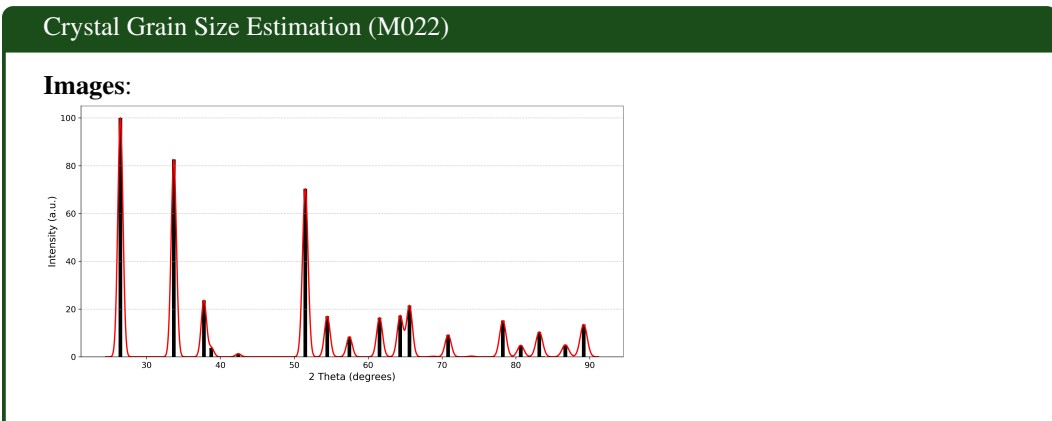
>
> **Question**:
> Given an XRD pattern of a material with a clearly defined most intense peak, apply the Scherrer equation to that peak to calculate the average crystallite size and reply with a single concise narrative paragraph stating the result and key parameters used (e.g., peak position, peak intensity), without bullet points, lists, or additional commentary.
> Based on the provided XRD pattern <image>, please estimate the average crystallite size of the sample by applying the Scherrer equation to the most intense XRD peak.
>
> **Answer**:
> Applying the Scherrer formula to the most intense (110) peak, the calculated average crystallite size is 0.1nm

## B  More Experimental Results

Please refer to Table 7, Table 8, Table 9, Table 10, Table 11, Table 12, Table 13, Table 14, Table 15, and Table 16 for additional experimental results.

## C  Larger Maximal Number of Generated Tokens

In the main paper, we set the maximal number of generated tokens to 1024 for all models to ensure fairness and cost-effectiveness. This results in poor performance for some reasoning MLLMs like the Gemini-2.5-Pro, which rapidly consumes all 1024 tokens for reasoning. To address this, we conducted an additional experiment with a larger maximal number of generated tokens (16384) for Gemini-2.0-Flash, Gemini-2.5-Flash, and Gemini-2.5-Pro in Table 17 and Table 18.

| Model | A001 | A003 | A006 | A007 | A008 | A009 | A010 | A011 |
|---|---|---|---|---|---|---|---|---|
| Claude-3-Opus | 39.00 | 20.00 | 20.50 | 24.00 | 6.50 | 0.50 | 5.50 | 2.00 |
| Claude-3.7-Sonnet | 70.50 | 26.11 | 28.00 | 9.00 | 18.50 | 1.00 | 0.00 | 54.00 |
| Doubao-1.5-vision-pro | 58.50 | 27.22 | 18.50 | 18.50 | 27.00 | 11.50 | 2.50 | 63.00 |
| Gemini-2.0-Flash | 57.00 | 41.11 | 12.50 | 0.00 | 9.00 | 3.00 | 0.00 | 9.00 |
| Gemini-2.5-Flash | 69.50 | 35.56 | 20.00 | 24.50 | 28.00 | 7.50 | 4.50 | 6.00 |
| Gemini-2.5-Pro | 39.50 | 0.00 | 0.00 | 0.00 | 0.00 | 0.00 | 0.00 | 1.00 |
| GPT-4o-2024-11-20 | 73.00 | 26.11 | 21.50 | 24.50 | 14.00 | 4.50 | 0.00 | 0.00 |
| GPT-4.1 | 81.00 | 30.00 | 23.00 | 22.50 | 18.50 | 14.50 | 0.00 | 4.50 |
| GPT-o1 | 67.50 | 35.00 | 6.00 | 13.50 | 8.50 | 3.00 | 3.50 | 40.50 |
| GPT-o3 | 72.00 | 45.00 | 24.00 | 13.50 | 7.50 | 16.00 | 0.00 | 18.00 |
| Grok-2-Vision-12-12 | 64.50 | 22.22 | 5.50 | 24.50 | 24.00 | 3.50 | 0.00 | 11.00 |
| InternVL-3-78B | 48.00 | 27.22 | 29.00 | 24.50 | 19.50 | 9.00 | 10.00 | 49.50 |
| InternVL-2.5-78B | 41.50 | 16.11 | 15.50 | 17.00 | 20.50 | 9.00 | 2.50 | 4.00 |
| Llama-3.2-Vision-90B | 61.00 | 42.22 | 26.00 | 25.50 | 2.50 | 10.00 | 0.00 | 0.00 |
| Llava-OneVision-72B | 54.00 | 20.00 | 0.00 | 17.00 | 7.00 | 15.00 | 11.50 | 76.50 |
| QwenVL-72B | 63.00 | 20.00 | 49.00 | 26.50 | 24.50 | 10.00 | 0.00 | 18.00 |

Table 7: Performance of different models on the Astronomy tasks (English).

| Model | A001 | A003 | A006 | A007 | A008 | A009 | A010 | A011 |
|---|---|---|---|---|---|---|---|---|
| Claude-3-Opus | 40.00 | 32.22 | 10.50 | 16.50 | 24.00 | 0.50 | 2.00 | 4.50 |
| Claude-3.7-Sonnet | 72.00 | 30.00 | 12.00 | 17.00 | 17.50 | 1.00 | 0.00 | 30.00 |
| Doubao-1.5-vision-pro | 54.00 | 21.11 | 7.00 | 15.50 | 19.50 | 7.50 | 4.00 | 63.00 |
| Gemini-2.0-Flash | 60.50 | 34.44 | 1.00 | 0.00 | 1.00 | 0.00 | 0.00 | 7.50 |
| Gemini-2.5-Flash | 70.00 | 33.33 | 22.00 | 23.00 | 29.50 | 5.50 | 3.50 | 7.00 |
| Gemini-2.5-Pro | 42.50 | 5.00 | 0.00 | 0.00 | 0.00 | 0.00 | 0.00 | 1.00 |
| GPT-4o-2024-11-20 | 73.00 | 25.00 | 17.00 | 16.50 | 10.50 | 3.00 | 0.00 | 0.00 |
| GPT-4.1 | 81.00 | 31.11 | 25.00 | 21.00 | 12.50 | 16.00 | 0.00 | 18.00 |
| GPT-o1 | 67.50 | 35.00 | 6.50 | 16.50 | 3.00 | 4.00 | 3.00 | 40.50 |
| GPT-o3 | 76.50 | 40.00 | 21.50 | 16.00 | 11.00 | 4.50 | 0.00 | 22.50 |
| Grok-2-Vision-12-12 | 60.00 | 21.67 | 10.50 | 21.00 | 23.50 | 5.00 | 0.00 | 4.50 |
| InternVL-3-78B | 59.50 | 41.11 | 13.50 | 23.00 | 13.00 | 3.00 | 5.00 | 40.50 |
| InternVL-2.5-78B | 49.00 | 37.22 | 11.50 | 14.50 | 21.50 | 5.00 | 0.00 | 0.00 |
| Llama-3.2-Vision-90B | 62.50 | 22.22 | 12.00 | 15.00 | 16.50 | 6.50 | 0.00 | 10.00 |
| Llava-OneVision-72B | 49.50 | 10.00 | 0.00 | 14.00 | 25.00 | 8.50 | 5.50 | 72.00 |
| QwenVL-72B | 58.50 | 25.00 | 39.50 | 25.00 | 9.50 | 5.50 | 0.00 | 4.50 |

Table 8: Performance of different models on the Astronomy tasks (Chinese).

As it can be seen, with increasing the maximal number of generated tokens, the performance of Gemini-2.5-Flash improves marginally, while Gemini-2.5-Pro shows a significant performance boost across all tasks, especially in the Earth and Life tasks. This indicates that Gemini-2.5-Pro is capable of solving more complex tasks with sufficient reasoning steps, but it requires a larger number of tokens to do so.

## D   Additional Evaluation Methods

For better reliability, we conduct two additional evaluations as follows:

**Human evaluation.** First, we randomly sample 10 questions from SFE (due to limited time) and collect the corresponding responses of six MLLMs (three close weight: GPT-o3, Claude-3.7-Sonnet, Grok-2-Vision-12-12; three open weight: InternVL-3-78B, Qwen2.5-VL-72B, Llava-OneVision-72B). Then, these responses are anonymized and provided to domain experts for ranking. The resulting aggregated order is:

| Model | C001 | C003 | C004 | C005 | C006 | C007 | C009 | C010 | C012 | C013 |
|---|---|---|---|---|---|---|---|---|---|---|
| Claude-3-Opus | 24.67 | 0.00 | 2.00 | 18.67 | 52.67 | 17.08 | 0.00 | 48.00 | 8.00 | 31.33 |
| Claude-3.7-Sonnet | 52.00 | 18.67 | 38.67 | 52.00 | 20.00 | 22.08 | 8.67 | 86.00 | 47.33 | 32.00 |
| Doubao-1.5-vision-pro | 43.33 | 2.67 | 18.00 | 32.00 | 12.67 | 10.83 | 2.00 | 66.00 | 44.00 | 26.00 |
| Gemini-2.0-Flash | 62.00 | 15.33 | 28.67 | 39.33 | 14.00 | 23.33 | 0.00 | 81.33 | 46.67 | 28.00 |
| Gemini-2.5-Flash | 64.67 | 6.67 | 19.33 | 48.00 | 10.00 | 8.33 | 1.33 | 82.00 | 64.00 | 14.67 |
| Gemini-2.5-Pro | 1.33 | 0.00 | 8.67 | 0.00 | 0.00 | 3.75 | 0.00 | 0.00 | 18.00 | 0.00 |
| GPT-4o-2024-11-20 | 48.00 | 0.00 | 23.33 | 36.67 | 0.00 | 21.25 | 0.00 | 86.00 | 29.33 | 20.67 |
| GPT-4.1 | 50.00 | 1.33 | 16.67 | 42.00 | 7.33 | 4.17 | 6.00 | 86.00 | 38.00 | 30.67 |
| GPT-o1 | 66.67 | 4.00 | 30.00 | 48.67 | 0.00 | 13.33 | 6.00 | 78.00 | 40.67 | 40.00 |
| GPT-o3 | 56.67 | 8.00 | 32.67 | 42.00 | 8.67 | 19.58 | 3.33 | 66.00 | 38.00 | 30.67 |
| Grok-2-Vision-12-12 | 36.67 | 1.33 | 27.33 | 30.00 | 5.33 | 16.67 | 0.00 | 68.67 | 12.00 | 23.33 |
| InternVL-3-78B | 34.67 | 0.00 | 12.00 | 17.33 | 16.00 | 5.42 | 0.00 | 79.33 | 18.67 | 26.00 |
| InternVL-2.5-78B | 34.00 | 0.00 | 16.67 | 23.33 | 0.00 | 4.17 | 0.00 | 71.33 | 13.33 | 26.67 |
| Llama-3.2-Vision-90B | 39.33 | 0.00 | 16.00 | 20.67 | 4.67 | 0.00 | 0.00 | 69.33 | 3.33 | 12.00 |
| Llava-OneVision-72B | 33.33 | 0.00 | 16.67 | 20.00 | 2.67 | 2.50 | 0.00 | 66.00 | 14.00 | 8.67 |
| QwenVL-72B | 18.67 | 4.00 | 22.67 | 25.33 | 2.00 | 10.83 | 0.00 | 48.00 | 34.67 | 18.00 |

| Model | C018 | C023 | C024 | C025 | C026 | C027 | C028 | C029 | C030 |
|---|---|---|---|---|---|---|---|---|---|
| Claude-3-Opus | 26.00 | 17.33 | 18.00 | 31.33 | 16.00 | 33.33 | 0.00 | 6.00 | 24.00 |
| Claude-3.7-Sonnet | 2.00 | 18.00 | 35.33 | 17.33 | 19.33 | 29.33 | 0.00 | 0.00 | 32.67 |
| Doubao-1.5-vision-pro | 32.67 | 19.33 | 31.33 | 51.33 | 21.33 | 22.67 | 11.33 | 8.67 | 27.33 |
| Gemini-2.0-Flash | 22.00 | 18.67 | 18.67 | 48.67 | 4.67 | 24.00 | 4.67 | 1.33 | 50.00 |
| Gemini-2.5-Flash | 0.00 | 14.67 | 37.33 | 49.33 | 13.33 | 14.67 | 8.00 | 2.67 | 0.00 |
| Gemini-2.5-Pro | 0.00 | 0.00 | 0.00 | 0.00 | 0.00 | 5.33 | 0.00 | 1.33 | 0.00 |
| GPT-4o-2024-11-20 | 12.00 | 15.33 | 28.67 | 42.67 | 8.00 | 24.00 | 3.33 | 7.33 | 4.00 |
| GPT-4.1 | 26.00 | 20.67 | 30.00 | 48.67 | 23.33 | 24.67 | 2.67 | 1.33 | 8.67 |
| GPT-o1 | 31.33 | 19.33 | 32.67 | 55.33 | 18.00 | 20.00 | 6.00 | 8.00 | 11.33 |
| GPT-o3 | 62.00 | 16.67 | 31.33 | 52.67 | 14.00 | 26.00 | 8.67 | 7.33 | 30.67 |
| Grok-2-Vision-12-12 | 26.00 | 19.33 | 22.00 | 48.67 | 14.00 | 23.33 | 2.67 | 4.67 | 25.33 |
| InternVL-3-78B | 24.00 | 19.33 | 22.00 | 47.33 | 5.33 | 21.33 | 0.67 | 15.33 | 20.00 |
| InternVL-2.5-78B | 12.67 | 19.33 | 37.33 | 51.33 | 4.00 | 20.67 | 0.67 | 0.00 | 25.33 |
| Llama-3.2-Vision-90B | 27.33 | 13.33 | 22.67 | 40.67 | 20.00 | 21.33 | 0.00 | 1.33 | 20.00 |
| Llava-OneVision-72B | 18.00 | 18.67 | 32.00 | 15.33 | 10.67 | 17.33 | 0.00 | 3.33 | 20.00 |
| QwenVL-72B | 20.67 | 19.33 | 24.00 | 56.00 | 3.33 | 25.33 | 2.00 | 5.33 | 12.67 |

Table 9: Performance of different models on the Chemistry tasks (English).

GPT-o3 ≈ Claude-3.7-Sonnet ≈ InternVL-3-78B > Grok-2-Vision-12-12 > Llava-OneVision-72B > Qwen2.5-VL-72B

**Multiple LLM judges.** We re-scored all responses of the above mentioned 6 MLLMs with 3 additional judges: Claude-3.7-Sonnet, Deepseek-V3, and Grok-2. The results are demonstrated in Table 19.

As it can be seen, all judges produce similar ranking compared to human evaluation. These results show that our conclusions are robust to assessments of both human and multiple LLM judges.

| Model | C001 | C003 | C004 | C005 | C006 | C007 | C009 | C010 | C012 | C013 |
|---|---|---|---|---|---|---|---|---|---|---|
| Claude-3-Opus | 15.33 | 0.00 | 20.67 | 16.67 | 38.67 | 0.00 | 1.33 | 45.33 | 4.00 | 28.00 |
| Claude-3.7-Sonnet | 57.33 | 16.00 | 27.33 | 53.33 | 14.67 | 13.75 | 8.00 | 86.00 | 40.67 | 28.67 |
| Doubao-1.5-vision-pro | 51.33 | 1.33 | 19.33 | 32.00 | 19.33 | 6.67 | 0.00 | 66.00 | 38.00 | 28.67 |
| Gemini-2.0-Flash | 42.00 | 10.67 | 15.33 | 30.00 | 26.67 | 10.83 | 0.00 | 76.00 | 51.33 | 32.67 |
| Gemini-2.5-Flash | 65.33 | 6.67 | 20.00 | 47.33 | 14.00 | 8.33 | 4.00 | 82.00 | 64.00 | 12.67 |
| Gemini-2.5-Pro | 0.00 | 0.00 | 10.67 | 0.00 | 0.00 | 3.75 | 0.00 | 6.00 | 18.00 | 2.00 |
| GPT-4o-2024-11-20 | 46.00 | 0.00 | 26.67 | 33.33 | 0.00 | 12.08 | 2.00 | 85.33 | 22.67 | 14.00 |
| GPT-4.1 | 46.67 | 0.00 | 23.33 | 36.67 | 10.67 | 2.50 | 1.33 | 86.00 | 24.67 | 32.00 |
| GPT-o1 | 62.00 | 3.33 | 33.33 | 50.67 | 6.00 | 18.33 | 0.00 | 66.00 | 38.67 | 46.00 |
| GPT-o3 | 54.67 | 10.67 | 32.00 | 35.33 | 4.67 | 10.42 | 2.67 | 60.00 | 36.00 | 38.67 |
| Grok-2-Vision-12-12 | 26.67 | 2.00 | 20.00 | 36.00 | 6.67 | 2.92 | 0.00 | 80.67 | 10.00 | 24.67 |
| InternVL-3-78B | 24.67 | 0.00 | 17.33 | 12.00 | 5.33 | 1.67 | 0.00 | 74.00 | 16.67 | 23.33 |
| InternVL-2.5-78B | 26.00 | 0.00 | 12.67 | 12.00 | 0.00 | 0.83 | 0.00 | 56.67 | 10.67 | 10.67 |
| Llama-3.2-Vision-90B | 22.67 | 0.00 | 10.67 | 18.00 | 5.33 | 0.00 | 0.00 | 64.67 | 0.00 | 8.67 |
| Llava-OneVision-72B | 19.33 | 0.00 | 16.00 | 18.00 | 0.00 | 1.67 | 0.00 | 78.00 | 10.00 | 8.67 |
| QwenVL-72B | 10.67 | 3.33 | 22.00 | 26.67 | 2.67 | 0.83 | 0.00 | 30.00 | 26.67 | 26.67 |

| Model | C018 | C023 | C024 | C025 | C026 | C027 | C028 | C029 | C030 |
|---|---|---|---|---|---|---|---|---|---|
| Claude-3-Opus | 23.33 | 15.33 | 17.33 | 32.67 | 18.00 | 29.33 | 0.00 | 8.67 | 27.33 |
| Claude-3.7-Sonnet | 2.67 | 16.67 | 25.33 | 23.33 | 8.00 | 28.00 | 0.00 | 4.67 | 30.00 |
| Doubao-1.5-vision-pro | 26.00 | 18.00 | 25.33 | 50.00 | 19.33 | 24.00 | 14.67 | 6.67 | 28.67 |
| Gemini-2.0-Flash | 12.00 | 20.00 | 15.33 | 46.00 | 8.67 | 23.33 | 6.67 | 1.33 | 48.67 |
| Gemini-2.5-Flash | 0.00 | 14.00 | 36.67 | 46.67 | 12.00 | 11.33 | 7.33 | 2.67 | 0.00 |
| Gemini-2.5-Pro | 0.00 | 0.00 | 0.00 | 0.00 | 0.00 | 2.00 | 0.00 | 0.00 | 0.00 |
| GPT-4o-2024-11-20 | 19.33 | 19.33 | 23.33 | 38.00 | 14.67 | 9.33 | 1.33 | 0.00 | 2.67 |
| GPT-4.1 | 28.00 | 18.00 | 26.67 | 32.00 | 15.33 | 24.00 | 2.67 | 3.33 | 18.00 |
| GPT-o1 | 34.00 | 14.67 | 26.00 | 54.67 | 18.67 | 24.00 | 5.33 | 4.67 | 16.00 |
| GPT-o3 | 52.00 | 17.33 | 22.00 | 54.00 | 27.33 | 25.33 | 8.00 | 9.33 | 40.00 |
| Grok-2-Vision-12-12 | 18.67 | 19.33 | 18.67 | 48.00 | 16.67 | 23.33 | 8.67 | 4.00 | 27.33 |
| InternVL-3-78B | 15.33 | 18.67 | 20.00 | 44.00 | 6.00 | 5.33 | 0.00 | 15.33 | 18.67 |
| InternVL-2.5-78B | 16.67 | 18.67 | 17.33 | 48.00 | 6.00 | 24.67 | 0.00 | 10.67 | 33.33 |
| Llama-3.2-Vision-90B | 21.33 | 19.33 | 22.67 | 37.33 | 16.67 | 25.33 | 0.00 | 4.00 | 20.00 |
| Llava-OneVision-72B | 16.00 | 20.00 | 20.00 | 39.33 | 4.67 | 22.67 | 0.00 | 2.00 | 20.00 |
| QwenVL-72B | 19.33 | 20.00 | 17.33 | 48.67 | 6.67 | 19.33 | 0.00 | 2.67 | 15.33 |

Table 10: Performance of different models on the Chemistry tasks (English).

| Model | E001 | E004 | E005 | E006 | E007 | E008 | E014 | E016 | E017 | E018 | E021 |
|---|---|---|---|---|---|---|---|---|---|---|---|
| Claude-3-Opus | 24.00 | 16.67 | 55.00 | 42.50 | 18.00 | 90.00 | 31.11 | 32.00 | 41.00 | 28.18 | 35.00 |
| Claude-3.7-Sonnet | 23.33 | 23.33 | 57.50 | 50.83 | 23.00 | 16.00 | 30.00 | 42.00 | 72.00 | 52.73 | 36.50 |
| Doubao-1.5-vision-pro | 24.00 | 20.00 | 29.17 | 29.17 | 2.00 | 38.00 | 68.89 | 14.00 | 60.00 | 23.64 | 15.75 |
| Gemini-2.0-Flash | 24.00 | 25.83 | 30.83 | 55.00 | 20.00 | 46.00 | 14.44 | 26.00 | 67.00 | 28.18 | 36.00 |
| Gemini-2.5-Flash | 56.67 | 18.33 | 27.50 | 27.50 | 10.00 | 28.00 | 14.44 | 33.00 | 48.00 | 23.64 | 38.00 |
| Gemini-2.5-Pro | 0.00 | 0.00 | 9.17 | 1.67 | 0.00 | 0.00 | 0.00 | 0.00 | 0.00 | 11.82 | 3.00 |
| GPT-4o-2024-11-20 | 18.00 | 20.00 | 28.33 | 50.83 | 28.00 | 42.00 | 28.89 | 50.00 | 65.00 | 30.00 | 25.75 |
| GPT-4.1 | 30.00 | 26.67 | 63.33 | 35.83 | 25.00 | 47.00 | 72.22 | 60.00 | 69.00 | 51.82 | 22.75 |
| GPT-o1 | 18.00 | 34.17 | 54.17 | 45.83 | 22.00 | 63.00 | 36.67 | 53.00 | 44.00 | 35.45 | 35.25 |
| GPT-o3 | 36.00 | 30.00 | 41.67 | 40.83 | 13.00 | 54.00 | 76.67 | 67.00 | 74.00 | 44.55 | 33.75 |
| Grok-2-Vision-12-12 | 46.00 | 13.33 | 51.67 | 31.67 | 17.00 | 21.00 | 23.33 | 57.00 | 72.00 | 46.36 | 20.75 |
| InternVL-3-78B | 0.00 | 18.33 | 56.67 | 41.67 | 18.00 | 46.00 | 48.89 | 15.00 | 23.00 | 50.00 | 23.50 |
| InternVL-2.5-78B | 0.00 | 14.17 | 43.33 | 45.00 | 20.00 | 44.00 | 62.22 | 64.00 | 63.00 | 30.00 | 30.50 |
| Llama-3.2-Vision-90B | 28.00 | 21.67 | 33.33 | 21.67 | 22.00 | 15.00 | 23.33 | 26.00 | 69.00 | 28.18 | 23.25 |
| Llava-OneVision-72B | 24.00 | 12.50 | 40.83 | 11.67 | 6.00 | 27.00 | 40.00 | 15.00 | 60.00 | 36.36 | 13.50 |
| QwenVL-72B | 12.00 | 15.83 | 25.00 | 50.83 | 3.00 | 63.00 | 18.89 | 20.00 | 63.00 | 19.09 | 15.75 |

Table 11: Performance of different models on the Earth tasks (English).

| Model | E001 | E004 | E005 | E006 | E007 | E008 | E014 | E016 | E017 | E018 | E021 |
|---|---|---|---|---|---|---|---|---|---|---|---|
| Claude-3-Opus | 24.00 | 26.67 | 38.33 | 32.50 | 7.00 | 56.00 | 42.22 | 19.00 | 38.00 | 30.00 | 35.25 |
| Claude-3.7-Sonnet | 23.33 | 23.33 | 58.33 | 32.50 | 19.00 | 15.00 | 62.22 | 30.00 | 75.00 | 43.64 | 36.25 |
| Doubao-1.5-vision-pro | 24.00 | 15.00 | 28.33 | 35.00 | 3.00 | 63.00 | 70.00 | 24.00 | 55.00 | 14.55 | 9.00 |
| Gemini-2.0-Flash | 25.33 | 24.17 | 27.50 | 53.33 | 18.00 | 27.00 | 41.11 | 24.00 | 48.00 | 20.00 | 39.25 |
| Gemini-2.5-Flash | 56.67 | 16.67 | 27.50 | 26.67 | 9.00 | 31.00 | 12.22 | 36.00 | 56.00 | 25.45 | 30.00 |
| Gemini-2.5-Pro | 0.00 | 3.33 | 14.17 | 1.67 | 0.00 | 0.00 | 5.56 | 0.00 | 0.00 | 19.09 | 2.25 |
| GPT-4o-2024-11-20 | 12.00 | 25.00 | 31.67 | 39.17 | 15.00 | 49.00 | 70.00 | 27.00 | 55.00 | 29.09 | 19.50 |
| GPT-4.1 | 24.00 | 27.50 | 60.00 | 50.00 | 7.00 | 74.00 | 72.22 | 30.00 | 68.00 | 54.55 | 41.50 |
| GPT-o1 | 30.00 | 32.50 | 54.17 | 40.83 | 3.00 | 63.00 | 70.00 | 48.00 | 56.00 | 33.64 | 21.00 |
| GPT-o3 | 18.00 | 40.00 | 35.83 | 36.67 | 8.00 | 54.00 | 80.00 | 24.00 | 74.00 | 33.64 | 29.25 |
| Grok-2-Vision-12-12 | 34.67 | 22.50 | 58.33 | 36.67 | 19.00 | 43.00 | 48.89 | 17.00 | 69.00 | 36.36 | 30.50 |
| InternVL-3-78B | 0.00 | 17.50 | 60.83 | 15.00 | 0.00 | 72.00 | 68.89 | 15.00 | 29.00 | 43.64 | 24.75 |
| InternVL-2.5-78B | 18.00 | 19.17 | 59.17 | 38.33 | 10.00 | 68.00 | 67.78 | 19.00 | 59.00 | 41.82 | 33.00 |
| Llama-3.2-Vision-90B | 6.00 | 21.67 | 45.83 | 44.17 | 20.00 | 60.00 | 56.67 | 26.00 | 38.00 | 27.27 | 23.25 |
| Llava-OneVision-72B | 24.00 | 19.17 | 46.67 | 9.17 | 8.00 | 63.00 | 61.11 | 15.00 | 57.00 | 30.00 | 15.75 |
| QwenVL-72B | 0.00 | 17.50 | 30.00 | 62.50 | 0.00 | 54.00 | 22.22 | 15.00 | 45.00 | 21.82 | 20.25 |

Table 12: Performance of different models on the Earth tasks (Chinese).

| Model | L001 | L003 | L005 | L006 | L007 | L008 | L009 |
|---|---|---|---|---|---|---|---|
| Claude-3-Opus | 11.0 | 0.0 | 9.0 | 0.0 | 18.0 | 67.5 | 37.5 |
| Claude-3.7-Sonnet | 63.0 | 0.0 | 0.0 | 3.0 | 40.0 | 75.0 | 22.5 |
| Doubao-1.5-vision-pro | 62.0 | 0.0 | 54.0 | 11.0 | 18.0 | 82.5 | 45.0 |
| Gemini-2.0-Flash | 61.0 | 0.0 | 9.0 | 1.0 | 29.0 | 82.5 | 60.0 |
| Gemini-2.5-Flash | 53.0 | 0.0 | 9.0 | 6.0 | 0.0 | 75.0 | 45.0 |
| Gemini-2.5-Pro | 54.0 | 0.0 | 0.0 | 2.0 | 0.0 | 84.17 | 30.0 |
| GPT-4o-2024-11-20 | 53.0 | 1.0 | 9.0 | 6.0 | 27.0 | 67.5 | 30.0 |
| GPT-4.1 | 71.0 | 0.0 | 18.0 | 4.0 | 38.0 | 82.5 | 37.5 |
| GPT-o1 | 76.0 | 0.0 | 9.0 | 4.0 | 27.0 | 82.5 | 30.0 |
| GPT-o3 | 74.0 | 0.0 | 18.0 | 13.0 | 18.0 | 82.5 | 37.5 |
| Grok-2-Vision-12-12 | 4.0 | 0.0 | 27.0 | 4.0 | 22.0 | 76.67 | 30.0 |
| InternVL-3-78B | 37.0 | 0.0 | 38.0 | 5.0 | 36.0 | 75.0 | 37.5 |
| InternVL-2.5-78B | 23.0 | 2.0 | 27.0 | 0.0 | 18.0 | 75.0 | 45.0 |
| Llama-3.2-Vision-90B | 6.0 | 0.0 | 36.0 | 9.0 | 0.0 | 82.5 | 45.0 |
| Llava-OneVision-72B | 24.0 | 0.0 | 27.0 | 9.0 | 18.0 | 75.0 | 45.0 |
| QwenVL-72B | 51.0 | 0.0 | 9.0 | 2.0 | 36.0 | 77.5 | 45.0 |

| Model | L010 | L014 | L015 | L016 | L017 | L019 | L020 |
|---|---|---|---|---|---|---|---|
| Claude-3-Opus | 0.0 | 42.0 | 27.0 | 18.0 | 1.33 | 56.67 | 24.0 |
| Claude-3.7-Sonnet | 0.0 | 66.0 | 27.0 | 27.0 | 10.0 | 77.5 | 27.0 |
| Doubao-1.5-vision-pro | 0.0 | 50.0 | 9.0 | 18.0 | 0.0 | 56.67 | 17.0 |
| Gemini-2.0-Flash | 0.0 | 63.0 | 27.0 | 45.0 | 0.0 | 55.0 | 25.0 |
| Gemini-2.5-Flash | 0.0 | 53.0 | 30.0 | 45.0 | 0.0 | 22.5 | 16.0 |
| Gemini-2.5-Pro | 0.0 | 4.0 | 9.0 | 18.0 | 0.0 | 62.5 | 3.0 |
| GPT-4o-2024-11-20 | 0.0 | 64.0 | 36.0 | 45.0 | 1.33 | 70.0 | 21.0 |
| GPT-4.1 | 0.0 | 70.0 | 27.0 | 44.0 | 1.33 | 78.33 | 22.0 |
| GPT-o1 | 0.0 | 72.0 | 27.0 | 44.0 | 9.33 | 72.5 | 25.0 |
| GPT-o3 | 1.67 | 74.0 | 18.0 | 36.0 | 4.0 | 69.17 | 28.0 |
| Grok-2-Vision-12-12 | 0.0 | 50.0 | 21.0 | 27.0 | 1.33 | 71.67 | 13.0 |
| InternVL-3-78B | 0.0 | 56.0 | 21.0 | 36.0 | 1.33 | 74.17 | 17.0 |
| InternVL-2.5-78B | 0.0 | 60.0 | 21.0 | 18.0 | 0.0 | 76.67 | 9.0 |
| Llama-3.2-Vision-90B | 0.0 | 48.0 | 21.0 | 18.0 | 0.0 | 64.17 | 19.0 |
| Llava-OneVision-72B | 0.0 | 18.0 | 18.0 | 18.0 | 1.33 | 72.5 | 11.0 |
| QwenVL-72B | 0.0 | 64.0 | 15.0 | 18.0 | 0.0 | 25.0 | 15.0 |

Table 13: Performance of different models on the Life tasks (English).

| Model | L001 | L003 | L005 | L006 | L007 | L008 | L009 |
|---|---|---|---|---|---|---|---|
| Claude-3-Opus | 29.0 | 0.0 | 9.0 | 0.0 | 18.0 | 60.0 | 30.0 |
| Claude-3.7-Sonnet | 57.0 | 0.0 | 0.0 | 2.0 | 36.0 | 77.5 | 24.2 |
| Doubao-1.5-vision-pro | 54.0 | 0.0 | 45.0 | 2.0 | 18.0 | 82.5 | 37.5 |
| Gemini-2.0-Flash | 51.0 | 2.0 | 0.0 | 0.0 | 27.0 | 90.0 | 60.0 |
| Gemini-2.5-Flash | 57.0 | 0.0 | 9.0 | 4.0 | 0.0 | 76.7 | 45.0 |
| Gemini-2.5-Pro | 77.0 | 0.0 | 0.0 | 5.0 | 0.0 | 82.5 | 45.0 |
| GPT-4o-2024-11-20 | 45.0 | 0.0 | 0.0 | 2.0 | 36.0 | 60.0 | 22.5 |
| GPT-4.1 | 68.0 | 0.0 | 16.0 | 10.0 | 41.0 | 85.0 | 30.0 |
| GPT-o1 | 60.0 | 0.0 | 0.0 | 5.0 | 36.0 | 82.5 | 45.0 |
| GPT-o3 | 60.0 | 0.0 | 18.0 | 6.0 | 9.0 | 90.0 | 45.0 |
| Grok-2-Vision-12-12 | 0.0 | 0.0 | 36.0 | 19.0 | 17.0 | 75.0 | 30.0 |
| InternVL-3-78B | 22.0 | 0.0 | 9.0 | 0.0 | 27.0 | 75.0 | 30.0 |
| InternVL-2.5-78B | 7.0 | 0.0 | 27.0 | 24.0 | 9.0 | 62.5 | 37.5 |
| Llama-3.2-Vision-90B | 6.0 | 0.0 | 36.0 | 4.0 | 15.0 | 77.5 | 22.5 |
| Llava-OneVision-72B | 24.0 | 0.0 | 45.0 | 7.0 | 18.0 | 75.0 | 37.5 |
| QwenVL-72B | 26.0 | 0.0 | 9.0 | 0.0 | 27.0 | 77.5 | 45.0 |

| Model | L010 | L014 | L015 | L016 | L017 | L019 | L020 |
|---|---|---|---|---|---|---|---|
| Claude-3-Opus | 0.0 | 51.0 | 30.0 | 18.0 | 2.0 | 63.33 | 20.0 |
| Claude-3.7-Sonnet | 0.0 | 62.0 | 9.0 | 36.0 | 9.33 | 75.83 | 22.0 |
| Doubao-1.5-vision-pro | 0.0 | 63.0 | 18.0 | 18.0 | 1.33 | 58.33 | 15.0 |
| Gemini-2.0-Flash | 0.0 | 53.0 | 27.0 | 27.0 | 2.67 | 23.33 | 19.0 |
| Gemini-2.5-Flash | 0.0 | 50.0 | 30.0 | 45.0 | 0.0 | 22.5 | 16.0 |
| Gemini-2.5-Pro | 0.0 | 8.0 | 0.0 | 24.0 | 0.0 | 61.67 | 1.0 |
| GPT-4o-2024-11-20 | 0.0 | 67.0 | 24.0 | 54.0 | 2.0 | 61.67 | 18.0 |
| GPT-4.1 | 0.0 | 76.0 | 27.0 | 36.0 | 1.33 | 67.5 | 20.0 |
| GPT-o1 | 0.0 | 67.0 | 12.0 | 45.0 | 6.0 | 57.5 | 23.0 |
| GPT-o3 | 0.0 | 75.0 | 9.0 | 39.0 | 4.0 | 60.0 | 27.0 |
| Grok-2-Vision-12-12 | 0.0 | 59.0 | 21.0 | 27.0 | 0.0 | 65.0 | 14.0 |
| InternVL-3-78B | 0.0 | 59.0 | 24.0 | 45.0 | 3.33 | 69.17 | 22.0 |
| InternVL-2.5-78B | 0.0 | 53.0 | 14.0 | 45.0 | 0.0 | 63.33 | 15.0 |
| Llama-3.2-Vision-90B | 0.0 | 40.0 | 11.0 | 18.0 | 3.33 | 56.67 | 4.0 |
| Llava-OneVision-72B | 2.5 | 51.0 | 36.0 | 21.0 | 0.0 | 55.83 | 11.0 |
| QwenVL-72B | 0.0 | 43.0 | 12.0 | 36.0 | 0.0 | 23.33 | 20.0 |

Table 14: Performance of different models on the Life tasks (Chinese).

| Model | M001 | M002 | M003 | M004 | M006 | M008 | M010 | M018 | M020 | M021 | M022 |
|---|---|---|---|---|---|---|---|---|---|---|---|
| Claude-3-Opus | 76.67 | 6.25 | 28.75 | 53.75 | 60.00 | 42.50 | 42.50 | 27.50 | 30.00 | 15.00 | 25.00 |
| Claude-3.7-Sonnet | 53.33 | 30.00 | 55.00 | 78.75 | 80.00 | 40.00 | 40.00 | 60.00 | 28.00 | 17.50 | 15.00 |
| Doubao-1.5-vision-pro | 60.00 | 26.25 | 67.50 | 90.00 | 80.00 | 22.50 | 45.00 | 40.00 | 34.00 | 20.00 | 22.50 |
| Gemini-2.0-Flash | 53.33 | 45.00 | 82.50 | 82.50 | 70.00 | 45.00 | 22.50 | 45.00 | 24.00 | 22.50 | 22.50 |
| Gemini-2.5-Flash | 66.67 | 45.00 | 90.00 | 85.00 | 70.00 | 60.00 | 17.50 | 52.50 | 28.00 | 22.50 | 25.00 |
| Gemini-2.5-Pro | 0.00 | 11.25 | 22.50 | 78.75 | 80.00 | 0.00 | 25.00 | 7.50 | 0.00 | 0.00 | 0.00 |
| GPT-4o-2024-11-20 | 43.33 | 28.75 | 71.25 | 90.00 | 50.00 | 22.50 | 65.00 | 47.50 | 36.00 | 30.00 | 22.50 |
| GPT-4.1 | 70.00 | 26.25 | 51.25 | 90.00 | 70.00 | 40.00 | 45.00 | 52.50 | 10.00 | 17.50 | 15.00 |
| GPT-o1 | 90.00 | 65.00 | 90.00 | 90.00 | 80.00 | 25.00 | 52.50 | 67.50 | 12.00 | 15.00 | 20.00 |
| GPT-o3 | 90.00 | 56.25 | 78.75 | 90.00 | 80.00 | 60.00 | 60.00 | 75.00 | 26.00 | 22.50 | 20.00 |
| Grok-2-Vision-12-12 | 50.00 | 25.00 | 51.25 | 71.25 | 30.00 | 35.00 | 17.50 | 22.50 | 34.00 | 15.00 | 10.00 |
| InternVL-3-78B | 43.33 | 8.75 | 52.50 | 71.25 | 80.00 | 17.50 | 15.00 | 40.00 | 24.00 | 30.00 | 15.00 |
| InternVL-2.5-78B | 46.67 | 3.75 | 48.75 | 71.25 | 90.00 | 30.00 | 15.00 | 47.50 | 24.00 | 30.00 | 20.00 |
| Llama-3.2-Vision-90B | 53.33 | 15.00 | 50.00 | 75.00 | 30.00 | 15.00 | 5.00 | 30.00 | 18.00 | 27.50 | 5.00 |
| Llava-OneVision-72B | 10.00 | 7.50 | 58.75 | 71.25 | 70.00 | 30.00 | 5.00 | 40.00 | 18.00 | 30.00 | 5.00 |
| QwenVL-72B | 50.00 | 8.75 | 63.75 | 78.75 | 80.00 | 17.50 | 5.00 | 50.00 | 12.00 | 25.00 | 0.00 |

Table 15: Performance of different models on the Materials tasks (English).

| Model | M001 | M002 | M003 | M004 | M006 | M008 | M010 | M018 | M020 | M021 | M022 |
|---|---|---|---|---|---|---|---|---|---|---|---|
| Claude-3-Opus | 56.67 | 7.50 | 37.50 | 62.50 | 30.00 | 27.50 | 20.00 | 37.50 | 24.00 | 22.50 | 27.50 |
| Claude-3.7-Sonnet | 20.00 | 21.25 | 43.75 | 78.75 | 80.00 | 40.00 | 42.50 | 55.00 | 40.00 | 20.00 | 22.50 |
| Doubao-1.5-vision-pro | 16.67 | 28.75 | 67.50 | 90.00 | 80.00 | 20.00 | 15.00 | 35.00 | 18.00 | 22.50 | 27.50 |
| Gemini-2.0-Flash | 60.00 | 47.50 | 82.50 | 78.75 | 80.00 | 25.00 | 0.00 | 32.50 | 28.00 | 25.00 | 10.00 |
| Gemini-2.5-Flash | 66.67 | 45.00 | 90.00 | 85.00 | 70.00 | 60.00 | 17.50 | 52.50 | 24.00 | 22.50 | 22.50 |
| Gemini-2.5-Pro | 0.00 | 2.50 | 11.25 | 78.75 | 70.00 | 15.00 | 65.00 | 0.00 | 0.00 | 0.00 | 0.00 |
| GPT-4o-2024-11-20 | 13.33 | 28.75 | 71.25 | 90.00 | 50.00 | 25.00 | 45.00 | 50.00 | 40.00 | 35.00 | 30.00 |
| GPT-4.1 | 20.00 | 28.75 | 60.00 | 90.00 | 90.00 | 30.00 | 15.00 | 50.00 | 24.00 | 22.50 | 22.50 |
| GPT-o1 | 76.67 | 63.75 | 90.00 | 90.00 | 90.00 | 47.50 | 15.00 | 55.00 | 26.00 | 22.50 | 20.00 |
| GPT-o3 | 86.67 | 35.00 | 78.75 | 90.00 | 80.00 | 47.50 | 60.00 | 57.50 | 28.00 | 12.50 | 22.50 |
| Grok-2-Vision-12-12 | 20.00 | 26.25 | 51.25 | 73.75 | 40.00 | 25.00 | 22.50 | 25.00 | 28.00 | 20.00 | 27.50 |
| InternVL-3-78B | 13.33 | 17.50 | 52.50 | 73.75 | 80.00 | 17.50 | 15.00 | 42.50 | 28.00 | 27.50 | 30.00 |
| InternVL-2.5-78B | 10.00 | 8.75 | 41.25 | 73.75 | 90.00 | 17.50 | 17.50 | 37.50 | 24.00 | 27.50 | 20.00 |
| Llama-3.2-Vision-90B | 10.00 | 16.25 | 48.75 | 73.75 | 18.89 | 25.00 | 45.00 | 15.00 | 24.00 | 25.00 | 17.50 |
| Llava-OneVision-72B | 20.00 | 7.50 | 62.50 | 73.75 | 80.00 | 17.50 | 40.00 | 25.00 | 28.00 | 30.00 | 20.00 |
| QwenVL-72B | 13.33 | 6.25 | 63.75 | 78.75 | 80.00 | 22.50 | 15.00 | 35.00 | 22.00 | 32.50 | 27.50 |

Table 16: Performance of different models on the Materials tasks (Chinese).

| Model | Astronomy | | Chemistry | | Earth | | Life | | Material | | Average | |
|---|---|---|---|---|---|---|---|---|---|---|---|---|
| | en | zh | en | zh | en | zh | en | zh | en | zh | en | zh |
| Gemini-2.0-Flash (1024) | 16.14 | 12.78 | 27.82 | 24.69 | 34.24 | 32.91 | 32.48 | 27.32 | 52.79 | 50.49 | 29.49 | 26.33 |
| Gemini-2.5-Flash (1024) | 24.30 | 24.11 | 23.67 | 23.47 | 31.99 | 30.53 | 25.03 | 25.10 | 56.39 | 55.90 | 28.03 | 27.63 |
| Gemini-2.5-Pro (1024) | 5.13 | 6.08 | 2.07 | 2.28 | 2.52 | 3.84 | 19.73 | 22.35 | 28.69 | 27.70 | 8.04 | 8.96 |
| Gemini-2.0-Flash (16384) | 15.68 | 19.20 | 40.23 | 28.38 | 31.15 | 27.23 | 34.64 | 26.07 | 55.47 | 54.28 | 33.75 | 27.87 |
| Gemini-2.5-Flash (16384) | 22.22 | 22.43 | 46.51 | 39.88 | 36.35 | 32.05 | 44.15 | 37.04 | 59.56 | 63.21 | 40.29 | 36.12 |
| Gemini-2.5-Pro (16384) | 27.57 | 28.13 | 54.15 | 48.81 | 40.27 | 39.50 | 47.35 | 38.56 | 70.49 | 72.13 | 46.15 | 42.77 |

Table 17: Experimental results of MLLMs on different disciplines using different maximal number of generated tokens.

| Model | L1 | | L2 | | L3 | | Exact Match | | | | Open Question | | | | MCQ | |
|---|---|---|---|---|---|---|---|---|---|---|---|---|---|---|---|---|
| | | | | | | | IoU | | LLM score | | Bertscore | | LLM score | | LLM score | |
| | en | zh | en | zh | en | zh | en | zh | en | zh | en | zh | en | zh | en | zh |
| Gemini-2.0-Flash (1024) | 41.43 | 36.46 | 26.86 | 23.70 | 22.00 | 21.60 | 0.91 | 0.44 | 21.38 | 19.28 | 0.70 | 0.65 | 35.67 | 25.19 | 39.47 | 37.39 |
| Gemini-2.5-Flash (1024) | 37.04 | 37.57 | 27.89 | 26.98 | 14.96 | 15.20 | 0.99 | 0.96 | 24.45 | 24.45 | 0.688 | 0.688 | 30.00 | 30.10 | 32.71 | 31.51 |
| Gemini-2.5-Pro (1024) | 17.46 | 18.78 | 6.16 | 6.94 | 1.36 | 2.24 | - | - | 8.11 | 8.60 | 0.548 | 0.558 | 12.50 | 15.96 | 6.30 | 6.94 |
| Gemini-2.0-Flash (16384) | 36.53 | 31.63 | 29.15 | 24.29 | 25.36 | 17.68 | 1.44 | 0.15 | 23.29 | 19.79 | 0.668 | 0.631 | 21.58 | 16.51 | 44.98 | 36.73 |
| Gemini-2.5-Flash (16384) | 42.92 | 42.62 | 35.86 | 30.95 | 27.12 | 22.40 | 3.17 | 2.68 | 30.57 | 29.10 | 0.675 | 0.672 | 27.70 | 28.17 | 48.68 | 39.47 |
| Gemini-2.5-Pro (16384) | 52.48 | 52.48 | 40.58 | 40.58 | 27.68 | 27.68 | 16.00 | 10.03 | 37.33 | 35.17 | 0.704 | 0.711 | 37.46 | 35.87 | 49.68 | 44.54 |

Table 18: Experimental results of MLLMs on different disciplines using different maximal number of generated tokens.

| MLLM Judge | GPT-4o | | Claude-3.7-Sonnet | | Deepseek-V3 | | Grok-2 | | Average | |
|---|---|---|---|---|---|---|---|---|---|---|
| | en | zh | en | zh | en | zh | en | zh | en | zh |
| Grok-2-Vision-12-12 | 24.97 | 25.10 | 33.82 | 33.29 | 22.99 | 23.46 | 25.19 | 23.32 | 26.74 | 26.29 |
| GPT-o3 | 34.08 | 31.60 | 45.70 | 46.25 | 35.82 | 35.33 | 34.05 | 34.28 | 37.41 | 36.86 |
| Claude-3.7-Sonnet | 31.52 | 29.23 | 35.43 | 31.42 | 32.07 | 25.66 | 30.53 | 28.49 | 32.38 | 28.7 |
| Qwen2.5-VL-72B | 24.17 | 21.51 | 23.51 | 23.08 | 23.23 | 22.61 | 23.11 | 19.92 | 23.50 | 21.78 |
| InternVL-3-78B | 26.52 | 24.30 | 25.58 | 23.38 | 26.40 | 24.79 | 26.48 | 23.94 | 26.24 | 24.10 |
| Llava-OneVision-72B | 22.10 | 23.39 | 22.42 | 22.56 | 22.42 | 22.97 | 22.81 | 22.12 | 22.43 | 22.76 |

Table 19: Performance comparison of different MLLMs evaluated by various LLM judges.