# OpenReview forum: "Scientists' First Exam: Probing Cognitive Abilities of MLLM via Perception, Understanding, and Reasoning"
_NeurIPS.cc/2025/Datasets_and_Benchmarks_Track — NeurIPS 2025 Datasets and Benchmarks Track poster_

### Official Review · Reviewer_HyFJ · 2025-07-02

**Rating:** 5
**Confidence:** 4

**Summary:**

This paper introduces the Scientists’ First Example (SFE) benchmark, an evaluation framework designed to assess the scientific cognitive capabilities of multimodal large language models (MLLMs) across perception and reasoning tasks. SFE comprises 830 expert-verified visual question answering (VQA) pairs spanning 66 tasks in five high-value scientific disciplines: Astronomy, Chemistry, Earth Science, Life Science, and Materials Science. Each task is constructed from native scientific data formats and formulated as bilingual VQA pairs, aligned with three cognitive capacity levels.

**Dataset Code Accessibility:**

Yes

**Ethical Considerations:**

No, there are no or only very minor ethics concerns

**Final Justification:**

The authors have addressed most of my concerns, and the method demonstrates solid technical contributions and empirical results. I recommend acceptance.

**Limitations Weaknesses:**

I have two questions:

1. The diversity of raw data formats may introduce practical usability challenges. Although the inclusion of 17 distinct scientific data formats enhances the realism of SFE, this diversity complicates model generalization and deployment.

2. Regarding lines 275-281 discussing overfitting, I did not fully understand why overfitting would occur, since these models have not been trained on the SFE dataset. However, my interpretation is that the authors suggest the pre-training corpus of large models might have already included such information. It would be helpful if the authors could briefly clarify the reason or their hypothesis for this observation.

**Strengths Contributions:**

Strength:

1.	The paper demonstrates exceptional clarity in its writing and visual presentation.

2.	SFE establishes a substantially higher difficulty threshold compared to prior scientific benchmarks(e.g., ScienceQA, MMLU).

3.	The inclusion of bilingual (English & Chinese)  prompts and answers ensures robust assessment of multilingual reasoning abilities, addressing global deployment contexts.

4.	By structuring tasks into three cognitive levels (L1: perception, L2: understanding, L3: reasoning), SFE enables better assessment of MLLM capabilities.

---

> ### Author Rebuttal · Authors · 2025-07-30
>
> We sincerely thank Reviewer HyFJ for the constructive comments and suggestions on SFE. We make responses as follows.
>
> > **Q1**: The diversity of raw data formats may introduce practical usability challenges.
>
> **A1**: Thank you for the constructive feedback. In SFE, we deliberately preserved this heterogeneity of data formats because **it reflects the reality of laboratory work.** Thus, we believe limiting SFE to pre-rendered images would **hide a key difficulty and bias MLLMs toward a narrow use case.**
>
> However, we fully understand that these heterogeneous data format could greatly complicate the benchmarking process. To reduce the burden, **for every format, we have provided a script that converts the raw file into a standard RGB image that is verified by domain experts to have sufficient information for question answering, so current MLLMs can be evaluated immediately.**
>
> Note that the collected and released data formats in SFE correspond to the most frequently used data formats in astronomy, chemistry, earth, life and material. Additionally, the structure of SFE is modular, permitting future additions or removals with ease. **By releasing both rendered and native files in SFE, we aim to catalyze research on MLLMs that can reason over authentic scientific data: an ability we regard as crucial yet still under-explored.**
>
> > **Q2**: Why overfitting would occur?
>
> **A2**: Thank you for the thoughtful feedback, and we are sorry to make the reviewer confuse. What we intended to describe in lines 275–281 is **not overfitting to the SFE benchmark itself (none of the evaluated MLLMs were ever trained on SFE), but rather a degradation that resembles overfitting when model size grows.**
>
> Specifically, the performance of some MLLMs (*e.g.*, Qwen) plateaus or even declines as their model sizes increase. Our hypothesis is that, during pre-training, the scientific corpus size does not scale proportionally with the overall corpus size. Therefore, the larger MLLMs learn from a smaller relative fraction of domain-specific knowledge, and **may end up memorizing instead of acquiring generalizable scientific reasoning skills, which hurts transfer to SFE.** We will revise the paper to clarify this point and replace the potentially misleading term “overfitting” with a more precise description.
>
> We sincerely hope our response can address the reviewer's concerns.

---

> > ### Comment · Reviewer_HyFJ · 2025-08-04
> >
> > Thanks for your reply! You have addressed all of my concern, but as i have already give a positive score, I decide to maintain my score as 5.

---

> > > ### Author Response · Authors · 2025-08-04
> > >
> > > Thank you for your helpful feedback and support! We are very pleased to see that your concerns have been addressed. We will carefully revise the paper as you suggested. If there are any further questions or comments, please feel free to post them here for discussion!

---

### Official Review · Reviewer_LSMX · 2025-07-03

**Rating:** 4
**Confidence:** 3

**Summary:**

This paper presents the Scientists’ First Exam (SFE), a new benchmark designed to evaluate the cognitive abilities of Multimodal Large Language Models (MLLMs) for scientific tasks. SFE focuses on three cognitive levels: signal perception, attribute understanding, and comparative reasoning. The benchmark contains 830 bilingual (EN/ZH) expert-verified VQA pairs across 66 multimodal scientific tasks covering five disciplines. The authors conduct extensive evaluations on 16 state-of-the-art MLLMs.

**Additional Feedback:**

## Questions for the Authors

1. In Figure 1(d), SFE shows a much higher average number of images per QA compared to other benchmarks. Is this design choice reasonable and how do you ensure that this does not introduce unintended bias when comparing model performance?
2. In Table 1, the task number for SciBench and other benchmarks is marked as "N/A". How is this defined? Is the task number inherently unavailable or was it not reported by the original benchmark authors?
3. In Table 4, the Gemini-2.5-Pro results for the IoU metric are missing. Does this indicate that the model failed to generate valid outputs for those tasks, or were these evaluations not conducted?
4. While the benchmark claims expert annotation and validation, could the authors elaborate on the specific annotation instructions and quality control mechanisms? How do you ensure consistent and unbiased annotations, especially across different cognitive levels and scientific disciplines?

## Typos & Suggestions

- Consider providing a more transparent discussion of benchmark limitations beyond just mentioning future work, such as the potential domain biases introduced by expert-selected tasks.
- It would be helpful to explicitly discuss why certain scientific directions (e.g., Biology vs. Earth science) differ significantly in task difficulty and model performance, as revealed in the results.

**Dataset Code Accessibility:**

Yes

**Ethical Considerations:**

No, there are no or only very minor ethics concerns

**Limitations Weaknesses:**

1. **Task Quantity Limitation**: Although there are 830 VQA pairs, the actual number of distinct scientific tasks is only 66. This relatively small number may limit the ability to comprehensively evaluate models across diverse scientific directions, especially for cross-domain comparisons.
2. **Inconsistent Data Distribution**: Figure 1(d) reveals a significant imbalance in image quantity per QA across benchmarks, with SFE having a notably higher number of images per QA. This raises concerns about data uniformity and whether model performance comparisons with other benchmarks are entirely fair.
3. **Benchmark Comparison Gaps**: In Table 1, some benchmark statistics such as task counts (e.g., SciBench) are missing. It is unclear whether these numbers are truly unavailable or simply omitted, which weakens the clarity of the benchmark positioning.

**Strengths Contributions:**

1. **Expert-Driven Task Design**: The benchmark benefits from domain expert involvement across task design, data collection, and annotation, which enhances its scientific validity and quality.
2. **Cognitive-Level Granularity**: The division into three cognitive levels provides a more fine-grained perspective on different capabilities of MLLMs, rather than focusing solely on general knowledge understanding.
3. **Multilingual and Multimodal Coverage**: The benchmark supports both English and Chinese and covers diverse scientific modalities such as spectra, molecular structures, and geospatial data, making it relevant for real-world scientific applications.

---

> ### Author Rebuttal · Authors · 2025-07-30
>
> We sincerely thank Reviewer LSMX for the constructive comments and suggestions on SFE. We make responses as follows.
>
> > **Q1**: Why limited number of distinct scientific tasks?
>
> **A1**: Thank you for highlighting this limitation. Our primary goal was not to maximise task count but to create, for the first time, **a cross-disciplinary, high-quality, expert-curated scientific benchmark that evaluates cognitive ablilities of current MLLMs**. Specifically, in SFE, each task was proposed and verified by **at least** one domain expert. However, since most experts master only one or two sub-topics, we had to consult more than 60 experts and eventually secure the active participation of about 20 of them. Given this finite expert budget, we inevitably faced a trade-off between quality, diversity and quantity. **We chose quality and diversity, believing that they are what the community now lacks most.** The resulting 66 tasks span astronomy, chemistry, earth, life and material, and require heterogeneous cognitive abilities, enabling meaningful cross-domain comparison. **By releasing both the data and the task construction pipeline, we hope SFE will serve as a template the community can readily enlarge to cover additional scientific directions in future work.**
>
> > **Q2**: Imbalance in image quantity per QA across benchmarks.
>
> **A2**: We appreciate the feedback. We argue that **the imbalance is an intrinsic property of the underlying scientific tasks rather than an arbitrary design choice.** For instance, in IUPAC Name Recognition Task (C003), a single molecular diagram is usually sufficient for solving the problem, whereas in Phase Identification Task (M020), a correct answer typically emerges only after comparing several X-ray diffraction patterns of related materials. **Enforcing an equal number of images across such heterogeneous tasks would remove information that scientists routinely consult**, and would render many VQAs unsolvable or unrealistic.
>
> Regarding fair comparison with other benchmarks, our evaluation is performed at the VQA level: a MLLM receives exactly the set of images that human experts deem necessary for that question, and metrics are averaged over VQAs instead of images. Hence, **a task that happens to require more images does not carry disproportionate weight in the final score.** Fig. 1(d) therefore should be interpreted as **evidence of the wider task spectrum covered by SFE rather than as an imbalance that biases performance measurements.** We are sorry to make the reviewer confuse, and will add this clarification to the revision.
>
> > **Q3**: Missing benchmark statistics in Table 1.
>
> **A3**: We thank the reviewer for highlighting the omission. Note that, in Table 1, only ChemBench and CURIE provide an official task taxonomy. Their task distributions is presented as follows, and will be added to the revision.
>
> ChemBench:
> | Task Name | # Examples |
> |---|---|
> | Name Prediction | 100 |
> | Property Prediction | 100 |
> | Yield Prediction | 100 |
> | Reaction Prediction | 100 |
> | Retrosynthesis | 100 |
> | Molecule Design | 100 |
> | Molecule Captioning | 100 |
> | Reagents Selection | 100 |
>
> CURIE:
> | Task Name | # Sub-tasks | # Examples |
> |---|---|---|
> | Density Functional Theory | 3 | 222 (3 x 74) |
> | Material Property Value Extraction | - | 17 |
> | Hartree-Fock | 2 | 53 (15 + 38) |
> | Error-Correction Zoo | - | 65 |
> | Geospatial Dataset Extraction | - | 19 |
> | Biodiversity Georeferencing | - | 138 |
> | Protein Sequence Reconstruction | - | 64 |
>
> Additionally, we will add VQA-pair counts of each benchmark to Table 1 to further enhance the transparency.
>
> | Benchmark | MMMU | ScienceQA | SCIBENCH | CMMU | ChemBench | SuperGPQA | SciEval | HLE | CURIE | SFE |
> |---|---|---|---|---|---|---|---|---|---|---|
> | QA Pairs | 11.5k | 21.2k | 869 | 3.6k | 800 | 26.5k | 15.9k | 3k | 580 | **830** |
>
> > **Q4**: Imbalance in image quantity per QA across benchmarks.
>
> **A4**: We thank the reviewer for the feedback. Please see **A2** for this question.
>
> > **Q5**: What is "N/A" in Table 1?
>
> **A5**: We thank the reviewer for the feedback. Please see **A3** for this question.
>
> > **Q5**: The IoU score of Gemini-2.5-Pro is missing in Table 4, why?
>
> **A5**: Thank you for the constructive feedback. The IoU metric for Gemini-2.5-Pro is missing in Table 4 because **the model was unable to generate outputs in the required format as specified by the task instructions.** Consequently, we could not compute the IoU score for this model. We will revise the caption of Table 4 to provide better clarification in the final version.
>
> > **Q6**: Elaboration on specific annotation instructions and quality control mechaisms.
>
> **A6**: Thank you for this thoughtful question. To ensure high-quality and unbiased annotations, **we developed detailed annotation guidelines in collaboration with domain experts**, as described in Section 3.1. These guidelines clearly defined **question types, expected answer formats, data requirements, and included tailored instructions and examples for each task**, thereby promoting consistent understanding. Moreover, for each task, VQA pairs were constructed by **1-3** sub-domain experts who collected or generated scientific raw data (*e.g.*, molecular structures, satellite images, etc.) and carefully formulated corresponding questions and answers. Annotated data were visualized into images to ensure compatibility with MLLMs, and the experts verified that **all necessary information for answering the questions was visually present and unambiguous in visualizations.**
>
> Quality control involved **expert cross-review within the same domain** followed by rule-based automated checks to assess correctness, clarity, and adherence to the guidelines. **Any disagreements or ambiguities were resolved through discussion or, if needed, adjudication by a third expert.** Additionally, **we held regular calibration meetings across sub-domains to align annotator understanding and practices.** This structured, multi-stage process across varied disciplines and cognitive levels helped us maintain consistent, unbiased, and high-quality annotations throughout the benchmark.
>
> We sincerely hope our response can address the reviewer's concerns. We will further improve clarification on these details in the revision.
>
> > **Q7**: Providing a more transparent discussion of benchmark limitations.
>
> **A7**: We appreciate the reviewer’s valuable suggestion regarding the transparency of benchmark limitations. Indeed, our benchmark construction heavily depends on expert annotation, which inevitably introduces domain biases, as the selected tasks and question types may reflect the specific interests and expertise of the annotators. This constraint means certain relevant tasks could be underrepresented or overlooked. Additionally, due to the limited number of experts, there remains a possibility of annotation errors that go undetected by all annotators.
>
> To mitigate these issues, **we plan to involve a broader and more diverse set of experts in future updates, and we have made our benchmark openly available with a feedback mechanism to actively encourage community contributions in identifying and correcting errors.** We will explicitly discuss these limitations in the revised version for greater transparency.
>
> It is important to emphasize, however, that one of our key contributions lies in the **successful development and implementation of a construction pipeline for creating a high-quality, cross-disciplinary, expert-curated scientific benchmark**. By openly releasing both our construction pipeline and the SFE benchmark curated by 20+ domain experts, **we aim to provide a readily extensible template for future research, facilitating the inclusion of additional scientific domains.**
>
> > **Q8**: Why certain scientific directions differ significantly in task difficulty and model performance.
>
> **A8**: Thank you for your valuable feedback. We acknowledge that there are significant differences in task difficulty and model performance across different disciplines in Table 3. We believe several factors contribute to these differences:
>
>  * **the availability and diversity of training data play a crucial role**: for example, Astronomy tasks suffer from data scarcity due to less publicly available resources, while fields like Material Science benefit from abundant, well-curated datasets (*e.g.*, the Materials Project database). This likely leads to **discrepancies in model capabilities**, as MLLMs generally perform better in domains with richer exposure during pre-training.
>
>  * **the question type varies across disciplines**. Eor example, Astronomy experts mainly submitted MCQ and Exact-Match questions, while Material Science experts focused more on analytical and open-ended questions, as shown in Table 6. **We believe these differences in question type reflect the intrinsic nature of each discipline.** Consequently, cross-discipline performance gaps emerge, as MLLMs generally perform better on MCQ and open-ended questions compared to Exact-Match questions, as demonstrated in Table 4.
>
> We will add more dicussions about this issue in the revision for better transparency. Importantly, we have conducted additional human and multi-judge evaluations during the rebuttal phase (Please see **A1** of Reviewer z17H), confirming the consistency and robustness of our performance ranking in Table 3. Thank you again for helping us strengthen our work.
>
> We sincerely hope our response can address the reviewer's concerns.

---

### Official Review · Reviewer_z17H · 2025-07-03

**Rating:** 5
**Confidence:** 3

**Summary:**

This paper presents the Scientists’ First Exam (SFE) benchmark that evaluates MLLMs on scientific tasks: signal perception, attribute understanding and comparative reasoning.

They then evaluate 16 MLLMs on the proposed benchmark.

**Dataset Code Accessibility:**

Yes

**Ethical Considerations:**

No, there are no or only very minor ethics concerns

**Limitations Weaknesses:**

One limitation is that in the metrics the authors only consider BERTScore and LLM-as-a-Judge. It would be nice to have part of the evaluations done by human annotators as well. Or, have Judges based on different LLMs to limit the amount of bias.

**Strengths Contributions:**

- The benchmark is very broad: spans 66 multimodal tasks across five disciplines and two languages. In total 830 VQA pairs.
- Run evaluations across 16 SOTA MLLMs

---

> ### Author Rebuttal · Authors · 2025-07-30
>
> We sincerely thank Reviewer z17H for the constructive comments and suggestions on SFE. We make responses as follows.
>
> > **Q1**: More evaluations to limit the amount of bias.
>
> **A1**: Thank you for the suggestion. We have run **two** additional evaluations and will add them in the revision.
>
> (1) **Human evaluation**: First, we randomly sample 10 questions from SFE (due to limited time) and collect the corresponding responses of six MLLMs (three close weight: GPT-o3, Claude-3.7-Sonnet, Grok-2-Vision-12-12; three open weight: InternVL-3-78B, Qwen2.5-VL-72B, Llava-OneVision-72B). Then, these responses are anonymized and provided to domain experts for ranking. The resulting aggregated order is:
>
> ```
> GPT-o3 ≈ Claude-3.7-Sonnet ≈ InternVL-3-78B > Grok-2-Vision-12-12 > Llava-OneVision-72B > Qwen2.5-VL-72B,
> ```
>
> **which is mostly consistent with the order given by the GPT-4o judge in the paper.** (GPT-o3 > Claude-3.7-Sonnet > InternVL-3-78B > Grok-2-Vision-12-12 > Qwen2.5-VL-72B > Llava-OneVision-72B)
>
> (2) **Multiple LLM judges**: We re-scored all responses of the above mentioned 6 MLLMs with **3 additional judges**: Claude-3.7-Sonnet, Deepseek-V3, and Grok-2. The results are demonstrated as follows:
>
> | MLLM \ Judge | GPT-4o | GPT-4o | Claude-3.7-Sonnet | Claude-3.7-Sonnet | Deepseek-V3 | Deepseek-V3 | Grok-2 | Grok-2 | Avg | Avg |
> |---|---|---|---|---|---|---|---|---|---|---|
> |  | en | zh | en | zh | en | zh | en | zh | en | zh |
> | Grok-2-Vision-12-12 | 24.97 | 25.10 | 33.82 | 33.29 | 22.99 | 23.46 | 25.19 | 23.32 | 26.74 | 26.29 |
> | GPT-o3 | 34.08 | 31.60 | 45.70 | 46.25 | 35.82 | 35.33 | 34.05 | 34.28 | **37.41** | **36.86** |
> | Claude-3.7-Sonnet | 31.52 | 29.23 | 35.43 | 31.42 | 32.07 | 25.66 | 30.53 | 28.49 | *32.38* | *28.7* |
> | Qwen2.5-VL-72B | 24.17 | 21.51 | 23.51 | 23.08 | 23.23 | 22.61 | 23.11 | 19.92 | 23.50 | 21.78 |
> | InternVL-3-78B | 26.52 | 24.30 | 25.58 | 23.38 | 26.40 | 24.79 | 26.48 | 23.94 | 26.24 | 24.10 |
> | Llava-OneVision-72B | 22.10 | 23.39 | 22.42 | 22.56 | 22.42 | 22.97 | 22.81 | 22.12 | 22.43 | 22.76 |
>
> As it can be seen, **all judges produce similar ranking**. These results show that:
>
> * our conclusions are **robust** to assessments of both human and multiple LLM judges.
> * using several heterogeneous LLM judges **successfully mitigates possible single-judge bias.**
>
> We will incorporate the new tables and analysis in the revision.
>
> We sincerely hope our response can address the reviewer's concerns.

---

> > ### Comment · Reviewer_z17H · 2025-08-08
> >
> > Thanks! This seems like a valuable addition.

---

### Official Review · Reviewer_AMSn · 2025-07-10

**Rating:** 4
**Confidence:** 4

**Summary:**

This paper proposed the Scientists’ First Example (SFE), a bilingual (zh and en) benchmark comprising 830 VQA pairs across 66 tasks, spanning 5 science disciplines. The SFE was constructed in close collaboration with domain experts in the processes of task design, answer collection as well as verification. The benchmark addresses the emerging area of MLLMs in the context of LLM for Science, where high-quality evaluation benchmarks play a crucial role in driving progress。

**Dataset Code Accessibility:**

Yes

**Dataset Code Comments:**

The authors provide Hugging Face links to the dataset and code.

**Ethical Considerations:**

No, there are no or only very minor ethics concerns

**Final Justification:**

The authors have addressed almost all of my concerns. Considering the limited number of QA pairs and tasks per domain, I have decided to maintain my score.

**Limitations Weaknesses:**

1. The number of tasks per domain is relatively limited. Including the distribution of tasks and VQA pairs per benchmark in Table 1 would improve transparency.
2. More details are required for task construction part, e.g. Are experts from academia or /and industry? How were they recruited? Did they get paid? How many experts for each tasks?
3. In Table 4, the performance of some models does not consistently decline as task difficulty increases. This irregularity needs to be addressed—what might be the underlying reason?

Minor:
1) It should be 19 scientific directions in Fig 1(b).
2) 2-> Fig. 2 in line 125.
3) Wrong reference of Fig. 5 in line 289.

**Strengths Contributions:**

1. The benchmark includes a diverse set of task types across five scientific disciplines, with particular emphasis on challenges such as long visual context..
2. The involvement of domain experts in the collection of data benchmarks ensures data quality and credibility.
3. The tasks vary in complexity and domain coverage, requiring different capabilities from MLLMs. This makes the benchmark a valuable reference for evaluating and improving MLLMs on scientific reasoning tasks.

---

> ### Author Rebuttal · Authors · 2025-07-30
>
> We sincerely thank Reviewer AMSn for the constructive comments and suggestions on SFE. We make responses as follows.
>
> > **Q1**: Why limited number of tasks per domain?
>
> **A1**: Thank you for highlighting this limitation. Our primary goal was not to maximise task count but to create, for the first time, **a cross-disciplinary, high-quality, expert-curated scientific benchmark that evaluates cognitive ablilities of current MLLMs**. Specifically, in SFE, each task was proposed and verified by **at least** one domain expert. However, since most experts master only one or two sub-topics, we had to consult more than 60 experts and eventually secure the active participation of about 20 of them. Given this finite expert budget, we inevitably faced a trade-off between quality, diversity and quantity. **We chose quality and diversity, believing that they are what the community now lacks most.** The resulting 66 tasks span astronomy, chemistry, earth, life and material, and require heterogeneous cognitive abilities, enabling meaningful cross-domain comparison. **By releasing both the data and the task construction pipeline, we hope SFE will serve as a template the community can readily enlarge to cover additional scientific directions in future work.**
>
> > **Q2**: Why limited statistics in Table 1?
>
> **A2**: We thank the reviewer for the helpful suggestion and will revise Table 1 to expose both VQA-pair counts and, when available, task distributions. Below are the detailed statistics of VQA-pair counts.
>
> | Benchmark | MMMU | ScienceQA | SCIBENCH | CMMU | ChemBench | SuperGPQA | SciEval | HLE | CURIE | SFE |
> |---|---|---|---|---|---|---|---|---|---|---|
> | QA Pairs | 11.5k | 21.2k | 869 | 3.6k | 800 | 26.5k | 15.9k | 3k | 580 | **830** |
>
> Note that, in Table 1, only ChemBench and CURIE provide an official task taxonomy. Their task distributions is presented as follows, and will be added to the revision.
>
> ChemBench:
> | Task Name | # Examples |
> |---|---|
> | Name Prediction | 100 |
> | Property Prediction | 100 |
> | Yield Prediction | 100 |
> | Reaction Prediction | 100 |
> | Retrosynthesis | 100 |
> | Molecule Design | 100 |
> | Molecule Captioning | 100 |
> | Reagents Selection | 100 |
>
> CURIE:
> | Task Name | # Sub-tasks | # Examples |
> |---|---|---|
> | Density Functional Theory | 3 | 222 (3 x 74) |
> | Material Property Value Extraction | - | 17 |
> | Hartree-Fock | 2 | 53 (15 + 38) |
> | Error-Correction Zoo | - | 65 |
> | Geospatial Dataset Extraction | - | 19 |
> | Biodiversity Georeferencing | - | 138 |
> | Protein Sequence Reconstruction | - | 64 |
>
> > **Q3**: More details about the task construction part.
>
> **A3**: We appreciate the reviewer’s request. All annotators are **senior researchers from academia**. Specifically, we contacted around **60** domain experts; Around **50** of them replied, and after a follow-up video call explaining our goals, around **20** volunteered to join the SFE team. They receive no monetary compensation and instead participate as co-authors. For every individual task, **1–3 of these experts defined the task and carried out the labeling, with results cross-checked by another team member.**
>
> > **Q4**: Irregularity in Table 4.
>
> **A4**: Thank you for pointing this out. In SFE, L1 to L3 are ordered by **required cognitive abilities, not by empirical difficulty.** Hence a performance decrease is not garenteed (*e.g.*, predicting chemical properties (L2) is probably more difficult than determining crystal group from multiple views (L3)).
>
> Empirically we observe a clear pattern:
>
> • Chat models usually lose performance from L1→L3.
>
> • Reasoning models keep almost the same score on L3 as on L1.
>
> Thus we argue that the "irregularity" is actually evidence that **reasoning capability is essential for higher-level questions and can compensate for the increased cognitive demand.** We will add this clarification to the revision.
>
> > **Q5**: Other minor issues.
>
> **A5**: Thank you for noticing these issues. We will revise and incorporate them in the final version.
>
> We sincerely hope our response can address the reviewer's concerns.

---

> > ### Comment · Reviewer_AMSn · 2025-08-08
> >
> > Thank you for your reply! You have addressed almost all of my concerns. Considering the limited number of QA pairs and tasks per domain, I have decided to maintain my score.

---

### Decision · Program_Chairs · 2025-09-18

**Decision:**

Accept (poster)

**Comment:**

This paper introduces the Scientists’ First Example (SFE) benchmark, an evaluation framework designed to assess the scientific cognitive capabilities of multimodal large language models (MLLMs) across perception and reasoning tasks. SFE comprises 830 expert-verified visual question answering (VQA) pairs spanning 66 tasks in five high-value scientific disciplines: Astronomy, Chemistry, Earth Science, Life Science, and Materials Science. Each task is constructed from native scientific data formats and formulated as bilingual VQA pairs, aligned with three cognitive capacity levels.

Strengths:
- Expert-in-the-loop task design, annotation, and verification, make the benchmark more credible
- Cross-disciplinary, multimodal, bilingual coverage increases benchmark utility and realism.
- Novel cognitive-level taxonomy (L1–L3) enables fine-grained diagnosis of model abilities beyond factual knowledge.

Weaknesses:
- Limited numbers of tasks per domain, and relatively small scale.
- Ealuation heavily relies on reliance on BERTScore and single LLM-as-a-judge.
- Some clarity issues in task construction details and benchmark comparisons across datasets.

The author's rebuttal addressed the concerns on the clarity issues and shows strong consistency between human expert and multi LLM eval. While the small scale issue remains, this work provides a timely and potentially impactful in a important field of LLM for science.